# Machine-Generated Text Detection Requires Fewer Machine-Human Mixed Texts

## Abstract

Machine-generated texts (MGTs) of large language models (LLMs) show significant potential in many fields but also pose challenges like fake news propagation and phishing, highlighting the need for MGT detection. Most paragraph-level detection methods implicitly assume that MGTs are entirely machine-generated and ignore the scenarios where only part of the MGT is machine-generated or inconsistent with human-generated text. To this end, this paper first reveals the prevalence of implicit human-machine mixed texts, which contain subtexts that are common to human texts, and then theoretically analyzes their impact on detection. Based on our theoretical findings, we develop a stacked detection enhancement framework decoupled from the detection model, which involves revisiting the detection optimization objective and the balance between feasibility and efficiency during optimization. Extensive experiments demonstrate its superior improvements over existing detectors. Notably, our boosting strategy can also work in a training-free manner, offering flexibility and scalability. The source code is available at `https://anonymous.4open.science/r/MGTD`.

## 1 Introduction

The rapid development of large language models (LLMs) (Achiam et al., 2023; Radford et al., 2019) has led to a surge in Machine-generated texts (MGTs). While these texts have shown significant potential in many applications, they have also posed severe challenges to fake news propagation (Zellers et al., 2019), phishing (Hong, 2012), and academic fraud (Alshurafat et al., 2024). For example, cybercriminals can create realistic phishing emails to commit fraud or generate fake product reviews to manipulate consumer decisions. These risks highlight the need for effective MGT detection to ensure safe and transparent AI systems. This paper will focus on paragraph-based detection, which can fully utilize contextual information to provide robust detection (Tulchinskii et al., 2024).

Feature-engineering detection methods identify MGT by using distinctive properties of generated text, e.g., output log probability (Mitchell et al., 2023; Solaiman et al., 2019), objectivity and sentiment of the language (Guo et al., 2023), cross entropy (Guo et al., 2024), and intrinsic dimensions (Tulchinskii et al., 2024). However, such methods require substantial expert knowledge and experience. Moreover, due to the complexity of textual data, manually extracted features based on limited data often fail to fully capture intricate patterns and structures, thus, leading to poor generalization across various generative models. By contrast, model-based detection methods use entire texts as inputs, allowing detectors to implicitly learn distinguishing features during training. These approaches are more flexible than feature engineering methods and have gained more attention, such as, energy-based models (Tulchinskii et al., 2024), small language models (Mireshghallah et al., 2023), LLM (Verma et al., 2024), and graph neural networks (Zhong et al., 2020). Besides, the representation quality of data is crucial for learning detection models, such as using pre-trained text features (Crothers et al., 2022) and probability lists from open-source LLMs (Wang et al., 2023).

However, these methods implicitly assume texts are entirely human- or machine-generated and ignore the possibility of mixed texts, where only parts are MGTs or inconsistent with human text. Under the circumstances, at least three key research questions have yet remain to be answered:

- **RQ1**: How common are human-machine mixed texts, and is it possible for comment texts to exist that are consistent with those of humans, even if the text is entirely machine-generated?

- **RQ2**: If the answer to RQ1 is yes, what are the challenges such mixed text brings to detection, and what benefits may we achieve by solving these challenges?

- **RQ3**: For the challenges of RQ2, how can we refine the detection model to overcome them?

This paper aims to study machine-generated text detection by solving these three issues.

First, although some research (Wang et al., 2023; Zhang et al., 2024b; Wang et al., 2024) has begun to focus on explicit mixed text, that is, text completed through human-machine collaboration, this paper reveals the prevalence of implicit mixed texts (**RQ1**, Section 2.2). Specifically, *even if the text is entirely machine-generated, LLMs, with their powerful generation capabilities, can generate texts consistent with human writing.* The human-likeness of MGT is verified by examining the Jaccard Similarity based on sentence words between humans and LLMs, as shown

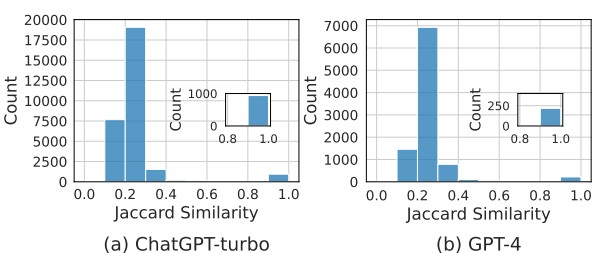

Figure 1: Jaccard similarity of sentence words between human-written texts and MGTs (ChatGPT and GPT-4). Texts with 100% similarity and same order (in fact, all texts with 100% similarity in our experiments met this condition) indicate the existence of implicit mixed texts.

in Fig. 1 (See Section 2.2 for detailed discussion). The consistent text present in MGT can be considered (implicit) mixed text to some extent. This insight suggests that mixed texts are far more prevalent than we had anticipated.

The common existence of mixed texts requires us to analyze their impact on detection (**RQ2**, Section 2.3). Therefore, we theoretically reveal the sample complexity of the best possible detector to achieve an AUROC of $\epsilon$. The results indicate that *sample complexity is proportional to the proportion of human-generated text within MGTs*, and the complexity is lowest when this proportion is 0. This implies that the existence of mixed texts hinders detection.

To tackle the issues exposed by RQ2 (**RQ3**, Section 3), we first propose a theoretical filtering-based enhancement framework and then gradually model it into a stacked enhancement framework, which includes identifying the suitable optimization objective that can address the above challenges and proposed a hard-EM-inspired approach that aids in balancing feasibility and efficiency of detection. Specifically, the proposed stacked framework performs self-enhancement through an iterative collaboration: (1) in the E-step, the detector filters out "human" sequences, yielding a simpler post-filter distribution. (2) In the M-step, our framework refines the same detector based on the remaining texts. The improved detector then serves as the detector for the next E-step. This interwoven process leads to self-enhancement. Extensive experiments demonstrate the proposed framework's boosting effectiveness across various LLMs.

Our contribution mainly lies in (1) revealing the existence of implicit human-machine mixed texts even if the text is entirely machine-generated, (2) theoretically proving its detrimental impact on detection, and (3) proposing a theory-inspired enhancement framework to boost detection. A detailed discussion of our contribution is given in Appendix B.

## 2 MGT DETECTION REQUIRES FEWER MIXED TEXTS

### 2.1 PRELIMINARY

**Text Data Definition**. Following the existing definition (Chakraborty et al., 2024), if the set of sentences is denoted as $\mathcal{S}$, we can define the human-generated sentence distribution as $h(s)$ for $s \in \mathcal{S}$, and similarly, the machine-generated sentence distribution as $m(s)$. This allows us to define texts containing multiple sentences under IID and non-IID settings.

- **Sentence IID Setting**. If text $S$ contains $n$ sentences $S := \{s_i\}_{i=1}^n$, and each sentence $s_i$ is i.i.d. drawn from either the human distribution $s_i \sim h(s)$ or machine distribution $s_i \sim m(s)$. Then, the human-generated text can be denoted as $S \sim h^{\otimes n}(s)$, while the machine-generated text is

$S \sim m^{\otimes n}(s)$, where $h^{\otimes n} := h \otimes h \otimes \ldots \otimes h$ ($n$ times) and $m^{\otimes n}$ denote the respective product distributions.

- **Sentence Non-IID Setting**. We follow a practical setting (Chakraborty et al., 2024; Loureiro et al., 2024) for various language tasks. Assume that $\rho$ characterizes the dependency strength between sentences $s_i$, and $\sum_i s_i$ is considered as the "average meaning" of these sentences. For instance, "medical knowledge" + "medical care" may convey an average meaning similar to "doctors". If $T_i$ is defined as the random variable concerning the $i$-th sentence, then the dependence between them is given as

$$\mathbb{E}\left[T_i \mid T_{i-1} = s_{i-1}, \cdots, T_1 = s_1\right] = \rho \frac{\sum_{k=1}^{i-1} s_k}{i-1} + (1-\rho)\mathbb{E}\left[T_i\right].$$

Here, $s_i$ operates in semantic vector space, not at the text level. We can find that when the strength $\rho = 0$, this setting degenerates into the IID setting. As $\rho$ increases, the current sentence becomes more dependent on the previous sentences.

**Problem Definition**. The task of MGT detection can be formulated as a binary classification problem. The detector $f$ maps the text $S$ to a real value $f(S) \in [0, 1]$, which indicates the confidence of machine generation. If $f(S)$ is greater than a predefined threshold $r$, text $S$ is predicted to be machine-generated; otherwise, it is human-generated. Assuming that text $S$ contains $n$ sentences, existing work (Chakraborty et al., 2024) has proven that the likelihood-ratio-based detector can achieve the upper bound of AUROC and is the best possible detector:

$$f^*(S) := \begin{cases} \text{Machine Text} & \text{if } m^{\otimes n}(S) \geq h^{\otimes n}(S), \\ \text{Human Text} & \text{if } m^{\otimes n}(S) < h^{\otimes n}(S). \end{cases}$$

## 2.2 Existence of Human-Machine Mixed Text

Most detection strategies implicitly assume that the text is entirely generated by either machines or humans, taking the entire text as input for detection. However, it is common for only parts of MGTs to be machine-generated. For example, people often use LLMs to modify text rather than relying entirely on AI to generate the entire text. Moreover, scenarios such as content expansion, dialogue continuation, and template filling all reflect the collaborative creation of text by humans and machines. Some works (Zeng et al., 2024; Zhang et al., 2024b) also focus on such mixed text.

In addition to studying explicit human-machine mixed text like existing work, this paper highlights that "mixed" text also exists in purely MGTs. Accordingly, we reveal a category of implicit human-machine mixed text: even if the text is entirely machine-generated, LLMs, with their advanced generative capabilities, can produce text consistent with human-written content, thus constituting mixed text to some extent. Examples include simple sentence structures (e.g., "Hello World"), fixed-format phrases (e.g., "Thank you for your letter"), and fixed patterns (e.g., specific places or names). This finding suggests that mixed texts are far more common than we expect.

For further verification, we calculate the Jaccard similarity based on sentence words between LLMs and humans to assess the human-likeness of the MGT. Fig. 1 presents some results, and full results are in Appendix H.3. Although most MGTs are different from human-written text, a notable portion of MGTs exhibit over 90% similarity (in fact, 100%) with human-generated texts, suggesting some sentences are challenging to differentiate. Furthermore, upon careful comparison of these 100% similar sentences, they are completely identical, which demonstrates the existence of implicit mixed text [1]. For the convenience of representation, we will refer to the implicit mixed text that contains consistent text as mixed text.

## 2.3 Detection Challenge of Mixed Text

The existence of (implicit) mixed text requires us to revisit MGT detection. This section theoretically analyzes the challenges that mixed text poses to detection under the IID setting. The theoretical results of the non-IID setting are shown in Appendix D.1, and we can obtain similar findings.

---

[1] Notably, 100% Jaccard similarity is only used to assess the human-likeness of the MGT. We neither aim to calculate an exact proportion of this mixing nor use it as an evaluation criterion. The number of consistent texts provides a lower bound on the existence, and a nonzero lower bound confirms that mixed text exists.

In the human-machine mixed text setting, we need to redefine text $S$ in Section 2.1. If the MGT $S$ contains $n$ sentences $S := \{s_i\}_{i=1}^n$, with $\alpha$ representing the proportion of human-generated sentences, then $(1-\alpha)n$ sentences $\{s_i\}_{i=1}^{(1-\alpha)n}$ are from $m(s)$ and $\alpha n$ sentences $\{s_i\}_{i=(1-\alpha)n+1}^n$ are from $h(s)$. Besides, the human-generated text $S = \{s_i\}_{i=1}^n$ consists of sentences i.i.d. drawn from human $h(s)$. Consequently, the machine-generated text is $S \sim m^{\otimes(1-\alpha)n}h^{\otimes\alpha n}(s)$, denoted as $M(S)$ for convenience, while the human-generated text is $S \sim h^{\otimes n}(s)$, denoted as $H(S)$. Then, the best possible detector under the mixed text setting is:

$$f^*(S) := \begin{cases} \text{Machine Text} & \text{if } M(S) \geq H(S), \\ \text{Human Text} & \text{if } M(S) < H(S). \end{cases}$$

Furthermore, inspired by the existing theoretical results (Chakraborty et al., 2024), we can derive the sentence complexity bound of MGT detection as follows.

**Theorem 1** (**Sentence Complexity of Mixed Text Detection under IID Setting**). *Assume the total variation distance between the human and machine distributions is $TV(m, h) = \delta > 0$. Let the text contain $n$ sentences, with $\alpha$ representing the proportion of human-generated sentences in the mixed text. To achieve an AUROC of $\epsilon$, the required sentences $n$ for the best possible detector is:*

$$n = \Omega\left(\frac{1}{\delta^2(1-\alpha)^2}\ln\left(\frac{1}{1-\epsilon}\right)\right).$$

The proof is given in Appendix E. This theorem reveals that achieving better detection performance (i.e., large $\epsilon$) requires higher sentence complexity $n$, aligning with existing findings (Kirchenbauer et al., 2024). In addition, the detection difficulty (sample complexity $n$) is directly proportional to the mixed degree $\alpha$. When $\alpha = 0$ (MGTs are completely machine-generated), the detector has the lowest complexity bound. Therefore, machine-generated text detection requires fewer human-machine mixed texts, i.e., smaller $\alpha$. The empirical evidence for this theoretical result is given in Appendix H.9.

## 3 PROPOSED METHOD

In this section, we will start with a theoretical enhancement method (Section 3.1) and then gradually explain how to model it as a stacked detection framework (Section 3.2).

### 3.1 CONCEPTUAL IMPROVEMENTS

The theoretical results from the previous section inspire us to filter out the common portions from mixed texts (i.e., reduce $\alpha$) to boost detection. However, its prerequisite is knowing which texts are mixed, contradicting the detection goal. Under the setting of Section 2.3, a compromise is to filter human-generated sequences on a ratio of $\alpha_s < \alpha$ from all text (assuming that a reasonable $\alpha_s$ is obtained and the attribution of sentences is known, to be addressed later), the following results demonstrate the detection improvement by this compromise.

**Theorem 2** (**Sentence Complexity of Filtering-based Method under IID Setting**). *Consider the MGT detection under the assumption of Theorem 1. If we filter an $\alpha_s$ ($< \alpha$) proportion of human-generated sentences from all texts, then to achieve an AUROC of $\epsilon$, the required sentences $n$ for the best possible detector is*

$$n = \Omega\left(\frac{1-\alpha_s}{\delta^2(1-\alpha)}\ln\left(\frac{1}{1-\epsilon}\right)\right).$$

Comparing Theorem 2 with Theorem 1, since $1 - \alpha_s < 1$, the filtering-based method has a lower complexity for achieving an AUROC of $\epsilon$, indicating the detection enhancement. Besides, when $\alpha_s = 0$, i.e., no filtering is performed, Theorem 2 degenerates into Theorem 1. The theoretical result under the non-IID setting is shown in Appendix D.2, where similar findings are obtained.

### 3.2 STACKED DETECTION ENHANCEMENT FRAMEWORK

Section 3.1 provides a theoretically guaranteed conceptual enhancement framework by filtering human-generated parts of texts. In this section, we propose a feasible implementation within the

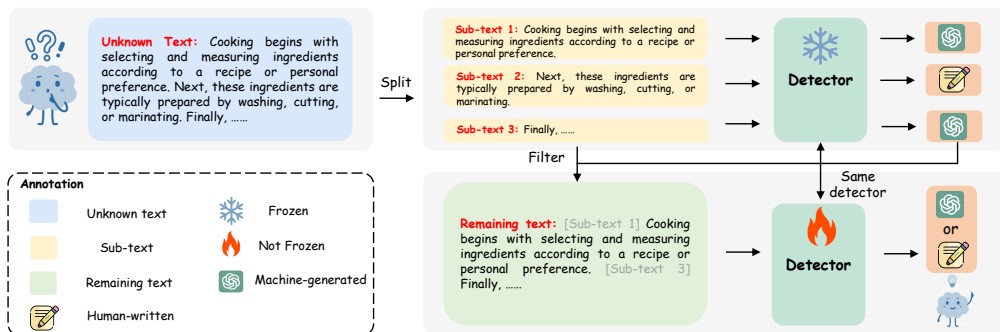

Figure 2: The inference process of the proposed enhancement framework. In the filtering step (top-right), the "Unknown Text" is split into sub-sequences. The trained detector runs on each sub-sequence, and any sub-sequence that the detector classifies as confidently "human-like" is filtered out; (2) In the detection step (bottom-right), the remaining text is concatenated and fed back into the exact same detector. The output of this second pass is the final prediction (Machine-generated or Human-written) for the whole text.

conceptual enhancement framework: the stacked detection enhancement framework, as shown in Fig. 2. Briefly, the framework first employs a detector to identify and filter human-generated texts (upper right), and then the same detector is used to make predictions on the refined text (lower right). Despite its apparent simplicity, it involves carefully designing the optimization objective, optimization strategy, and detector design. We will introduce them below.

**Optimization Objective**. Given a dataset $\mathcal{D} = \{(S_i, y_i)\}_i$, where $y_i \in \{0, 1\}$ indicates whether given text $S_i$ is machine-generated ($y_i = 1$) or human-generated ($y_i = 0$). Assuming the detection model is parameterized by $\theta$. Section 3.1 shows that a suitable optimization strategy is not to directly maximize the log-likelihood function $\log p(y_i, S_i, \theta)$ w.r.t. the complete text $S$, but $\log p(y_i, \hat{S}_i, \theta)$, where $\hat{S}_i$ is the remaining part after text $S_i$ is filtered out of some human parts.

To better characterize $\hat{S}_i$, we introduce vector $z_i \in \{0, 1\}^{n_i}$ to represent sentences' labels, where $n_i$ denotes the number of sentences in text $S_i$, and $z_{i,j}$ indicates whether the $j$-th sentence in $S_i$ is human-generated ($z_{i,j} = 0$) or machine-generated ($z_{i,j} = 1$). Therefore, $\hat{S}_i = S_i \odot z_i$, where $\odot$ is element-wise multiplication. Then the optimization goal of the detection model is to maximize the marginal likelihood of the observed data:

$$\hat{\theta} = \arg\max_\theta \sum_{(S_i, y_i) \in \mathcal{D}} \log \sum_{z_i} p(y_i, S_i, z_i; \theta). \tag{1}$$

**Optimization Strategy**. Considering Eq. (1) is often intractable due to the unobserved $z_i$, a feasible approach is to use the classical Expectation-maximization (EM) algorithm (Dempster et al., 1977) to find the maximum likelihood estimate of the marginal likelihood by iteratively applying: (1) Expectation Step (E-step). Compute expectation of the log-likelihood function of $\theta$, with respect to the current conditional distribution of $z_i$ given $(S_i, y_i)$ and estimates of the parameters $\theta^t$: $Q(\theta; \theta^t) := \sum_{(S_i, y_i)} \sum_{z_i} p(z_i | S_i, y_i; \theta^t) \log p(y_i, S_i, z_i; \theta)$. (2) Maximization Step (M-step). Maximize over $\theta$ the expectation $Q(\theta; \theta^t)$: $\theta^{t+1} := \arg\max_\theta Q(\theta; \theta^t)$.

However, directly using the EM algorithm is challenging. Specifically, if we remove an $\alpha_s$ proportion of human-generated sentences, the space size of $z_i$ is $\sum_{k=0}^{\alpha_s n_i} \binom{n_i}{k}$. Consequently, the number of forward passes during the E-step is $\mathcal{O}(2^n)$, which is computationally infeasible. Additionally, since the space of $z_i$ is large, the classic EM algorithm tends to allocate too much probability mass to the tail, wasting probability mass on unimportant hidden variables (Samdani et al., 2012).

These considerations make it natural to turn to the hard-EM algorithm (Wen et al., 2023). At this point, the optimization strategy follows the following coordinate ascent algorithm:

• **Hard E-step**: The E-step can be accomplished in a hard manner by choosing the best-fit $z_i$:

$$Q(\theta; \theta^t) := \sum_{(S_i, y_i)} \log p(y_i, S_i, z_i; \theta), \text{ where } z_i = \arg\max_{z_i} p(z_i | S_i, y_i; \theta^t).$$

- **Hard M-step**: Maximize $Q(\theta, \theta^t)$ over $\theta$:

$$\theta^{t+1} := \arg\max_\theta Q(\theta; \theta^t).$$

Compared with the classical EM algorithm, the hard E-step only selects the best-fitting mode, reducing the computational complexity of $Q(\theta; \theta^t)$. Additionally, the hard M-step focuses more on the important probability mass, which has been proven by many works (Samdani et al., 2012; Parker & Yu, 2021; Wen et al., 2023) to perform better. Nevertheless, we also provide a classical EM version, detailed and evaluated in Appendix G and H.11.

**Detector Design**. We then show how this optimization strategy can be modeled as the stacked detection framework of Fig. 2. In the hard E-step, we first need to compute the posterior distribution $p(z_i|S_i, y_i; \theta^t)$ of the latent variable $z_i$ for text $S_i$, label $y_i$ under current parameters $\theta^t$. However, $y_i$ is unknown. To this end, revisiting the posterior of the $z_i$:

$$p\left(z_i \mid S_i, y_i; \theta^t\right) = \frac{p\left(y_i \mid S_i, z_i; \theta^t\right) p\left(z_i \mid S_i; \theta^t\right)}{\sum_{z_i'} p\left(y_i \mid S_i, z_i'; \theta^t\right) p\left(z_i' \mid S_i; \theta^t\right)}.$$

We have $z_i = \arg\max_{z_i} p\left(y_i \mid S_i, z_i; \theta^t\right) p\left(z_i \mid S_i; \theta^t\right)$, which also requires $\mathcal{O}(2^n)$ computations. This motivates us to the approximation of $z_i$. We find that the effect of $z_i$ on $p\left(z_i \mid S_i; \theta^t\right)$ is often much larger than the change in $p\left(y_i \mid S_i, z_i; \theta^t\right)$, which allows us to reasonably approximate $z_i \approx \arg\max_{z_i} p\left(z_i \mid S_i; \theta^t\right)$, reducing the calculation to $\mathcal{O}(n)$ (See Appendix F.2 for the detail discussion of this approximation). Since $z_i$ denotes sentence's attribution, $p(z_i|S_i; \theta^t)$ can be obtained by the detection model $f(S, \theta)$, and for the $j$-th sentence's attribution $z_{i,j}$, if $I(\cdot)$ is an indicator function, we have

$$z_{i,j} = I(f(S_{i,j}, \theta^t) \geq 0.5), \tag{2}$$

However, this may not be appropriate in practical detection scenarios. Firstly, existing studies (Wang et al., 2023) highlight the difficulty of sentence-level detection, e.g., identifying the attribution of "Please sit down!" is difficult. Therefore, we extend sentence-level detection to sequence-level ($k$ sentences as a sequence ($k \geq 1$), and $k = 1$ represents sentence level). Here, we borrow the definition of sentences, that is, the number of sequences is $n_i$, and the $j$-th sequence of $S_i$ is $S_{i,j}$. Besides, we need to ensure the filtering ratio $\alpha_s \leq \alpha$ to avoid misfiltering. Therefore, compared with 0.5 in Eq. (2), a stricter threshold $r_e$ is needed to reduce the risk of mistakenly filtering MGT, i.e., $r_e \ll 0.5$.

Second, we set a maximum filtering ratio $\tau$ to mitigate the risk of incorrect classification due to unrestricted filtering. For example, if 9 out of 10 human-generated sentences are filtered out, classifying the entire text as machine-generated based on the remaining single sentence is questionable.

---

**Algorithm 1** Stacked Detection Framework

1: **Input:** Train data $\mathcal{D} = \{(S_i, y_i)\}_{i=1}^N$, the detection model $f(S, \theta^0)$, training epochs $T$, filtering ratio $\tau$, E-step detection threshold $r_e$, and learning rate $\eta$.
2: **procedure** INFERENCE($S_i, f(S, \theta^t)$)     ▷ Inference
3:     Split $S_i$ into a set of sequences $\{S_{i,j}\}$, where each $S_{i,j}$ contains at most $k$ sentences.
4:     Calculate $z_i$ according to Eq. (3).
5:     $\hat{S}_i = S_i \odot z_i$.
6:     **Return** $f(\hat{S}_i, \theta)$.
7: **end procedure**
8: **procedure** TRAIN($\mathcal{D}, f(S, \theta^0)$)     ▷ Training
9:     **for** $t = 0$ **to** $T - 1$ **do**
10:         **for** each batch of samples $\mathcal{D}_B \sim \mathcal{D}$ **do**
11:             Calculate $\mathcal{Q}(\theta, \theta^t)$ according to Eq. (4), where $\hat{y}_i = $ Inference $(S_i, f(S, \theta^t))$.     ▷ E-step
12:             $\theta^{t+1} = \theta + \eta \nabla_\theta \mathcal{Q}(\theta, \theta^t)$.     ▷ M-step
13:         **end for**
14:     **end for**
15:     **Return** the trained model $f(S, \theta^T)$.
16: **end procedure**

---

Based on these two constraints, the calculation of $z_i$ is as follows:

$$z_{i,j} = I\left(f(S_{i,j}, \theta^t) \geq r_e \text{ and } j \in \mathcal{T}_{\tau n_i}(\{f(S_{i,j'}, \theta^t)\}_{j'})\right), \tag{3}$$

where $\mathcal{T}_{\tau n_i}(\cdot)$ denotes the index set of top $\tau n_i$ largest values. Then, we can get the filtered text $\hat{S}_i = S_i \odot z_i$. These two constraints reflects the caution: we would rather leave some potential "noise" in the text than risk removing sequences that contain valuable signals.

Accordingly, the model prediction $\hat{y}_i = f(\hat{S}_i, \theta)$, and

$$Q(\theta, \theta^t) = \sum_{(S_i, y_i)} y_i \log(\hat{y}_i) + (1 - y_i) \log(1 - \hat{y}_i). \tag{4}$$

In the hard M-step, we need to maximize $Q(\theta, \theta^t)$ to update the model parameters, which can be achieved by gradient ascent with the learning rate $\eta$: $\theta^{t+1} = \theta^t + \eta \nabla_\theta Q(\theta, \theta^t)$.

**Overall Framework**. Alg. 1 shows the algorithm flow, and a detailed description is as follows:

- For **model inference** (Lines 2-7): (1) the "Unknown Text" is first split into a set of smaller sequences (Line 3); (2) any sub-sequence that the detector $f(S, \theta)$ classifies as confidently human is identified (Line 4); (3) we concatenate all the remaining sub-sequences into a new, shorter "Filted text"; and (4) we run the exact same detector model $f(S, \theta)$ a second time, but on this new "Filter text". The output of this second pass is the final prediction (Human-written or AI-generated) for the entire text.

- For **model training** (Line 8-16): (1) in the E-step, we calculate the best-fit latent variables $z_i$. We do this by running the entire Inference Pipeline (Lines 2-7) on the training data using the current detector, $f(S, \theta^t)$. This gives us the filtered text for each training sample. (2) In the M-step, we update the detector's parameters to $\theta^{t+1}$. We do this by optimizing the objective function in Eq. 4 only on the filtered data we obtained from the Hard E-step.

**Framework Analysis**. The effectiveness of our framework can be understood through the lens of information bottleneck theory, which states that during forward propagation, a neural network filters out information irrelevant to the prediction, gradually focusing on the most crucial parts of the input (Guan et al., 2019). According to this theory, the proposed framework performs iterative self-enhancement: (1) in the E-step, the detector identifies and filters out high confidence "human" sequences, thereby supplying more distinctive remaining texts to the same detector in the M-step; (2) in the M-step, the detector learns on the remaining texts, leading to improvements that are used in the following E-step for even more accurate filtering of "human" sequences. This interwoven process leads to self-enhancement. A more formal validity description is given in Appendix F.5. At the same time, Appendix F.5 also understands the effectiveness from the perspective of attention mechanism and text granularity. Besides, for more discussion of the proposed framework (e.g., time complexity, limitations, etc), see Appendix F.

## 4 EXPERIMENTS

### 4.1 EXPERIMENT SETUP

**Datasets and Baselines**. The experiments are conducted on the MGT detection benchmark, MGT-Bench (He et al., 2023), and we use three datasets: Essay (Verma et al., 2024) and Reuters (Verma et al., 2024) with implicit mixed text, and SQuAD1 (He et al., 2023) with explicit mixed texts. In Essay and Reuters, MGTs are generated by ChatGPT, GPT-4, ChatGPT-turbo, ChatGLM, Dolly, and Claude, while in SQuAD1, MGTs are from ChatGPT, GPT-4, ChatGPT-turbo, ChatGLM, Dolly, and StableLM. In addition, we use paraphrasing data from DetectRL (Wu et al., 2024) for robustness evaluation. Baselines include closed-source detector: GPTZero (GPTZero, 2023), feature-based methods: Log-Likelihood (Solaiman et al., 2019), Rank (Gehrmann et al., 2019), Log-Rank (Mitchell et al., 2023), DetectGPT (Mitchell et al., 2023), Fast-DetectGPT (F-DetectGPT) (Bao et al., 2024), as well as model-based detectors: OpenAI-D (Solaiman et al., 2019), ChatGPT-D (Guo et al., 2023), MPU (Tian et al., 2024), and RADAR (Hu et al., 2023). We apply the proposed stacked framework to the OpenAI-D, ChatGPT-D, MPU, and RADAR, denoted as OpenAI-STK, ChatGPT-STK, MPU-STK, and RADAR-STK, respectively. More detailed descriptions and parameter settings are given in Appendix H.

**Evaluation Metrics**. We first use the area under the receiver operating characteristic curve (AU-ROC). Besides, considering that a low false positive rate (i.e., human-generated texts being misclassified as machines) can mitigate repercussions for users (Fraser et al., 2024), we report performance as the true positive rate at a fixed false positive rate $K$ (TPR@FPR-$K$). In the experiments, $K$ is set to 0.5%. To compare with GPTZero, which outputs hard labels, we also report the performance concerning Accuracy in Appendix H.8. All experiments are repeated 5 times. The best results are bolded, and the second-best results are underlined.

Table 1: Performance concerning TPR@FPR-0.5%. Detectors are trained on ChatGPT text.

| Dataset | Method | ChatGPT | GPT-4 | GPT-turbo | ChatGLM | Dolly | Claude/StableLM | Avg. |
|---|---|---|---|---|---|---|---|---|
| Essay | Log-Likelihood | $24.08_{\pm22.67}$ | $37.70_{\pm30.63}$ | $23.12_{\pm24.38}$ | $5.86_{\pm7.02}$ | $12.45_{\pm12.04}$ | $2.48_{\pm3.06}$ | 17.62 |
| | Rank | $55.60_{\pm3.65}$ | $51.31_{\pm5.91}$ | $65.84_{\pm6.93}$ | $53.49_{\pm6.09}$ | $35.11_{\pm4.81}$ | $25.12_{\pm5.07}$ | 47.75 |
| | Log-Rank | $28.72_{\pm26.90}$ | $46.48_{\pm36.20}$ | $25.04_{\pm26.83}$ | $28.19_{\pm27.90}$ | $17.34_{\pm15.65}$ | $2.96_{\pm3.73}$ | 24.79 |
| | DetectGPT | $37.04_{\pm10.21}$ | $24.75_{\pm10.03}$ | $5.52_{\pm1.93}$ | $21.45_{\pm14.25}$ | $15.45_{\pm7.05}$ | $4.48_{\pm2.57}$ | 18.12 |
| | F-DetectGPT | $4.24_{\pm1.51}$ | $3.85_{\pm2.05}$ | $31.28_{\pm3.18}$ | $35.74_{\pm3.97}$ | $0.00_{\pm0.00}$ | $0.16_{\pm0.20}$ | 12.55 |
| | ChatGPT-D | $80.08_{\pm7.56}$ | $78.11_{\pm10.49}$ | $39.12_{\pm9.24}$ | $94.30_{\pm4.04}$ | $34.42_{\pm5.75}$ | $1.60_{\pm1.01}$ | 54.61 |
| | **ChatGPT-STK** | $86.56_{\pm9.62}$ | $83.61_{\pm11.57}$ | $46.32_{\pm9.41}$ | $96.47_{\pm3.68}$ | $44.98_{\pm10.42}$ | $4.24_{\pm2.98}$ | 60.36 |
| | OpenAI-D | $78.32_{\pm39.18}$ | $87.70_{\pm20.72}$ | $70.24_{\pm35.62}$ | $96.55_{\pm5.32}$ | $66.70_{\pm32.73}$ | $22.56_{\pm17.88}$ | 70.34 |
| | **OpenAI-STK** | $96.96_{\pm1.83}$ | $\underline{96.89}_{\pm1.73}$ | $88.48_{\pm2.79}$ | $98.63_{\pm0.94}$ | $\mathbf{81.55}_{\pm7.87}$ | $28.24_{\pm7.54}$ | 81.79 |
| | MPU | $\mathbf{99.92}_{\pm0.16}$ | $\mathbf{99.26}_{\pm0.40}$ | $67.92_{\pm12.35}$ | $\mathbf{99.60}_{\pm0.51}$ | $76.14_{\pm6.98}$ | $59.92_{\pm18.84}$ | 83.79 |
| | **MPU-STK** | $\mathbf{99.92}_{\pm0.16}$ | $\mathbf{99.26}_{\pm0.40}$ | $65.44_{\pm16.47}$ | $\mathbf{99.60}_{\pm0.51}$ | $\underline{79.06}_{\pm6.65}$ | $\mathbf{78.64}_{\pm19.68}$ | 86.99 |
| | RADAR | $96.88_{\pm2.25}$ | $96.56_{\pm0.56}$ | $\underline{92.64}_{\pm4.47}$ | $98.96_{\pm0.90}$ | $65.75_{\pm8.58}$ | $58.64_{\pm8.36}$ | 84.90 |
| | **RADAR-STK** | $\underline{98.16}_{\pm1.38}$ | $95.82_{\pm1.78}$ | $\mathbf{94.64}_{\pm3.89}$ | $\underline{99.12}_{\pm0.59}$ | $70.39_{\pm8.22}$ | $\underline{64.96}_{\pm7.80}$ | **87.18** |
| Reuters | Log-Likelihood | $77.84_{\pm5.19}$ | $14.88_{\pm5.98}$ | $86.08_{\pm3.38}$ | $93.76_{\pm2.03}$ | $11.20_{\pm4.45}$ | $15.04_{\pm6.86}$ | 49.80 |
| | Rank | $48.88_{\pm1.59}$ | $35.92_{\pm2.88}$ | $58.40_{\pm3.94}$ | $40.56_{\pm1.85}$ | $18.56_{\pm2.27}$ | $6.24_{\pm1.87}$ | 34.76 |
| | Log-Rank | $82.40_{\pm5.24}$ | $25.92_{\pm7.08}$ | $90.96_{\pm4.12}$ | $96.80_{\pm0.88}$ | $14.00_{\pm4.82}$ | $17.60_{\pm8.29}$ | 54.61 |
| | DetectGPT | $4.40_{\pm2.62}$ | $0.64_{\pm0.54}$ | $2.32_{\pm1.87}$ | $2.56_{\pm2.80}$ | $0.48_{\pm0.47}$ | $3.04_{\pm1.61}$ | 2.24 |
| | F-DetectGPT | $48.00_{\pm9.48}$ | $6.80_{\pm1.88}$ | $92.96_{\pm1.65}$ | $88.96_{\pm4.80}$ | $0.00_{\pm0.00}$ | $0.48_{\pm0.39}$ | 39.53 |
| | ChatGPT-D | $98.00_{\pm2.25}$ | $94.32_{\pm3.97}$ | $96.08_{\pm2.23}$ | $98.48_{\pm0.78}$ | $59.76_{\pm13.36}$ | $11.84_{\pm6.11}$ | 76.41 |
| | **ChatGPT-STK** | $99.28_{\pm0.39}$ | $96.16_{\pm1.15}$ | $98.08_{\pm1.17}$ | $98.72_{\pm0.47}$ | $64.56_{\pm8.32}$ | $30.32_{\pm8.23}$ | 81.19 |
| | OpenAI-D | $96.88_{\pm4.26}$ | $84.08_{\pm9.42}$ | $96.56_{\pm5.32}$ | $98.00_{\pm1.13}$ | $49.44_{\pm5.83}$ | $19.92_{\pm5.21}$ | 74.15 |
| | **OpenAI-STK** | $99.52_{\pm0.30}$ | $95.36_{\pm2.19}$ | $99.76_{\pm0.20}$ | $98.48_{\pm0.53}$ | $62.72_{\pm5.44}$ | $39.44_{\pm7.20}$ | 82.55 |
| | MPU | $\mathbf{100.00}_{\pm0.00}$ | $97.92_{\pm1.06}$ | $\underline{99.92}_{\pm0.16}$ | $\mathbf{99.60}_{\pm0.25}$ | $72.64_{\pm7.02}$ | $75.68_{\pm12.92}$ | 90.96 |
| | **MPU-STK** | $\mathbf{100.00}_{\pm0.00}$ | $98.08_{\pm1.14}$ | $\mathbf{100.00}_{\pm0.00}$ | $99.44_{\pm0.41}$ | $72.80_{\pm8.95}$ | $\underline{84.40}_{\pm10.39}$ | 92.45 |
| | RADAR | $\mathbf{100.00}_{\pm0.00}$ | $\mathbf{99.92}_{\pm0.16}$ | $99.68_{\pm0.30}$ | $\mathbf{99.92}_{\pm0.16}$ | $\mathbf{89.68}_{\pm2.08}$ | $\mathbf{95.68}_{\pm1.53}$ | **97.48** |
| | **RADAR-STK** | $\mathbf{100.00}_{\pm0.00}$ | $\mathbf{99.92}_{\pm0.16}$ | $99.68_{\pm0.30}$ | $\mathbf{99.92}_{\pm0.16}$ | $\underline{89.68}_{\pm2.25}$ | $\mathbf{95.68}_{\pm1.53}$ | **97.48** |

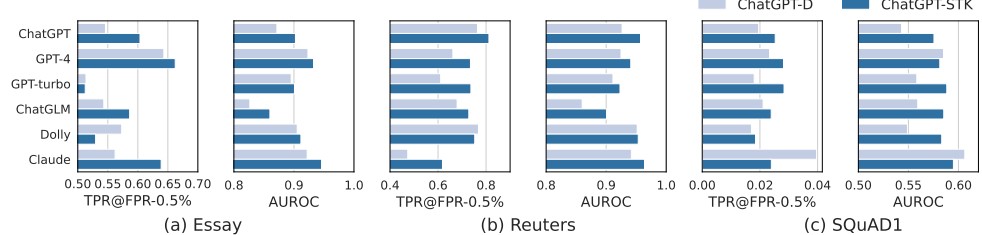

Figure 3: Average detection performance (x-axis) of detectors (ChatGPT-D and our boosting strategy ChatGPT-STK) tested across various LLMs, where these detectors are trained on texts generated by specific LLM (y-axis).

## 4.2 PERFORMANCE COMPARISON

**Cross-LLM Performance**. The detector usually has no prior knowledge of LLM. Therefore, detectors are trained on texts generated by a specific LLM and tested on texts across various LLMs. Table 1 presents the comparison of TPR@FPR-0.5% when detectors are trained on ChatGPT texts and tested on various LLM texts. See Table 2 and Table 3 of Appendix H.4 for more performance comparison results. Firstly, it can be observed that the proposed strategy significantly improves the detection performance of the original detectors. For example, on the Essay dataset, ChatGPT-STK increases the TPR@FPR-0.5% of ChatGPT-D from an average of 54.61% to 60.36%. In addition to the improvement in average performance, for specific LLM text detection (each cell), the proposed strategy shows enhancement potential in most settings. This property underscores the practicality of our method, given that real-world scenarios often lack prior knowledge of the specific generative models involved. Besides, the cross-LLM performance is not necessarily inferior to the intra-LLM performance in MGT detection. This may depend on the quality of the generated text. For example, ChatGLM (cross-LLM) outperforms ChatGPT (intra-LLM). Similar findings are observed in related work (He et al., 2023). Finally, aligning with existing findings (Wu et al., 2024), feature-based methods are less effective than model-based methods, echoing the conclusion that manually designed features struggle to cover the extensive and complex patterns in texts, thus highlighting the advantages of model-based approaches.

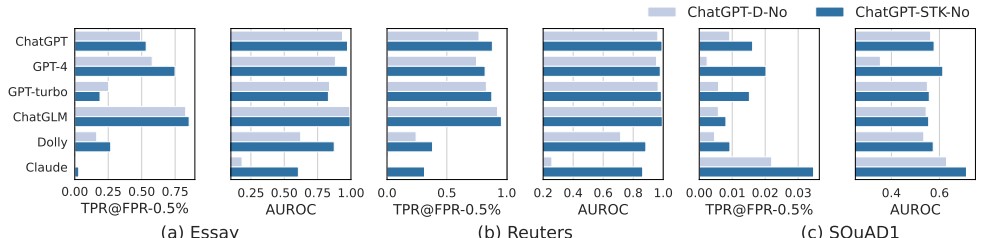

Figure 4: Performance (x-axis) of the un-fine-tuned detectors tested on various LLM texts (y-axis).

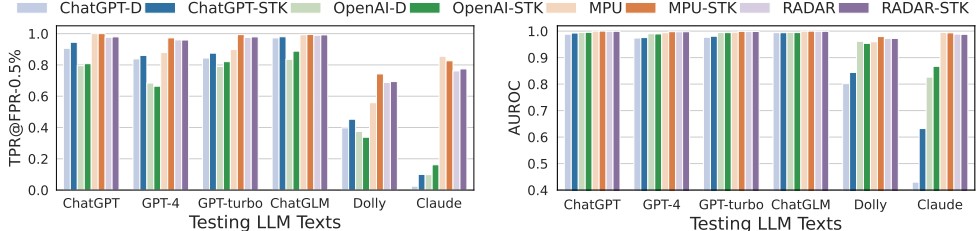

Figure 5: Performance under cross-domain setting. The Essay dataset served as the source domain, and the Reuters dataset as the target domain. The detector is trained on ChatGPT texts.

Beyond training on ChatGPT-generated texts, we assess detectors trained extensively on various other LLM texts. Fig. 3 presents the average performance (x-axis) of detectors (ChatGPT-D and ChatGPT-STK) tested across various LLMs, with training on different LLM texts (y-axis). Results for OpenAI-STK, MPU-STK, and RADAR-STK are shown in Fig. 11-13, and more detailed performances appear in Tables 4-13 in Appendix H.4. The proposed strategy markedly improves cross-LLM detection capability for these detectors.

**Enhancement in a Training-free Manner**. Previously, we demonstrated the enhancement effect of using the proposed stacked framework (architecture + optimization strategy) on existing detectors. A more stringent setting involves evaluating the enhancement effect on already trained detectors using only the stacked structure without additional training. Fig. 4 shows the enhancement effect on ChatGPT-D when only using the proposed stacked structure without fine-tuning (denoted as ChatGPT-D-No and ChatGPT-STK-No). Encouragingly, even without retraining, the proposed stacked framework exhibits significant enhancement. This plug-and-play property provides high flexibility and scalability, making it a practical solution.

**Cross-domain Performance**. In addition to cross-LLM performance, we also evaluated the cross-domain performance, with the results shown in Fig. 5. In this evaluation, the Essay dataset served as the source domain and the Reuters dataset as the target domain. The results indicate that models employing the proposed enhancement strategy demonstrated superior detection performance in most setups, strongly supporting the effectiveness of the proposed strategy in improving the detector's cross-domain generalization capability.

**Enhancement of Auto-regressive-based Detector**. Admittedly, the proposed strategy may affect the performance of auto-regressive-based detectors since filtering will destroy the context at the filtering point. To this end, we use GPT-2 as the detection model (using the last token embedding with a fully connected layer for binary classification) to explore the impact on auto-regressive models, as shown in Fig. 6. We can find that the proposed framework still demonstrates enhanced effectiveness. The possible reason is that the persistent influence of important tokens in the attention mechanism can alleviate the destructive effect of text structure. For more theoretical discussion, see the second point of Appendix F.5. This property highlights the broad applicability of the proposed framework.

**Robust Performance against Paraphrasing Attack**. Existing research (Sadasivan et al., 2023) generally indicates that MGT detection is susceptible to paraphrase attacks, where attackers attempt to bypass the detector by rewriting text without altering its semantics. To address this, we evaluated the robustness enhancement brought by the proposed framework using three types of paraphrase attack data provided by the DetectRL dataset, with results shown in Fig. 7. For more robustness evaluation results, please refer to Appendix H.6. It is evident that even in adversarial environments,

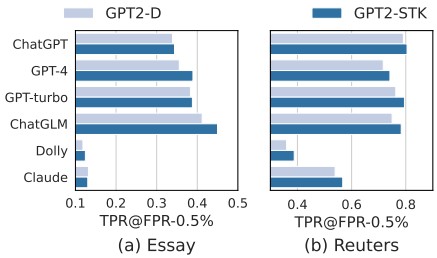 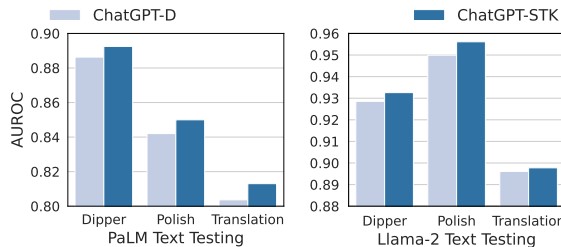

Figure 6: Enhancements to GPT2-based Detector, which is trained on ChatGPT texts.

Figure 7: Enhance the robustness of ChatGPT-D. Here we use three attacks: Dipper, Polish, and Translation.

the proposed strategy surprisingly enhances the detector's robustness against these paraphrase attacks, underscoring the broad applicability of the strategy.

## 5 CONCLUSION

This paper emphasises the importance of human-machine mixed text in MGT detection. Firstly, statistical analysis of existing datasets has empirically revealed the widespread presence of mixed texts, even for pure MGTs. Then, we have theoretically demonstrated their negative impact on detection. Based on our theoretical findings, we have designed a stacked detection enhancement framework. Through theoretical analysis and extensive experimental evaluation, we have demonstrated the detection enhancement capabilities of the proposed framework. Moreover, the stacked structure can be seamlessly integrated into existing trained detectors in a training-free manner, thus achieving flexible enhancement. Notably, our primary technique contribution lies in the conceptual framework (i.e., filtering a portion of human-generated text from all texts), and the proposed EM-inspired approach is merely one feasible implementation within the conceptual framework. This conceptual insight presents a promising direction for future work to further enhance detection.

ETHICS STATEMENT

This paper presents work whose goal is to advance MGT detection, and it may potentially have positive impacts in fields such as news authenticity verification and academic integrity maintenance. Therefore, it does not involve human subjects, practices to data set releases, potentially harmful insights, potential conflicts of interest and sponsorship, discrimination/bias/fairness concerns, privacy and security issues, legal compliance, and research integrity issues.

REPRODUCIBILITY STATEMENT

Our code is available at `https://anonymous.4open.science/r/MGTD`. All datasets used in this study (Essay, Reuters, and SQuAD1) are publicly available. Detailed implementation details (e.g., learning rate, training epochs, optimizer, and hyperparameters $k$, $\tau$, and $r_e$ of the proposed framework) are provided in the Appendix. We report the average results over five random seeds, including standard deviations.

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

APPENDIX

# A   THE USE OF LARGE LANGUAGE MODELS

We used LLMs solely to help correct grammatical and spelling errors and improve the clarity of the paper. All technical descriptions, experimental designs, analyses, and conclusions were conceived, performed, and verified entirely by the authors, without LLMs' involvement. The authors have thoroughly reviewed and validated all language model-assisted edits to ensure the accuracy and integrity of the paper.

# B   DETAILED DISCUSSION OF OUR CONTRIBUTIONS

Our contribution lies in revealing the existence of implicit human-machine mixed texts and their theoretical harm to detection, as well as how theoretical results could really help develop better detection methods even in the presence of mixed texts.

- **We reveal the existence of implicit human-machine mixed texts (Section 2.2)**. Existing works (Zhang et al., 2024b; Wang et al., 2024; Dugan et al., 2020) emphasize purely mixed text scenarios, where the text is a mix of human-written and machine-generated content. For example, RoFT Dugan et al. (2020) studied mixed texts where machines continue writing human text. However, we extend this to more general mixed scenarios, even if the text is entirely machine-generated, LLMs, with their advanced generative capabilities, can produce text indistinguishable from human-written content, thus qualifying as mixed text to some extent. Considering the difficulty of detecting mixed texts (Zhang et al., 2024b), this study highlights that the actual detection challenge may be far more severe than recognized, advocating for attention to (implicit) mixed text detection, even when the text is entirely machine-generated.

- **We theoretically demonstrate the impact of mixed texts on detection (Section 2.3)**. Intuitively, the human-generated text mixed into MGT acts as noise, and ignoring its treatment may affect detection. Existing work (Zhang et al., 2024b; Wang et al., 2024) has empirically shown that detecting such mixed texts is challenging, lacking theoretical guarantees in machine learning. Therefore, we take the first step to provide a theoretical guarantee of the impact of mixed texts on detection. This theoretical foundation is not only more general but also systematically reveals the mechanism by which mixed texts affect the performance of detection models, guiding the development of more effective detection strategies (i.e., our stacked detection framework).

- **We develop a theoretically inspired detection enhancement framework (Section 3)**, which involves ensuring theoretical validity (Section 3.1), revisiting the detection optimization objective, and the balance between the feasibility and efficiency of the optimization process (Section 3.2). Notably, our primary technique contribution and novelty lies in the conceptual framework rather than just specific implementation details (i.e., EM algorithm). Although our current implementation is based on the hard-EM algorithm, this is merely one feasible implementation within the framework; alternatives include using EM algorithm variants Neal & Hinton (1998); Cheung (2005) and variational inference Jordan et al. (1999). This conceptual insight and framework design can inspire future research in this field.

- **We demonstrate our effectiveness from both theoretical and experimental aspects (Section 3.1, 4, and H)**. Theorem 2 and 4 provide theoretical bounds for the proposed framework, and they have a smaller complexity when compared with Theorem 1 and 3. Extensive experiments on cross-LLM, cross-domain, and resistance to paraphrasing attacks demonstrate the enhancement of the proposed framework. Remarkably, the proposed strategy can also enhance detection in a training-free manner (i.e., simply configuring the proposed stacked structure without optimization). This plug-and-play property makes it more flexible and scalable.

## C  RELATED WORKS

Existing detection efforts can be categorized into watermark-based, language feature-based, and deep learning-based methods (Fraser et al., 2024).

The watermarks hidden in the text indicate that the text is AI-generated. Existing work (Kirchenbauer et al., 2023) suggested mildly promoting the use of predefined "green" tokens during text generation and proposed a statistical test method to detect watermarks. Unigram-Watermark (Zhao et al., 2024) was proposed by extending existing approaches with a simplified fixed grouping strategy. Obviously, watermark-based methods require high privileges over the model, which limits their wide applicability. Furthermore, recent work (Zhang et al., 2024a; Jovanović et al., 2024) has shown that strong watermarking is difficult to achieve and is vulnerable to watermark attacks.

Language feature-based methods exploit the unique properties of text to distinguish between natural and generated text. DetectGPT (Mitchell et al., 2023) based on the observation that minor perturbations in generated text result in a lower log probability for the rewritten text than the original sample. Similar work (Solaiman et al., 2019) achieved good performance by detecting based on the higher log probabilities of generated text compared to natural text. Through analyzing the characteristics of generated text, existing work (Guo et al., 2023) found that responses from ChatGPT tend to be more objective, formal, and less emotional. In addition, the intrinsic dimension of the text is a good metric (Tulchinskii et al., 2024), where the average intrinsic dimension of the generated text is about 1.5 lower than that of natural text. These methods, based on specific data (generative models), extract different features and struggle to comprehensively consider the features of generated text.

Deep learning-based methods do not involve explicit feature extraction but instead use the entire text as input, allowing the detector to learn implicit features during training. The energy-based model was utilized to differentiate between real and generated text (Tulchinskii et al., 2024). Recent work found that smaller and partially trained language models are better general detectors of machine-generated text (Mireshghallah et al., 2023), and the detector should maximize performance for more advanced generation models (Pagnoni et al., 2022). Considering the difficulty existing detection models have in capturing the factual structure of documents, a graph-based model (Zhong et al., 2020) was proposed to represent the factual structure of a given document. SeqXGPT (Wang et al., 2023) used a probability list from open-source models as input for the detector model, rather than the raw text itself. These methods focus on supervised training of specific data, and extensive research has found that under these settings, the models' generalization ability significantly decreases (Liang et al., 2023; Ippolito et al., 2020). For example, Detectors frequently misclassify texts written by non-native English speakers (Liang et al., 2023). Besides, certain categories, such as recipes, are easier to detect than others, such as stories or news (Tulchinskii et al., 2024). Different sampling strategies also significantly affect detection performance, with texts generated using nucleus sampling being the most challenging to detect (Ippolito et al., 2020).

## D  MORE THEORETICAL RESULTS

### D.1  SENTENCE COMPLEXITY OF MIXED TEXT DETECTION UNDER NON-IID SETTING

Following the non-IID setting from Section 2.1, assume that text $S$ contains $L$ independent sequences $\{v_i\}_{i=1}^{L}$, where each sequence $v_i$ consists of $c_i$ dependent sentences. This assumption is reasonable due to factors such as topic independence and context switching. Then, we can derive the following result.

**Theorem 3 (Sentence Complexity of Mixed Text Detection under Non-IID Setting).** *Assume the total variation distance between the human and machine distributions is $TV(m, h) = \delta > 0$. Let the text contain $n$ sentences, with $\alpha$ representing the proportion of human-generated sentences in*

*the mixed text. To achieve an AUROC of $\epsilon$, the required sentences $n$ for the best possible detector is:*

$$
n = \Omega \left( \frac{1}{(1-\alpha)^2 \delta^2} \ln \left( \frac{1}{1-\epsilon} \right) + \frac{1}{(1-\alpha)\delta} \sum_{j=1}^{L} (c_j - 1)\rho_j \right.
$$

$$
\left. + \left( \frac{1}{(1-\alpha)^3 \delta^3} \cdot \left( \sum_{j=1}^{L} (c_j - 1) \rho_j \right) \cdot \ln \left( \frac{1}{1-\epsilon} \right) \right)^{1/2} \right). \tag{5}
$$

We can get similar findings as Theorem 1. Besides, when the dependence coefficient $\rho_j = 0$ for all $j$, then Theorem 3 under the non-IID setting degenerates into Theorem 1 under the IID setting. In summary, Theorem 1 and Theorem 3 inspire us that machine-generated text detection requires fewer human-machine mixed texts.

### D.2 SENTENCE COMPLEXITY OF FILTERING-BASED METHOD UNDER NON-IID SETTING

**Theorem 4** (**Sentence Complexity of Filtering-based Method under Non-IID Setting**). *Consider the MGT detection under the assumption of Theorem 3. If we filter an $\alpha_s$ $(< \alpha)$ proportion of human-generated sentences from all texts, then to achieve an AUROC of $\epsilon$, the required sentences $n$ for the best possible detector is*

$$
n = \Omega \left( \frac{1-\alpha_s}{(1-\alpha)^2 \delta^2} \ln \left( \frac{1}{1-\epsilon} \right) + \frac{1}{(1-\alpha)\delta} \sum_{j=1}^{L} (c_j - 1)\rho_j \right.
$$

$$
\left. + \left( \frac{1-\alpha_s}{(1-\alpha)^3 \delta^3} \cdot \left( \sum_{j=1}^{L} (c_j - 1) \rho_j \right) \cdot \ln \left( \frac{1}{1-\epsilon} \right) \right)^{1/2} \right).
$$

## E PROOFS

### E.1 PROOF OF THEOREM 1

*Proof.* Our proof follows the proof of Theorem 1 in existing work (Chakraborty et al., 2024). First, according to our assumption, the Total Variance Distance $TV(m, h)$ between machine-generated sentence distribution and human-generated sentence distribution is $\delta$ where $\delta > 0$. According to the definition of TV distance, there exists some set $A \in \mathcal{S}$ such that given the sentences $s^m \sim m(s)$ and $s^h \sim h(s)$, it holds

$$
p(s^m \in A) - p(s^h \in A) = \delta
$$

If we set $p\left(s^h \in A\right) = q$, it implies that $p\left(s^m \in A\right) = q+\delta$. For convenience, we define $\beta = 1-\alpha$, which denotes the proportion of machine-generated sentences in the human-machine mixed text. Then, in the human-machine mixed text of $n$ sentences $\{s_i\}_{i=1}^{n}$, $\beta n$ sentences $\{s_i\}_{i=1}^{\beta n}$ are from $m(s)$ and $\alpha n$ sentences $\{s_i\}_{i=\beta n+1}^{n}$ are from $h(s)$. Given $p\left(s^h \in A\right) = q$ and $p\left(s^m \in A\right) = q+\delta$, we can deduce that, on average, $(q + \beta\delta)n$ number of sentences will be in $A$. Similarly, for purely human-generated text of $n$ sentences $\{s_i\}_{i=1}^{n}$ from $h(s)$, on average, $qn$ number of sentences will be in $A$. Therefore, applying the Chernoff bound, we have

$$
\mathbb{P} \left( \text{ at least } \left( q + \frac{\beta\delta}{2} \right) n \text{ sentences of human-machine mixed text are in } A \right) \le \exp^{-\frac{\beta^2 \delta^2 n}{2}} \tag{6}
$$

and

$$
\mathbb{P} \left( \text{ at most } \left( q + \frac{\beta\delta}{2} \right) n \text{ sentences of human-generated text are in } A \right) \le \exp^{-\frac{\beta^2 \delta^2 n}{2}} \tag{7}
$$

Let $A'$ denote the set of $n$-tuples containing more than $\left(q + \frac{\beta\delta}{2}\right)n$ sentences of $A$. If the mixed text distribution is defined as $m^{\otimes\beta n}h^{\otimes\alpha n}$ and the human-generated text distribution is $h^{\otimes n}$, we can bound

$$
\begin{aligned}
\mathrm{TV}\left(m^{\otimes\beta n}h^{\otimes\alpha n}, h^{\otimes n}\right) &\geq p\left(\{s_i^m\}_{i=1}^{\beta n} \cup \{s_i^h\}_{i=1}^{\alpha n} \in A'\right) - p\left(\{s_i^h\}_{i=1}^{n} \in A'\right) \\
&\geq \left(1 - \exp^{-\frac{\beta^2\delta^2 n}{2}}\right) - \exp^{-\frac{\beta^2\delta^2 n}{2}} \\
&= 1 - 2\exp^{-\frac{\beta^2\delta^2 n}{2}}.
\end{aligned}
\tag{8}
$$

Invoking existing theoretical result (Proposition 1 in Chakraborty et al., 2024), we have

$$
AUROC \leq \frac{1}{2} + TV\left(m^{\otimes\beta n}h^{\otimes\alpha n}, h^{\otimes n}\right) - \frac{TV\left(m^{\otimes\beta n}h^{\otimes\alpha n}, h^{\otimes n}\right)^2}{2}
\tag{9}
$$

If we need the AUROC of the best possible detector to satisfy $AUROC \geq \epsilon$, then we need

$$
\frac{1}{2} + TV\left(m^{\otimes\beta n}h^{\otimes\alpha n}, h^{\otimes n}\right) - \frac{TV\left(m^{\otimes\beta n}h^{\otimes\alpha n}, h^{\otimes n}\right)^2}{2} \geq \epsilon
\tag{10}
$$

Since the left-hand side is the monotonically increasing function of $TV\left(m^{\otimes\beta n}h^{\otimes\alpha n}, h^{\otimes n}\right)$, combined with Eq. (8), it holds from the minimum value that

$$
\frac{1}{2} + \left(1 - 2\exp^{-\frac{\beta^2\delta^2 n}{2}}\right) - \frac{\left(1 - 2\exp^{-\frac{\beta^2\delta^2 n}{2}}\right)^2}{2} \geq \epsilon
\tag{11}
$$

After expanding the squares and rearranging the terms, we get

$$
\frac{1-\epsilon}{2} \geq \exp^{-n\beta^2\delta^2}
\tag{12}
$$

Taking the logarithm of both sides of the inequality and sorting them, we can get

$$
n \geq \frac{1}{\beta^2\delta^2}\ln\left(\frac{2}{1-\epsilon}\right) = \frac{1}{(1-\alpha)^2\delta^2}\ln\left(\frac{2}{1-\epsilon}\right)
\tag{13}
$$

The theorem is proved.

$\square$

### E.2 PROOF OF THEOREM 3

*Proof.* This proof also follows the existing work (Chakraborty et al., 2024). First, we need to introduce a lemma from existing work (Dhurandhar, 2013) to support our proof.

**Lemma 5.** *Let $n$ be the number of samples drawn sequentially from $\mathbb{P}\left(S_1, S_2 \cdots S_n\right) = \prod_{j=1}^{L}\tau_j$, where $\tau_j$ are independent subsets consisting of $c_j$ dependent sequences $\left(s_1, s_2 \cdots s_{c_j}\right)$ such that $\sum_{j=1}^{L} c_j = n$. Under dependence structure in (16), for any $\delta > \frac{\sum_{l=1}^{L}(c_j - 1)\rho_j}{n}$, it holds that*

$$
\mathbb{P}(\bar{S} - \mathbb{E}[\bar{S}] \geq \delta) \leq \exp\frac{-2\left(n\delta - \sum_{j=1}^{L}\left(c_j - 1\right)\rho_j\right)^2}{n}
$$

$$
\mathbb{P}(\mathbb{E}[\bar{S}] - \bar{S} \geq \delta) \leq \exp\frac{-2\left(n\delta - \sum_{j=1}^{L}\left(c_j - 1\right)\rho_j\right)^2}{n}
$$

*where $\bar{S} = \frac{1}{n}\sum_{n=1}^{n} s_i$ and $\mathbb{E}\left[S_i \mid S_{i-1} = s_{i-1}, \cdots, S_1 = s_1\right] = \frac{\rho}{i-1}\sum_{k=1}^{i-1} s_k + (1-\rho)\mathbb{E}\left[S_i\right]$.*

This lemma provides a theoretical upper bound in the non-iid setting. When the dependence strength $\rho = 0$, it degenerates to the Chernoff bound.

Based on this lemma, we can follow a similar proof in the iid setting. If we set $p\left(s^h \in A\right) = q$, it implies that $p\left(s^m \in A\right) = q + \delta$. For convenience, we define $\beta = 1 - \alpha$, which denotes the proportion of machine-generated sentences in the human-machine mixed text. Then, in the human-machine mixed text of $n$ sentences $\{s_i\}_{i=1}^n$, $\beta n$ sentences $\{s_i\}_{i=1}^{\beta n}$ are from $m(s)$ and $\alpha n$ sentences $\{s_i\}_{i=\beta n+1}^n$ are from $h(s)$. Given $p\left(s^h \in A\right) = q$ and $p\left(s^m \in A\right) = q + \delta$, we can deduce that, on average, $(q + \beta\delta)n$ number of sentences will be in $A$. Similarly, for purely human-generated text of $n$ sentences $\{s_i\}_{i=1}^n$ from $h(s)$, on average, $qn$ number of sentences will be in $A$. Therefore, applying the above lemma, we have

$$\mathbb{P}\left( \text{ at least } \left(q + \frac{\beta\delta}{2}\right) n \text{ sentences of mixed text are in } A \right) \leq \exp^{\frac{-2\left(\beta n \frac{\delta}{2} - \sum_{j=1}^L (c_j - 1)\rho_j\right)^2}{n}} \quad (14)$$

and

$$\mathbb{P}\left( \text{ at most } \left(q + \frac{\beta\delta}{2}\right) n \text{ sentences of human-written text are in } A \right) \leq \exp^{\frac{-2\left(\beta n \frac{\delta}{2} - \sum_{j=1}^L (c_j - 1)\rho_j\right)^2}{n}} \tag{15}$$

Let $A'$ denote the set of $n$-tuples containing more than $\left(q + \frac{\beta\delta}{2}\right) n$ sentences of $A$. If the mixed text distribution is defined as $m^{\otimes \beta n} h^{\otimes \alpha n}$ and the human-generated text distribution is $h^{\otimes n}$, we have

$$\begin{aligned}
\text{TV}\left(m^{\otimes \beta n} h^{\otimes \alpha n}, h^{\otimes n}\right) &\geq p\left(\{s_i^m\}_{i=1}^{\beta n} \cup \{s_i^h\}_{i=1}^{\alpha n} \in A'\right) - p\left(\{s_i^h\}_{i=1}^n \in A'\right) \\
&\geq \left(1 - \exp^{\frac{-2\left(\beta n \frac{\delta}{2} - \sum_{j=1}^L (c_j - 1)\rho_j\right)^2}{n}}\right) - \exp^{\frac{-2\left(\beta n \frac{\delta}{2} - \sum_{j=1}^L (c_j - 1)\rho_j\right)^2}{n}} \quad (16) \\
&= 1 - 2\exp^{\frac{-2\left(\beta n \frac{\delta}{2} - \sum_{j=1}^L (c_j - 1)\rho_j\right)^2}{n}}.
\end{aligned}$$

If we need the AUROC of the best possible detector to satisfy $AUROC \geq \epsilon$, then we need

$$\frac{1}{2} + TV\left(m^{\otimes \beta n} h^{\otimes \alpha n}, h^{\otimes n}\right) - \frac{TV\left(m^{\otimes \beta n} h^{\otimes \alpha n}, h^{\otimes n}\right)^2}{2} \geq \epsilon \tag{17}$$

Since the left-hand side is the monotonically increasing function of $TV\left(m^{\otimes \beta n} h^{\otimes \alpha n}, h^{\otimes n}\right)$, combined with Eq. (16), it holds from the minimum value that

$$\frac{1}{2} + \left(1 - 2\exp^{\frac{-2\left(\beta n \frac{\delta}{2} - \sum_{j=1}^L (c_j - 1)\rho_j\right)^2}{n}}\right) - \frac{\left(1 - 2\exp^{\frac{-2\left(\beta n \frac{\delta}{2} - \sum_{j=1}^L (c_j - 1)\rho_j\right)^2}{n}}\right)^2}{2} \geq \epsilon \quad (18)$$

After expanding the squares and rearranging the terms, we get

$$\frac{1 - \epsilon}{2} \geq \exp^{\frac{-4\left(\beta n \frac{\delta}{2} - \sum_{j=1}^L (c_j - 1)\rho_j\right)^2}{n}} \tag{19}$$

Let's denote $\alpha = \sum_{j=1}^L (c_j - 1)\rho_j$ and $\gamma(\epsilon) = \ln\left(\frac{2}{1-\epsilon}\right)$. Further rearranging, we get

$$\ln \gamma(\epsilon) \leq \frac{4}{n}\left(\beta n \frac{\delta}{2} - \alpha\right)^2 = n\beta^2 \delta^2 - 4\alpha\beta\delta + \frac{4}{n}\alpha^2 \tag{20}$$

Further rearranging the terms, we can get

$$\beta^2 \delta^2 n^2 - n\left(4\beta\alpha\delta + \gamma(\epsilon)\right) + 4\alpha^2 \geq 0 \tag{21}$$

The solution of the quadratic equation is given by

$$
\begin{aligned}
n &\geq \frac{\gamma(\epsilon)}{2\beta^2\delta^2} + 2\frac{\alpha}{\beta\delta} + \frac{1}{2\beta^2\delta^2}\sqrt{(4\alpha\beta\delta + \gamma(\epsilon))^2 - 16\alpha^2\beta^2\delta^2} \\
&= \frac{\gamma(\epsilon)}{2\beta^2\delta^2} + 2\frac{\alpha}{\beta\delta} + \frac{1}{2\beta^2\delta^2}\sqrt{\gamma(\epsilon)^2 + 8\alpha\beta\delta\gamma(\epsilon)} \\
&= \frac{\gamma(\epsilon)}{2\beta^2\delta^2} + 2\frac{\alpha}{\beta\delta} + \frac{1}{2\beta^2\delta^2}\sqrt{\frac{2\gamma(\epsilon)^2 + 16\alpha\beta\delta\gamma(\epsilon)}{2}} \\
&\geq \frac{\gamma(\epsilon)}{2\beta^2\delta^2} + 2\frac{\alpha}{\beta\delta} + \frac{1}{2\beta^2\delta^2}\left(\frac{\sqrt{2}\gamma(\epsilon) + 4\sqrt{\alpha\beta\delta\gamma(\epsilon)}}{2}\right) \\
&= \frac{(2+\sqrt{2})\gamma(\epsilon)}{4\beta^2\delta^2} + 2\frac{\alpha}{\beta\delta} + \sqrt{\frac{\alpha\gamma(\epsilon)}{\beta^3\delta^3}}
\end{aligned}
\tag{22}
$$

If using the order notation, we will obtain

$$
\begin{aligned}
n = \Omega\Bigg( &\frac{1}{(1-\alpha)^2\delta^2}\ln\left(\frac{1}{1-\epsilon}\right) + \frac{1}{(1-\alpha)\delta}\sum_{j=1}^{L}(c_j-1)\rho_j \\
&+ \left(\frac{1}{(1-\alpha)^3\delta^3}\cdot\left(\sum_{j=1}^{L}(c_j-1)\rho_j\right)\cdot\ln\left(\frac{1}{1-\epsilon}\right)\right)^{1/2}\Bigg).
\end{aligned}
\tag{23}
$$

Then the theorem is proved.

□

### E.3   Proof of Theorem 2

*Proof.* Similar to the proof for Theorem 1, if we define $p\left(s^h \in A\right) = q$, it implies that $p\left(s^m \in A\right) = q + \delta$. Then, in the human-machine mixed text of $(1-\alpha_s)n$ sentences $\{s_i\}_{i=1}^{(1-\alpha_s)n}$, where $\beta n$ sentences $\{s_i\}_{i=1}^{\beta n}$ are from $m(s)$ and $(\alpha-\alpha_s)n$ sentences $\{s_i\}_{i=\beta n+1}^{(1-\alpha_s)n)}$ are from $h(s)$. Given $p\left(s^h \in A\right) = q$ and $p\left(s^m \in A\right) = q + \delta$, we can deduce that, on average, $(1-\alpha_s)qn + \beta\delta n$ number of sentences will be in $A$. Similarly, for human-generated text, we collect $(1-\alpha_s)n$ sentences $\{s_i\}_{i=1}^{(1-\alpha_s)n}$ from $h(s)$ and on average $(1-\alpha_s)qn$ number of sentences will be in $A$. Therefore, applying the Chernoff bound, we have

$$
\mathbb{P}\left(\text{ at least }\left((1-\alpha_s)q + \frac{\beta\delta}{2}\right)n \text{ sentences of mixed text are in } A\right) \leq \exp^{-\frac{\beta^2\delta^2 n}{2(1-\alpha_s)}}
\tag{24}
$$

and

$$
\mathbb{P}\left(\text{ at most }\left((1-\alpha_s)q + \frac{\beta\delta}{2}\right)n \text{ sentences of human-written text are in } A\right) \leq \exp^{-\frac{\beta^2\delta^2 n}{2(1-\alpha_s)}}
\tag{25}
$$

Let $A'$ denote the set of $(1-\alpha_s)n$-tuples containing more than $\left((1-\alpha_s)q + \frac{\beta\delta}{2}\right)n$ sentences of $A$. If the mixed text distribution is defined as $m^{\otimes\beta n}h^{\otimes(\alpha-\alpha_s)n}$, and the human-generated distribution is $h^{\otimes(1-\alpha_s)n}$, we can bound

$$
\begin{aligned}
\mathrm{TV}\left(m^{\otimes\beta n}h^{\otimes(\alpha-\alpha_s)n}, h^{\otimes(1-\alpha_s)n)}\right) &\geq p\left(\{s_i^m\}_{i=1}^{\beta n}\cup\{s_i^h\}_{i=1}^{(\alpha-\alpha_s)n} \in A'\right) - p\left(\{s_i^h\}_{i=1}^{(1-\alpha_s)n} \in A'\right) \\
&\geq \left(1 - \exp^{-\frac{\beta^2\delta^2 n}{2(1-\alpha_s)}}\right) - \exp^{-\frac{\beta^2\delta^2 n}{2(1-\alpha_s)}} \\
&= 1 - 2\exp^{-\frac{\beta^2\delta^2 n}{2(1-\alpha_s)}}.
\end{aligned}
$$

If we define $\beta_s = \frac{\beta}{\sqrt{1-\alpha_s}}$, then TV $\left(m^{\otimes \beta n} h^{\otimes(\alpha-\alpha_s)n}, h^{\otimes(1-\alpha_s)n}\right) \geq 1 - 2\exp^{-\frac{\beta_s^2 \delta^2 n}{2}}$. Similar to the proof of Theorem 1, we have

$$n \geq \frac{1}{\beta_s^2 \delta^2} \ln\left(\frac{2}{1-\epsilon}\right) = \frac{1-\alpha_s}{(1-\alpha)^2 \delta^2} \ln\left(\frac{2}{1-\epsilon}\right) \tag{26}$$

The theorem is proved. $\square$

### E.4 PROOF OF THEOREM 4

*Proof.* Similar to the proof for Theorem 3, if we define $p\left(s^h \in A\right) = q$, it implies that $p\left(s^m \in A\right) = q + \delta$. Then, in the human-machine mixed text of $(1-\alpha_s)n$ sentences $\{s_i\}_{i=1}^{(1-\alpha_s)n}$, where $\beta n$ sentences $\{s_i\}_{i=1}^{\beta n}$ are from $m(s)$ and $(\alpha-\alpha_s)n$ sentences $\{s_i\}_{i=\beta n+1}^{(1-\alpha_s)n)}$ are from $h(s)$. Given $p\left(s^h \in A\right) = q$ and $p\left(s^m \in A\right) = q + \delta$, we can deduce that, on average, $(1-\alpha_s)qn + \beta\delta n$ number of sentences will be in $A$. Similarly, for human-generated text, we collect $(1-\alpha_s)n$ sentences $\{s_i\}_{i=1}^{(1-\alpha_s)n}$ from $h(s)$ and on average $(1-\alpha_s)qn$ number of sentences will be in $A$. Therefore, applying Lemma 5, we have

$$\mathbb{P}\left(\text{ at least } \left((1-\alpha_s)q + \frac{\beta\delta}{2}\right)n \text{ sentences of mixed text are in } A\right) \leq \exp^{\frac{-2\left(\beta n \frac{\delta}{2} - \sum_{j=1}^{L}(c_j-1)\rho_j\right)^2}{(1-\alpha_s)n}} \tag{27}$$

and

$$\mathbb{P}\left(\text{ at most } \left((1-\alpha_s)q + \frac{\beta\delta}{2}\right)n \text{ sentences of human text are in } A\right) \leq \exp^{\frac{-2\left(\beta n \frac{\delta}{2} - \sum_{j=1}^{L}(c_j-1)\rho_j\right)^2}{(1-\alpha_s)n}} \tag{28}$$

Let $A'$ denote the set of $(1-\alpha_s)n$-tuples containing more than $\left((1-\alpha_s)q + \frac{\beta\delta}{2}\right)n$ sentences of $A$. If the mixed text distribution is defined as $m^{\otimes \beta n} h^{\otimes(\alpha-\alpha_s)n}$, and the human-generated distribution is $h^{\otimes(1-\alpha_s)n}$, we can bound

$$\text{TV}\left(m^{\otimes \beta n} h^{\otimes(\alpha-\alpha_s)n}, h^{\otimes(1-\alpha_s)n}\right) \geq p\left(\{s_i^m\}_{i=1}^{\beta n} \cup \{s_i^h\}_{i=1}^{(\alpha-\alpha_s)n} \in A'\right) - p\left(\{s_i^h\}_{i=1}^{(1-\alpha_s)n} \in A'\right)$$

$$\geq \left(1 - \exp^{\frac{-2\left(\beta n \frac{\delta}{2} - \sum_{j=1}^{L}(c_j-1)\rho_j\right)^2}{(1-\alpha_s)n}}\right) - \exp^{\frac{-2\left(\beta n \frac{\delta}{2} - \sum_{j=1}^{L}(c_j-1)\rho_j\right)^2}{(1-\alpha_s)n}}$$

$$= 1 - 2\exp^{\frac{-2\left(\beta n \frac{\delta}{2} - \sum_{j=1}^{L}(c_j-1)\rho_j\right)^2}{(1-\alpha_s)n}}. \tag{29}$$

If we need the AUROC of the best possible detector to satisfy $AUROC \geq \epsilon$, we need

$$\frac{1}{2} + \left(m^{\otimes \beta n} h^{\otimes(\alpha-\alpha_s)n}, h^{\otimes(1-\alpha_s)n}\right) - \frac{\left(m^{\otimes \beta n} h^{\otimes(\alpha-\alpha_s)n}, h^{\otimes(1-\alpha_s)n}\right)^2}{2} \geq \epsilon \tag{30}$$

Since the left-hand side is the monotonically increasing function of TV $\left(m^{\otimes \beta n} h^{\otimes(\alpha-\alpha_s)n}, h^{\otimes(1-\alpha_s)n}\right)$, combined with Eq. (29), it holds from the minimum value that

$$\frac{1}{2} + \left(1 - 2\exp^{\frac{-2\left(\beta n \frac{\delta}{2} - \sum_{j=1}^{L}(c_j-1)\rho_j\right)^2}{(1-\alpha_s)n}}\right) - \frac{\left(1 - 2\exp^{\frac{-2\left(\beta n \frac{\delta}{2} - \sum_{j=1}^{L}(c_j-1)\rho_j\right)^2}{(1-\alpha_s)n}}\right)^2}{2} \geq \epsilon \tag{31}$$

After expanding the squares and rearranging the terms, we get

$$\frac{1-\epsilon}{2} \geq \exp^{\frac{-4\left(\beta n \frac{\delta}{2} - \sum_{j=1}^{L}(c_j-1)\rho_j\right)^2}{(1-\alpha_s)n}} \tag{32}$$

Let's denote $\alpha = \sum_{j=1}^{L} (c_j - 1) \rho_j$ and $\gamma(\epsilon) = (1 - \alpha_s) \ln \left( \frac{2}{1-\epsilon} \right)$. Further rearranging, we get

$$\ln \gamma(\epsilon) \leq \frac{4}{n} \left( \beta n \frac{\delta}{2} - \alpha \right)^2 = n\beta^2 \delta^2 - 4\alpha\beta\delta + \frac{4}{n}\alpha^2 \tag{33}$$

Then similar to the proof of Theorem 3 and using $(1 - \alpha_s) \ln \left( \frac{2}{1-\epsilon} \right)$ to replace $\ln \left( \frac{2}{1-\epsilon} \right)$, we have

$$n = \Omega \left( \frac{1 - \alpha_s}{(1 - \alpha)^2 \delta^2} \ln \left( \frac{1}{1 - \epsilon} \right) + \frac{1}{(1 - \alpha)\delta} \sum_{j=1}^{L} (c_j - 1)\rho_j \right.$$

$$\left. + \left( \frac{1 - \alpha_s}{(1 - \alpha)^3 \delta^3} \cdot \left( \sum_{j=1}^{L} (c_j - 1) \rho_j \right) \cdot \ln \left( \frac{1}{1 - \epsilon} \right) \right)^{1/2} \right).$$

The theorem is proved. $\qquad\qquad\qquad\qquad\qquad\qquad\qquad\qquad\qquad\qquad\qquad\qquad\square$

## F    MORE DISCUSSION OF THE PROPOSED FRAMEWORK

### F.1    THEORETICAL RESULTS

**Relation between Theory and Detection Methods**. The proposed theoretical bounds aimed at exploring the relationship between maximum detection potential and data (mixed texts), rather than specific detection methods. This data-driven perspective allows us to propose a conceptually enhanced framework rather than designing specific detectors, making it more applicable. Therefore, in the paper, we are pleased to see enhancements to various detection methods, rather than proposing a specific detection method.

**Relation between Theory and the Proposed Stacked Enhancement Framework**. Theorem 1 and 3 reveal that the detection challenge (sample complexity $n$) is directly proportional to the mixed degree $\alpha$. This finding motivates us to filter human parts in mixed texts (i.e., reduce the mixed degree $\alpha$) to enhance detection, which is the core idea of the proposed framework. Meanwhile, Theorem 2 and 4 provide theoretical bounds for the proposed framework. They have a smaller complexity when compared with Theorems 1 and 3 (Theorem 1 vs. 2, Theorem 3 vs. 4), thus theoretically proving its enhanced effectiveness.

### F.2    RATIONALITY OF THE APPROXIMATION OF $z_i$

In the main text, we argue the effect of different $z_i$ on $p(z_i|S_i; \theta^t)$ is usually much greater than $p(y_i|S_i, z_i; \theta^t)$, which allows us to reasonably approximate $z_i \approx \arg\max_{z_i} p(z_i|S_i; \theta^t)$. We use proof by contradiction to show that this approximation is mild.

Specifically, let $z_1 = \arg\max_{z_i} p(z_i|S_i; \theta^t)$, $z_2 = \arg\max_{z_i} p(y_i|S_i, z_i; \theta^t)p(z_i|S_i; \theta^t)$. If the approximation is unreasonable, then $z_1 \neq z_2$, let us assume that there are $k$ positions between $z_1$ and $z_2$ that are different. For a well-trained detector, the detection results are usually close to 0 or 1. For example, it is reasonable to assume that $p(z_{i,j}|S_i; \theta^t) > 0.95$ or $< 0.05$, which is often more serious in reality. Then, we know $p(z_1|S_i; \theta^t) > 20^k p(z_2|S_i; \theta^t)$. If $z_2 = \arg\max_{z_i} p(y_i|S_i, z_i; \theta^t)p(z_i|S_i; \theta^t) \neq z_1$, then at least $p(y_i|S_i, z_1; \theta^t) > 20^k p(y_i|S_i, z_2; \theta^t)$ must be satisfied. Since $p(y_i|S_i, z_1; \theta^t)$ represents the detection probability of paragraph $S_i \odot z_1$, this means that modifying two subsequences of a paragraph can change the detection probability by a factor of 400, which is generally difficult. Therefore, our approximation is mild.

### F.3    DIFFERENCES FROM EXISTING MIXED TEXT DETECTION WORKS

Despite some recent attention (Zhang et al., 2024b; Wang et al., 2024) to human-machine mixed text detection, our work differs significantly from them in several key aspects:

- **Scenario**. Different from previous research focused solely on pure mixed texts combining human-written and machine-generated content, we extend our investigation to implicit human-machine mixed texts: even if the text is entirely machine-generated, LLMs, with their advanced generative capabilities, can produce text indistinguishable from human-written content. Considering the difficulty of detecting mixed texts, our work highlights that the actual detection challenge may be far more severe than recognized, even when the text is entirely machine-generated.

- **Evaluation**. Previous studies experimentally demonstrated the challenges of mixed text detection in specific datasets, lacking theoretical guarantees in machine learning. Accordingly, our work takes the first step to theoretically elucidate the detrimental impact of mixed texts on detection efforts. Furthermore, our theoretical findings help identify factors influencing detection and inspire innovative detection algorithms.

- **Methodology**. Unlike previous research, which did not design detectors specifically for mixed-text scenarios, we propose a theory-inspired enhancement strategy. This strategy is applicable to existing model-based detectors and has been extensively validated through comprehensive experiments.

### F.4 Differences from Other Hard-EM-based Methods

Although hard-EM has also been used in areas such as weak supervision to estimate unknown (noisy) labels, our application is tailored specific to MGT detection. Specifically, in MGT detection, it is necessary to jointly model multiple subsequences. Directly applying hard-EM here requires enumerating all possible label combinations for these sequences, yielding an exponential complexity of $\mathcal{O}(2^n)$, where $n$ is the number of subsequences. To this end, we leverage the fact that the effect of $z_i$ on $p(z_i|S_i; \theta_t)$ is often much larger than the change in $p(y_i|S_i, z_i; \theta_t)$, and introduce a mild assumption $z_i \approx \arg\max_{z_i} p(z_i|S_i; \theta^t)$, thereby reducing the estimation problem back to a linear, tractable $\mathcal{O}(n)$.

### F.5 Formal Description of the Proposed Framework's Validity

The effectiveness of the proposed framework can be understood from three aspects: information bottleneck theory, attention mechanism, and the granularity of text processing:

- **Bottleneck theory**, which states that during forward propagation, a neural network filters out information irrelevant to the prediction, gradually focusing on the most crucial parts of the input (Guan et al., 2019). According to this theory, the effectiveness of existing detectors in mixed texts arises from the layer-by-layer filtering of noise information (i.e., human-generated text in machine-generated text). Our proposed stacked framework explicitly filters out a large portion of noise information, enabling more efficient learning of key discriminative sequences. Let us denote information in a random variable $Z$ as $I(Z)$, the machine part text in the mixed text $S$ as $S_m$, the original detector and our enhanced version as $f_{origin}$ and $f_{our}$, then we have $I(S) \geq I(\hat{S}) \geq I(S_m) \geq I(f_{our}(S_m))$ according to the architecture in the framework. Furthermore, according to the prediction goal of the detector, we have $I(f_{our}(S_m)) = I(f_{origin}(S))$. Therefore, $I(S) \geq I(\hat{S}) \geq I(S_m) \geq I(f_{our}(S_m)) = I(f_{origin}(S))$. According to the bottleneck theory, the proposed stacked framework achieves iterative collaborative self-enhancement through the same detector: (1) in the E-step, the detector filters out "human" sequences, thereby supplying higher-quality text to the same detector in the M-step; (2) the higher-quality model learned in the M-step then serves as a detector in the next E-step, filtering out more accurate "human" sequences.

- **Attention mechanism**. Existing theoretical results (Theorem 3.1 in (Liu et al., 2023)) establish the time consistency of attention weights: if the $l$-th token's attention weight $\alpha_{t,l}$ is large at step $t$, it $\alpha_{t+1,l}$ likely remains large at step $t+1$. Thus, once important tokens are identified, they continue to impact detection, regardless of other unimportant tokens. Corresponding to the proposed framework, since it retains the key machine text (i.e., the important tokens that determine the prediction as machine) in the mixed text, these influential sub-sequences consistently affect detection over time, even if the other human text (tokens that are not important for prediction as machine) is removed.

- **Granularity of text processing**. Most models default to categorizing a text $S$ as either machine or human (the format $S$, where $S \sim h(S)$ or $m(S)$), while we refine this to the sentence level (the

format $S = [s_1; s_2; ...; s_n]$ where each $s_i \sim h(s)$ or $m(s)$). This allows us to focus on key local inconsistencies between machine-generated and human-generated text during detection, leading to better detection. Our work takes the first step from the paragraph level to the sentence level, but undeniably, an even finer granularity is to further understand from sentence level to clause level (the format $s_i = "a_1, a_2, ..., and a_n"$ where each $a_i \sim h(s)$ or $a_i \sim m(s)$), which may provide valuable insights for future detection enhancements.

### F.6  TIME COMPLEXITY

For Transformer-based detectors, assuming the text length is $N$ and the embedding dimension is $d$, the time complexity of the original detector is $O(dN^2)$. In our stacked detection framework, the E-step divides the long text into several sequences of lengths $\{N_i\}_i$, resulting in a time complexity of $\mathcal{O}(\sum_i dN_i^2) = \mathcal{O}(dN^2)$, which is usually lower than that of the original detector in practice. For the M-step, the complexity is also $\mathcal{O}(dN^2)$. Since the length of the filter text does not exceed $N$, it is also not higher than that of the original detector. Therefore, our complexity is $\mathcal{O}(dN^2)$, and the actual running time does not exceed twice that of the original detector. We will further discuss the time complexity from the perspective of empirical experiments in Appendix H.13.

### F.7  LIMITATION

The proposed framework has the following potential limitations:

- To maintain the framework's flexibility and efficiency, we adopt a simple fixed-length sequence approach, which may split non-independent sequences. Encouragingly, extensive experimental results (Table 1-17) demonstrate that this straightforward approximation strategy already achieves satisfactory enhancement effects. Even so, this indeed presents a promising direction for future work to further enhance detection.

- The proposed framework is applied only to model-based detectors, but its core idea of filtering human portions from mixed texts is universal, which may inspire future research.

- The effectiveness of the proposed stacked framework lies in accurately determining whether a subsequence belongs to the MGT in the first step. If a detector is incapable of recognizing subsequences (i.e., very weak detectors), we cannot guarantee our effectiveness. However, the significance of enhancing such detectors is minimal, as poor detection capability makes them difficult to apply in practice. We encourage more attention to enhancing detectors that perform well.

- The proposed method is generally limited to short texts. We believe that starting with paragraph-level detection and exploring shorter texts in future research is promising and likely to be an ongoing trend.

- The mathematical framework used in the theoretical analysis relies on a fixed mixing ratio $\alpha$. This design aims to clearly isolate the core mechanism of the mixing ratio, ensuring tractability when analyzing how mixing factors affect detection performance. We hope to take the first step to provide a theoretical foundation and intuition for more complex models to be proposed later. Future theoretical research can model more complex scenarios, such as those with variable mixing ratios.

## G  SOFT STACKED DETECTION ENHANCEMENT FRAMEWORK

In addition to the default stacked detection framework optimized by the hard EM algorithm, we provide a soft (traditional) EM version. For the soft E-step, we calculate the expected value of the log-likelihood function of $\theta$, with respect to the current conditional distribution of $z_i$ given $(S_i, y_i)$ and the current estimates of the parameters $\theta^t$:

$$Q(\theta; \theta^t) := \sum_{(S_i, y_i) \in \mathcal{D}} \sum_{z_i} p(z_i | S_i, y_i; \theta^t) \log p(y_i, S_i, z_i; \theta).$$

For the posterior distribution $p(z_i | S_i, y_i; \theta^t)$, similar to the approximation of the hard-EM version, we approximate $p(z_i | S_i, y_i; \theta^t) \approx p(z_i | S_i; \theta^t)$. This can be predicted by the detection model

---

**Algorithm 2** Soft Stacked Detection Enhancement Framework

---

1: **Input:** Train data $\mathcal{D} = \{(S_i, y_i)\}_{i=1}^N$, the detection model $f(S, \theta^0)$, training epochs $T$, filtering ratio $\tau$, E-step detection threshold $r_e$, and learning rate $\eta$.
2: **procedure** TRAIN($\mathcal{D}$)                                              ▷ Detector Training
3:     **for** $t = 0$ **to** $T - 1$ **do**
4:         **for** each batch of samples $\mathcal{D}_B \sim \mathcal{D}$ **do**
5:             Calculate $\mathcal{Q}(\theta, \theta^t)$ according to Eq. (34).                    ▷ E-step
6:             $\theta^{t+1} = \theta + \eta \nabla_\theta \mathcal{Q}(\theta, \theta^t)$.                              ▷ M-step
7:         **end for**
8:     **end for**
9:     **Return** the trained model $f(S, \theta^T)$.
10: **end procedure**
11: **procedure** INFERENCE($S_i, f(S, \theta^t)$)                           ▷ Detector Inference
12:     Split $S_i$ into a set of sequences $\{S_{i,j}\}$, where each $S_{i,j}$ contains at most $k$ sentences.
13:     Calculate $z_i$ according to Eq. (3).
14:     $\hat{S}_i = S_i \odot z_i$.
15:     **Return** $f(\hat{S}_i, \theta)$.
16: **end procedure**

---

$f(S, \theta)$, i.e.,

$$p(z_i \mid S_i; \theta^t) := \prod_{j=1}^{n_i} f\left(S_{i,j}, \theta^t\right)^{z_i} \left(1 - f\left(S_{i,j}, \theta^t\right)\right)^{1-z_i}$$

For each $z_i$, we can model $p(y_i | S_i, z_i; \theta)$ as:

$$p(y_i | S_i, z_i; \theta) = f(S_i \odot z_i, \theta)$$

Therefore, we have

$$Q(\theta, \theta^t) = \sum_{(S_i, y_i)} \sum_{z_i} \prod_{j=1}^{n_i} f\left(S_{i,j}, \theta^t\right)^{z_i} \left(1 - f\left(S_{i,j}, \theta^t\right)\right)^{1-z_i}$$
$$(y_i \log\left(f(S_i \odot z_i, \theta)\right) + (1 - y_i) \log\left(1 - f(S_i \odot z_i, \theta)\right)) \tag{34}$$

In the M-step, we need to maximize $Q(\theta, \theta^t)$ to update the model parameters:

$$\theta^{t+1} = \arg\max_\theta \mathcal{Q}(\theta, \theta^t), \tag{35}$$

which can be updated by the gradient descent algorithm.

For the inference phase, we use the same processing as the hard version described in the main text and calculate $z_i$ according to Eq. (3), and the model's prediction is $f(S_i \odot z_i, \theta)$. The detailed algorithm flow is shown in Alg. 2. Besides, an experimental evaluation of this version is given in Appendix H.11.

## H  ADDITIONAL EXPERIMENTS

### H.1  DATASETS AND BASELINES

The detailed descriptions of the dataset are shown as follows:

- **Essay** (Verma et al., 2024). This dataset comprises 1,000 samples derived from essays found on IvyPanda, spanning a range of subjects and educational levels, from high school to university. For the dataset construction, the researchers employ ChatGPT-turbo to create a $< prompt >$ that aligns with each essay's content. Subsequently, this crafted prompt serves as input for various LLMs (ChatGPT, GPT-4, ChatGPT-turbo, ChatGLM, Dolly, and Claude) to produce corresponding essays. This process allows for the generation of diverse essay samples based on the original content.

- **Reuters** (Verma et al., 2024). This dataset is based on the Reuters 50-50 authorship identification dataset, encompassing 1,000 news articles authored by 50 different journalists, with each contributing 20 pieces. Similar to the generated process of the Essay dataset, the researchers initially utilized ChatGPT-turbo to generate a $< headline >$ for each article. These generated headlines were then employed to formulate prompts, which were subsequently used to query various LLMs (ChatGPT, GPT-4, ChatGPT-turbo, ChatGLM, Dolly, and Claude) to produce MGTs.

- **SQuAD1** (He et al., 2023). This dataset is derived from the SQuAD1 dataset (Rajpurkar, 2016) and comprises 1000 entries of context-based inquiries. Each entry features a single human-provided response alongside six answers generated by LLMs (ChatGPT, GPT-4, ChatGPT-turbo, ChatGLM, StableLM). To simulate the mixed text scenario, each sample in this dataset is concatenated into a question and an answer, e.g., Q1+human answer, Q1+machine answer. Therefore, the common question part makes them considered as mixed text, and the question parts can be considered human texts.

- **DetectRL** (Wu et al., 2024). The text of this dataset is collected from arXiv academic abstracts, XSum news, Writing Prompts stories, and Yelp Reviews, and MGTs are generated using four LLMs: GPT-3.5-turbo (ChatGPT), PaLM-2-bison (PaLM), Claude-instant (Claude), and Llama-2-70b (Llama-2). The paraphrased text is generated by Dipper paraphraser (Krishna et al., 2023), Polish (polished using LLM), and Back Translation (Google Translate from English to Chinese and then to English).

All datasets are randomly divided into the training, validation, and test sets with a ratio of 2: 1: 1.

The specific details of the baseline methods are shown as follows:

- **GPTZero** (GPTZero, 2023). It is a commercially available AI detector that employs an end-to-end deep learning approach, trained on text datasets from the web, education, and AI generated from a range of LLMs.

- **Likelihood** (Solaiman et al., 2019). It is a simple "zero-shot" baseline using a threshold on the total probability of an LLM. Here, the LLM is gpt2-medium.

- **Rank** (Gehrmann et al., 2019). It uses a threshold on the average rank of words to identify whether the text is sampled from the top of the distribution of the LLM. Here gpt2-medium is also adopted.

- **Log-Rank** (Mitchell et al., 2023). It uses a threshold on the average log-rank of each word in the text, and the calculated LLM is also gpt2-medium.

- **DetectGPT** (Mitchell et al., 2023). It assumes that the text generated by LLM is usually located near the local minimum point in the log-probability range of the model, and thus evaluates the text by quantifying how small perturbations affect the log-probability under LLM.

- **Fast-DetectGPT** (Bao et al., 2024). The idea is similar to DetectGPT, but a more efficient sampling step is used to replace the perturbation step of DetectGPT.

- **ChatGPT-D** (Guo et al., 2023). It uses pure answered text or QA pairs to train a detection model (here using the RoBERTa model) using the HC3 dataset.

- **OpenAI-D** (Solaiman et al., 2019). It fine-tunes a RoBERTa model with GPT2-generated texts, which is designed mainly for detecting GPT2 outputs.

- **MPU** (Tian et al., 2024). It proposes a Multiscale Positive-Unlabeled (MPU) training framework and is trained with MPU from a pretrained RoBERTa-Base model.

- **RADAR** (Hu et al., 2023). It learns a robust detection model by using adversarial training of a paraphraser and a detector, where the paraphraser aims to generate realistic content to evade AI text detection, while the detector tries to detect such content.

### H.2 IMPLEMENTATION DETAILS

For fair comparison, both the model-based baselines (i.e., ChatGPT-D, OpenAI-D, MPU, RADAR) and the enhanced version are fine-tuned on the training set and use the same hyperparameters. Specifically, all detection models are fine-tuned for 5 epochs, and the Adam optimizer is used for training, with a learning rate of 5e-5. The hyperparameter setting is tuned once for each (Base Detector + Dataset) combination. Specifically, for ChatGPT-STK, the E-step detection threshold $r_e$ is set to 0.01 in Essay, Reuters, and SQuAD1 datasets, the length $k$ is set to 3, 3, 1, and the maximum

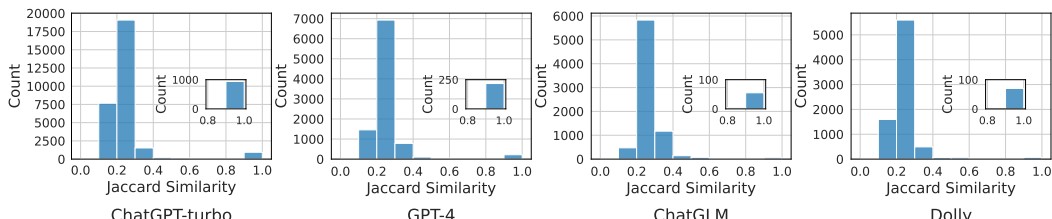

Figure 8: Sencence word's Jaccard similarity between human-written texts and MGTs in the Essay dataset. Texts with 100% similarity and the same order (in fact, all texts with 100% similarity in our experiments met this condition) indicate the existence of implicit mixed text.

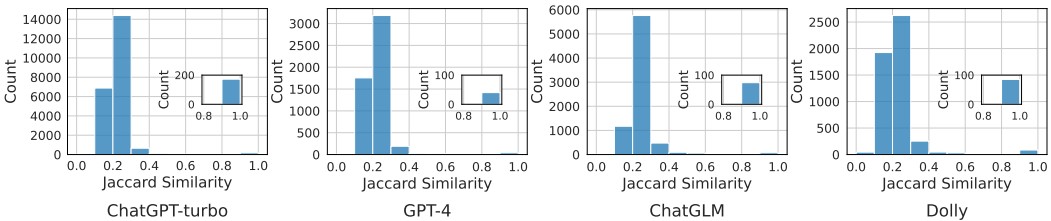

Figure 9: Sencence word's Jaccard similarity between human-written texts and MGTs in the Reuters dataset. Texts with 100% similarity and the same order (in fact, all texts with 100% similarity in our experiments met this condition) indicate the existence of implicit mixed text.

filter ratio $\tau$ is set to 0.2, 0.2, and 0.8. For OpenAI-STK, the E-step detection threshold $r_e$ is set to 0.01, 0.005, 0.005 in Essay, Reuters, and SQuAD1 datasets, the length $k$ is set to 4, 4, 1, and the maximum filter ratio $\tau$ is set to 0.5. For MPU-STK and RADAR-STK, the E-step detection threshold $r_e$ is set to 0.005 in Essay, Reuters, and SQuAD1 datasets, the length $k$ is set to 7, 3, 1, and the maximum filter ratio $\tau$ is set to 0.5. For more parameter analysis, see Appendix H.13. Notably, the default settings are obtained through grid search, aiming to show the maximum potential of detection enhancement. Our sensitivity analysis (Appendix H.13) shows that compared to the original detection models (with the x-axis value of 0), the enhanced models generally outperform them across various hyperparameter settings, reducing the need for performance tuning and have wide practical applicability. For the sentence split, we use the sent_tokenize function from the nltk tool library to divide texts into sentences. For feature-based methods, we use the default implementation of MGTBench [2]. The experiments are conducted on a PC with an Inter(R) Xeon(R) Gold 6230 CPU @ 2.10GHz, 60GB memory, and a NVIDIA Tesla V100.

## H.3 ADDITIONAL RESULTS FOR IMPLICITLY MIXED TEXT

It is worth noting that the Jaccard similarity is calculated as follows: for each MGT sentence in the dataset, the human-generated sentences in the dataset are traversed, and the sentence with the greatest Jaccard similarity to the MGT sentence is selected as the Jaccard similarity between the MGT and the human text. Fig. 8 and Fig. 9 show the Jaccard similarity of words in LLMs and human-generated sentences in two pure MGT datasets (the SQuAD1 dataset consists of explicit mixed text and is therefore not evaluated). Furthermore, considering the diversity of texts, we verified the implicit mixed phenomenon in Chinese texts Tao et al. (2024) and code texts Hayawi et al. (2024), as shown in Fig. 10. It can be seen that even with a small amount of human text used for evaluation, using this strict similarity, the existence of implicit mixed text can be demonstrated.

## H.4 MORE RESULTS FOR PERFORMANCE COMPARISON

First, in addition to the performance comparison of detectors trained on ChatGPT texts shown in Table 1 in the main text, we also extensively evaluated the effectiveness of the detectors trained on different LLM texts to demonstrate the versatility of the proposed strategy. The results of these

---

[2]https://github.com/xinleihe/MGTBench

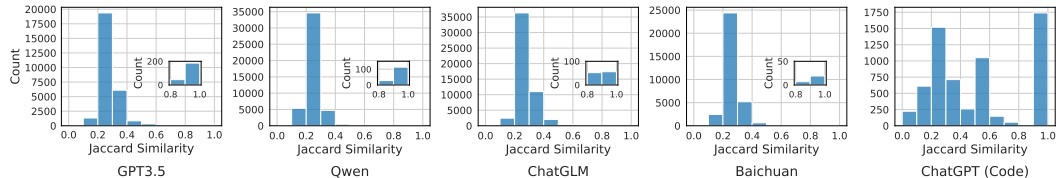

Figure 10: Sencence word's Jaccard similarity between human-written texts and MGTs in the Chinese texts (first four sub-figures) and code texts (last sub-figure). Texts with 100% similarity and the same order (in fact, all texts with 100% similarity in our experiments met this condition) indicate the existence of implicit mixed text.

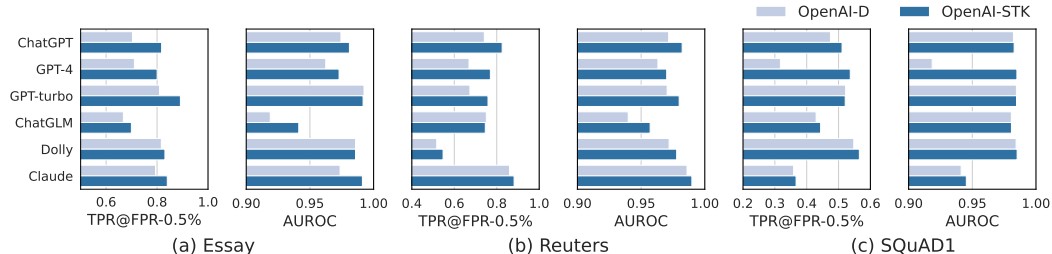

Figure 11: Average detection performance (x-axis) of detectors (OpenAI-D and our boosting strategy OpenAI-STK) tested across various LLMs, where these detectors are trained on texts generated by specific LLM (y-axis).

evaluations are presented in Table 2-13. From these tables, it is evident that the proposed strategy effectively enhances original detectors (i.e., ChatGPT-D, OpenAI-D, and MPU) in most settings. For example, the proposed strategy demonstrates a significant improvement for ChatGPT-D in approximately 98% of the settings (194 out of 198), and it offers a competitive improvement for OpenAI-D in approximately 86% of the settings (169 out of 198). The widespread enhancement on unknown LLM texts underscores the practical applicability of the proposed strategy, as it aligns more closely with real-world scenarios. Besides, we find that all detectors exhibit low performance on SQuAD1. The reason is that SQuAD1 is an explicit mixed-text dataset where every sample consists of a human-written question concatenated with either a human or machine answer, which echoes our theoretical results: the mixed text hinders detection. Finally, our enhancement is less pronounced on SQuAD1. This is because that our framework's effectiveness relies on the base detector's ability to confidently identify and filter the "human-like" parts in the E-step. If a detector is incapable of recognizing subsequences (i.e., very weak detectors), we cannot guarantee our effectiveness. Nevertheless, our enhancement effect is encouraging and remains positive.

Second, in addition to the enhancements shown in Fig. 3 for ChatGPT-D, Fig. 11-13 demonstrate the improved performance for three other model-based detectors used in the experiments, OpenAI-D, MPU, and RADAR. We observe similar findings as in Section 4.2, where the original models' detection performance is enhanced in most settings, with a more pronounced improvement for weaker detectors.

Third, we include ROC curves in Fig. 14. As observed, when focusing on extremely low FPR (0%–5%), the enhanced detectors show significant advantages, which is highly valuable for MGT detection that prioritize low false positives.

Finally, we further demonstrate the broad effectiveness of our augmentation framework on the Chinese dataset Tao et al. (2024) and the code dataset Hayawi et al. (2024). Table 14 presents the performance of both the original detectors and their enhanced versions (-E) for detecting Chinese MGTs and code MGTs. We observe that the proposed enhancement strategy still exhibits promising potential for both Chinese and code texts.

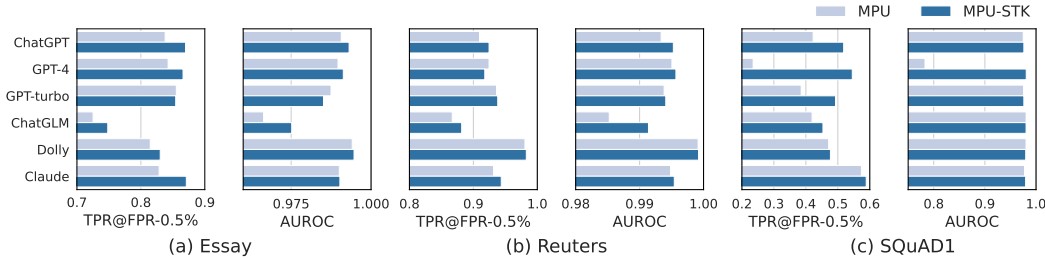

Figure 12: Average detection performance (x-axis) of detectors (MPU and our boosting strategy MPU-STK) tested across various LLMs, where these detectors are trained on texts generated by specific LLM (y-axis).

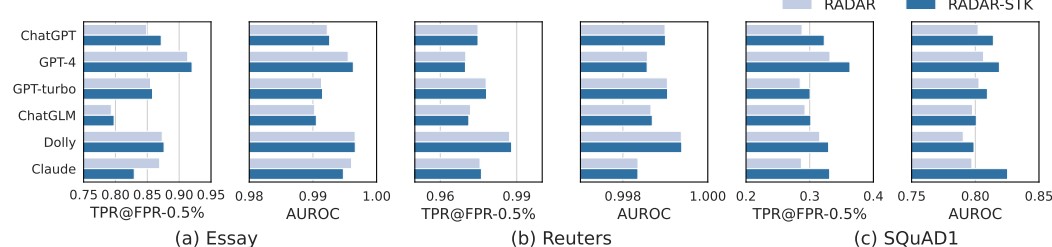

Figure 13: Average detection performance (x-axis) of detectors (RADAR and our boosting strategy RADAR-STK) tested across various LLMs, where these detectors are trained on texts generated by specific LLM (y-axis).

### H.5   EVALUATION OF EXPECTED COST UNDER PLAUSIBLE BASE RATES

In this section, we provide a detailed analysis of total expected cost as a function of the base rate. Concretely, we define a formal expected cost function:

$$E(Cost) = P(Human) \cdot P(FP) \cdot C_{FP} + P(Machine) \cdot P(FN) \cdot C_{FN}.$$

where $P(Human)$ and $P(Machine)$ are the base rates. Besides, we model a plausible cost asymmetry, $C_{FP} = 10$ and $C_{FN} = 1$, to reflect the significant repercussions of misclassifying human-generated text. Table 15 reports the total expected cost across a range of plausible MGT base rates (10% to 50%), demonstrating that our method achieves lower total expected cost in imbalanced real-world scenarios. Notably, this setup corresponds to the ChatGPT column for the Essay task in Table 1, where MPU and MPU-STK both display consistent, superior performance (e.g., 99.92%), achieving the same expected cost.

### H.6   MORE RESULTS FOR ROBUSTNESS COMPARISON

In addition to the robustness results shown in the main text, Fig. 15 shows the robust performance on OpenAI-D. It can be seen that the proposed strategy can also improve the robustness of the detection model in adversarial environments, which is very valuable.

### H.7   PERFORMANCE COMPARISON UNDER SHORTER TEXTS

Our paper mainly focuses on paragraph-level detection. This is because the core mechanism of our stacked framework relies on filtering text into potentially shorter segments, which fundamentally performs better on longer, document-level content. However, in addition to sentence-level detection, given that our framework does not have a limit on text length, this section will evaluate the proposed framework on shorter texts.

To this end, we tested on the Essay dataset, limiting the maximum text length to 64 words. The evaluation of detection performance is shown in Table 16. Compared with the performance for longer

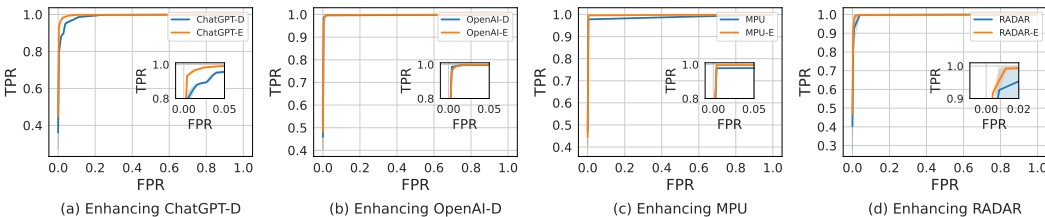

(a) Enhancing ChatGPT-D     (b) Enhancing OpenAI-D     (c) Enhancing MPU     (d) Enhancing RADAR

Figure 14: ROC curves when detecting ChatGPT texts on Essay dataset. Detectors are trained on ChatGPT texts.

Table 2: Performance concerning TPR@FPR-0.5%. Detectors are trained on ChatGPT text.

| Dataset | Method | ChatGPT | GPT-4 | GPT-turbo | ChatGLM | Dolly | Claude/StableLM | Avg. |
|---|---|---|---|---|---|---|---|---|
| | Log-Likelihood | $1.04_{\pm 0.57}$ | $5.00_{\pm 2.93}$ | $2.11_{\pm 0.29}$ | $1.85_{\pm 1.12}$ | $5.93_{\pm 3.44}$ | $21.85_{\pm 10.14}$ | 6.30 |
| | Rank | $2.43_{\pm 0.43}$ | $3.81_{\pm 1.04}$ | $1.87_{\pm 0.68}$ | $1.97_{\pm 0.28}$ | $4.30_{\pm 1.01}$ | $7.98_{\pm 1.52}$ | 3.73 |
| | Log-Rank | $1.97_{\pm 1.19}$ | $4.88_{\pm 3.25}$ | $2.34_{\pm 0.83}$ | $1.97_{\pm 1.35}$ | $4.88_{\pm 2.72}$ | $19.77_{\pm 11.36}$ | 5.97 |
| | DetectGPT | $0.35_{\pm 0.69}$ | $0.36_{\pm 0.48}$ | $0.47_{\pm 0.44}$ | $0.58_{\pm 0.73}$ | $0.58_{\pm 0.37}$ | $0.81_{\pm 0.28}$ | 0.52 |
| | F-DetectGPT | $4.28_{\pm 1.35}$ | $6.90_{\pm 1.53}$ | $2.69_{\pm 1.41}$ | $2.77_{\pm 1.12}$ | $5.35_{\pm 2.19}$ | $28.44_{\pm 3.93}$ | 8.41 |
| SQuAD1 | ChatGPT-D | $1.39_{\pm 1.07}$ | $1.19_{\pm 0.84}$ | $1.64_{\pm 1.07}$ | $0.81_{\pm 0.46}$ | $1.28_{\pm 1.13}$ | $5.43_{\pm 1.85}$ | 1.96 |
| | **ChatGPT-STK** | $1.16_{\pm 0.73}$ | $4.52_{\pm 1.67}$ | $1.17_{\pm 0.64}$ | $0.81_{\pm 0.46}$ | $1.51_{\pm 1.36}$ | $6.01_{\pm 3.61}$ | 2.53 |
| | OpenAI-D | $\underline{41.85}_{\pm 16.81}$ | $45.00_{\pm 18.20}$ | $37.54_{\pm 16.76}$ | $\underline{29.48}_{\pm 11.36}$ | $58.02_{\pm 24.87}$ | $\underline{72.83}_{\pm 14.76}$ | 47.45 |
| | **OpenAI-STK** | $\mathbf{44.86}_{\pm 17.81}$ | $\underline{50.48}_{\pm 23.10}$ | $41.64_{\pm 21.26}$ | $\mathbf{29.83}_{\pm 13.42}$ | $66.40_{\pm 22.81}$ | $\mathbf{73.41}_{\pm 12.49}$ | 51.10 |
| | MPU | $30.64_{\pm 27.70}$ | $29.40_{\pm 29.38}$ | $\underline{51.23}_{\pm 29.07}$ | $23.01_{\pm 18.60}$ | $\underline{61.05}_{\pm 31.23}$ | $58.38_{\pm 13.31}$ | 42.28 |
| | **MPU-STK** | $37.80_{\pm 17.68}$ | $\mathbf{67.26}_{\pm 22.09}$ | $\mathbf{59.88}_{\pm 22.36}$ | $28.44_{\pm 12.41}$ | $59.65_{\pm 30.22}$ | $57.92_{\pm 11.15}$ | **51.83** |
| | RADAR | $27.05_{\pm 14.62}$ | $30.71_{\pm 10.01}$ | $34.27_{\pm 10.12}$ | $19.54_{\pm 13.00}$ | $31.74_{\pm 4.38}$ | $29.48_{\pm 11.57}$ | 28.80 |
| | **RADAR-STK** | $33.18_{\pm 11.40}$ | $32.26_{\pm 8.40}$ | $38.25_{\pm 5.54}$ | $24.16_{\pm 9.88}$ | $32.33_{\pm 1.93}$ | $33.53_{\pm 11.31}$ | 32.28 |

texts (Table 3), detecting short texts is indeed more challenging, consistent with our theoretical findings (Theorems 1 and 3). Nevertheless, our proposed framework still provides an encouraging improvement in detection.

### H.8 PERFORMANCE COMPARISON W.R.T. ACCURACY

In addition to demonstrating the performance in terms of TPR@FPR-0.5% and AUROC, we also present a performance comparison in terms of accuracy to compare with GPTZero, as shown in Table 17. Consistent with the previous results, the proposed enhancement strategy significantly improves detection accuracy in most settings. For example, in the Essay dataset, the accuracy of OpenAI-D increased from 91.86 to 92.64. Moreover, the zero-shot performance of GPTZero is inferior to the fine-tuned model-based detectors. This is particularly evident in the SQuAD1 dataset, which has more mixed texts, indicating its limitations in detecting mixed texts.

### H.9 PERFORMANCE COMPARISON UNDER DIFFERENT MIXED DEGREES

We constructed explicit mixed test texts with different mixing levels based on the Essay dataset. Specifically, for each test text in the Essay dataset, we replaced $n$ random sentences (ranging from 1 to 5) with human-written ones for the same prompt. The detection performance is presented in Table 16. First, as the level of text mixing increases, all detectors' performance gradually decreases, which verifies our theoretical results that mixed text hurts detection. Second, the superiority of the proposed enhancement framework becomes significant, underscoring the inherent challenge of mixed text detection and highlighting the effectiveness of our method.

### H.10 VISUALIZATION

To more intuitively demonstrate the effectiveness of the proposed enhancement framework, we use t-SNE tool to visualize the last hidden state of ChatGPT-D (first row) and ChatGPT-STK (second row) when detecting different LLM texts (different columns), as shown in Fig. 17. Firstly, it can be observed that compared with the original detectors (first row), the enhanced detectors (second row) can better distinguish between human and machine texts. Secondly, MGTs lack diversity compared to human texts, which is consistent with existing findings (Fröhling & Zubiaga, 2021).

Table 3: Performance concerning AUROC. Detectors are trained on ChatGPT text.

| Dataset | Method | ChatGPT | GPT-4 | GPT-turbo | ChatGLM | Dolly | Claude/StableLM | Avg. |
|---|---|---|---|---|---|---|---|---|
| Essay | Log-Likelihood | $98.48_{\pm0.43}$ | $96.46_{\pm1.05}$ | $98.85_{\pm0.42}$ | $98.78_{\pm0.15}$ | $90.64_{\pm0.35}$ | $92.16_{\pm0.32}$ | 95.90 |
| | Rank | $92.84_{\pm0.99}$ | $90.93_{\pm0.74}$ | $98.19_{\pm0.37}$ | $81.51_{\pm1.32}$ | $80.45_{\pm1.41}$ | $86.69_{\pm1.06}$ | 88.44 |
| | Log-Rank | $98.64_{\pm0.43}$ | $96.85_{\pm0.89}$ | $98.92_{\pm0.40}$ | $98.94_{\pm0.16}$ | $90.41_{\pm0.50}$ | $91.47_{\pm0.45}$ | 95.87 |
| | DetectGPT | $96.86_{\pm0.97}$ | $95.64_{\pm0.45}$ | $44.40_{\pm1.73}$ | $95.60_{\pm0.62}$ | $92.55_{\pm0.58}$ | $46.36_{\pm0.71}$ | 78.57 |
| | F-DetectGPT | $83.86_{\pm1.52}$ | $84.45_{\pm1.58}$ | $94.12_{\pm0.72}$ | $93.62_{\pm0.73}$ | $55.86_{\pm2.54}$ | $58.96_{\pm1.52}$ | 78.48 |
| | ChatGPT-D | $98.71_{\pm0.98}$ | $98.14_{\pm1.63}$ | $91.31_{\pm2.24}$ | $99.73_{\pm0.16}$ | $86.39_{\pm7.11}$ | $48.70_{\pm11.69}$ | 87.16 |
| | **ChatGPT-STK** | $98.98_{\pm1.26}$ | $98.41_{\pm1.91}$ | $92.78_{\pm1.29}$ | $99.80_{\pm0.25}$ | $89.96_{\pm6.56}$ | $61.59_{\pm13.52}$ | 90.25 |
| | OpenAI-D | $99.55_{\pm0.78}$ | $99.73_{\pm0.34}$ | $99.25_{\pm0.53}$ | $99.87_{\pm0.20}$ | $\underline{98.59}_{\pm0.86}$ | $87.63_{\pm2.72}$ | 97.44 |
| | **OpenAI-STK** | $99.87_{\pm0.14}$ | $\underline{99.82}_{\pm0.13}$ | $99.57_{\pm0.10}$ | $99.95_{\pm0.03}$ | $\mathbf{98.75}_{\pm0.44}$ | $90.64_{\pm1.32}$ | 98.10 |
| | MPU | $\underline{100.00}_{\pm0.01}$ | $\mathbf{99.91}_{\pm0.07}$ | $98.63_{\pm0.21}$ | $\underline{99.99}_{\pm0.02}$ | $97.74_{\pm0.46}$ | $98.14_{\pm0.55}$ | 99.07 |
| | **MPU-STK** | $\mathbf{100.00}_{\pm0.00}$ | $\mathbf{99.91}_{\pm0.07}$ | $98.50_{\pm0.27}$ | $\mathbf{99.99}_{\pm0.01}$ | $98.07_{\pm0.39}$ | $\mathbf{99.41}_{\pm0.29}$ | **99.31** |
| | RADAR | $99.93_{\pm0.02}$ | $99.78_{\pm0.09}$ | $\underline{99.85}_{\pm0.05}$ | $99.97_{\pm0.02}$ | $97.61_{\pm0.59}$ | $\underline{98.22}_{\pm0.26}$ | 99.22 |
| | **RADAR-STK** | $99.94_{\pm0.03}$ | $99.72_{\pm0.13}$ | $\mathbf{99.86}_{\pm0.06}$ | $99.97_{\pm0.01}$ | $97.88_{\pm0.62}$ | $98.20_{\pm0.42}$ | _99.26_ |
| Reuters | Log-Likelihood | $97.59_{\pm0.37}$ | $74.85_{\pm0.50}$ | $98.54_{\pm0.38}$ | $99.54_{\pm0.22}$ | $60.09_{\pm2.13}$ | $85.66_{\pm1.04}$ | 86.05 |
| | Rank | $84.00_{\pm0.88}$ | $74.12_{\pm1.24}$ | $91.82_{\pm0.60}$ | $70.97_{\pm1.65}$ | $54.88_{\pm2.45}$ | $67.78_{\pm1.14}$ | 73.93 |
| | Log-Rank | $97.86_{\pm0.35}$ | $79.95_{\pm0.29}$ | $98.82_{\pm0.37}$ | $99.65_{\pm0.20}$ | $61.24_{\pm2.15}$ | $84.88_{\pm1.13}$ | 87.07 |
| | DetectGPT | $92.78_{\pm1.34}$ | $85.71_{\pm2.02}$ | $49.24_{\pm1.65}$ | $91.59_{\pm1.67}$ | $83.41_{\pm1.73}$ | $66.59_{\pm2.53}$ | 78.22 |
| | F-DetectGPT | $96.23_{\pm0.19}$ | $66.02_{\pm1.53}$ | $98.85_{\pm0.30}$ | $98.58_{\pm0.35}$ | $40.38_{\pm1.31}$ | $66.16_{\pm2.58}$ | 77.70 |
| | ChatGPT-D | $99.79_{\pm0.23}$ | $99.14_{\pm0.65}$ | $99.64_{\pm0.37}$ | $99.46_{\pm0.49}$ | $89.21_{\pm5.80}$ | $68.66_{\pm12.94}$ | 92.65 |
| | **ChatGPT-STK** | $99.89_{\pm0.17}$ | $99.29_{\pm0.48}$ | $99.86_{\pm0.20}$ | $99.54_{\pm0.17}$ | $91.13_{\pm3.36}$ | $84.61_{\pm6.87}$ | 95.72 |
| | OpenAI-D | $99.80_{\pm0.20}$ | $99.46_{\pm0.31}$ | $99.82_{\pm0.20}$ | $99.68_{\pm0.23}$ | $96.95_{\pm0.51}$ | $87.26_{\pm1.08}$ | 97.16 |
| | **OpenAI-STK** | $99.99_{\pm0.01}$ | $\underline{99.83}\pm0.07$ | $\mathbf{100.00}_{\pm0.00}$ | $99.83_{\pm0.12}$ | $98.00_{\pm0.37}$ | $91.69_{\pm1.10}$ | 98.22 |
| | MPU | $\mathbf{100.00}_{\pm0.00}$ | $99.81_{\pm0.08}$ | $99.99_{\pm0.02}$ | $\mathbf{99.99}_{\pm0.01}$ | $97.55_{\pm0.63}$ | $98.69_{\pm0.35}$ | 99.34 |
| | **MPU-STK** | $\mathbf{100.00}_{\pm0.00}$ | $99.80_{\pm0.14}$ | $\mathbf{100.00}_{\pm0.00}$ | $99.98_{\pm0.01}$ | $\underline{98.02}_{\pm0.58}$ | $\underline{99.36}_{\pm0.36}$ | 99.53 |
| | RADAR | $\mathbf{100.00}_{\pm0.00}$ | $\mathbf{100.00}_{\pm0.00}$ | $\underline{99.99}_{\pm0.01}$ | $99.99_{\pm0.02}$ | $\mathbf{99.60}_{\pm0.20}$ | $\mathbf{99.82}_{\pm0.11}$ | **99.90** |
| | **RADAR-STK** | $\mathbf{100.00}_{\pm0.00}$ | $\mathbf{100.00}_{\pm0.00}$ | $\underline{99.99}_{\pm0.01}$ | $\mathbf{99.99}_{\pm0.01}$ | $\mathbf{99.60}_{\pm0.20}$ | $\mathbf{99.82}_{\pm0.11}$ | **99.90** |
| SQuAD1 | Log-Likelihood | $69.62_{\pm0.28}$ | $69.61_{\pm1.06}$ | $67.18_{\pm0.67}$ | $65.67_{\pm0.56}$ | $67.24_{\pm1.10}$ | $85.82_{\pm1.96}$ | 70.86 |
| | Rank | $63.45_{\pm0.91}$ | $64.71_{\pm0.99}$ | $62.30_{\pm0.77}$ | $61.56_{\pm0.80}$ | $64.16_{\pm0.86}$ | $73.52_{\pm1.14}$ | 64.95 |
| | Log-Rank | $68.78_{\pm0.34}$ | $69.52_{\pm0.79}$ | $66.33_{\pm0.59}$ | $64.99_{\pm0.65}$ | $66.83_{\pm1.04}$ | $85.50_{\pm1.97}$ | 70.33 |
| | DetectGPT | $52.60_{\pm2.74}$ | $47.47_{\pm1.42}$ | $52.05_{\pm1.87}$ | $50.88_{\pm2.47}$ | $50.24_{\pm2.20}$ | $51.12_{\pm2.44}$ | 50.73 |
| | F-DetectGPT | $67.91_{\pm0.23}$ | $63.84_{\pm1.73}$ | $64.67_{\pm0.20}$ | $63.36_{\pm0.42}$ | $66.32_{\pm1.36}$ | $79.62_{\pm0.63}$ | 67.62 |
| | ChatGPT-D | $56.27_{\pm1.38}$ | $43.03_{\pm2.98}$ | $54.69_{\pm1.29}$ | $53.76_{\pm1.57}$ | $54.41_{\pm1.32}$ | $63.75_{\pm2.58}$ | 54.32 |
| | **ChatGPT-STK** | $56.99_{\pm0.85}$ | $58.55_{\pm1.43}$ | $55.14_{\pm1.48}$ | $55.03_{\pm1.41}$ | $55.28_{\pm0.59}$ | $64.33_{\pm1.40}$ | 57.55 |
| | OpenAI-D | $\underline{99.19}_{\pm0.28}$ | $\mathbf{99.34}_{\pm0.27}$ | $98.63_{\pm0.75}$ | $\mathbf{94.95}_{\pm1.21}$ | $\underline{97.74}_{\pm0.28}$ | $\underline{99.53}_{\pm0.22}$ | 98.23 |
| | **OpenAI-STK** | $\mathbf{99.30}_{\pm0.28}$ | $\underline{99.31}_{\pm0.28}$ | $\underline{98.73}_{\pm0.75}$ | $\underline{94.90}_{\pm0.98}$ | $\mathbf{97.84}_{\pm0.39}$ | $\mathbf{99.62}_{\pm0.15}$ | **98.28** |
| | MPU | $98.61_{\pm0.54}$ | $98.79_{\pm0.71}$ | $98.71_{\pm0.83}$ | $93.53_{\pm1.36}$ | $96.45_{\pm0.94}$ | $98.85_{\pm0.48}$ | 97.49 |
| | **MPU-STK** | $98.70_{\pm0.35}$ | $99.21_{\pm0.59}$ | $\mathbf{98.74}_{\pm0.73}$ | $93.71_{\pm1.47}$ | $96.51_{\pm0.77}$ | $98.84_{\pm0.45}$ | 97.62 |
| | RADAR | $82.07_{\pm1.92}$ | $80.63_{\pm1.26}$ | $81.19_{\pm1.82}$ | $78.14_{\pm1.21}$ | $77.76_{\pm2.07}$ | $81.53_{\pm2.07}$ | 80.22 |
| | **RADAR-STK** | $83.45_{\pm1.85}$ | $81.46_{\pm1.00}$ | $83.17_{\pm0.73}$ | $79.05_{\pm0.96}$ | $79.00_{\pm1.69}$ | $82.37_{\pm2.33}$ | 81.42 |

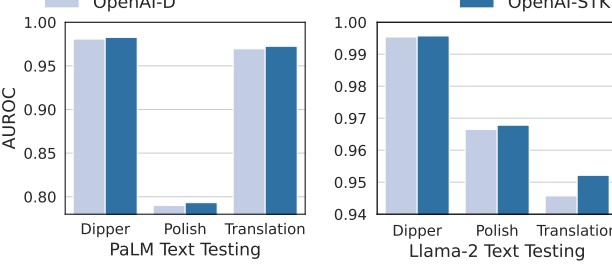

Figure 15: Enhance the robustness of OpenAI-D. Here we use three attacks: Dipper, Polish, and Translation.

## H.11 COMPARISON WITH SOFT STACKED DETECTION ENHANCEMENT FRAMEWORK

This section compares two EM-inspired detection enhancement frameworks to highlight the rationality of the hard EM-based approach. Due to the significant memory and computational costs of the soft version, we only experimented on the SQuAD1 dataset with fewer sequence numbers, setting the sequence number to 2. The results are shown in Fig. 18, where variants with the "soft" suffix represent the soft version. As can be seen, the overall performance of the soft version is inferior

Table 4: Performance concerning AUROC. Detectors are trained on GPT-4 texts.

| Dataset | Method | ChatGPT | GPT-4 | GPT-turbo | ChatGLM | Dolly | Claude/StableLM | Avg. |
|---|---|---|---|---|---|---|---|---|
| Essay | Log-Likelihood | $98.48_{\pm 0.43}$ | $96.46_{\pm 1.05}$ | $98.85_{\pm 0.42}$ | $98.78_{\pm 0.15}$ | $90.64_{\pm 0.35}$ | $92.16_{\pm 0.32}$ | 95.90 |
| | Rank | $92.84_{\pm 0.99}$ | $90.93_{\pm 0.74}$ | $98.19_{\pm 0.37}$ | $81.51_{\pm 1.32}$ | $80.45_{\pm 1.41}$ | $86.69_{\pm 1.06}$ | 88.44 |
| | Log-Rank | $98.64_{\pm 0.43}$ | $96.85_{\pm 0.89}$ | $98.92_{\pm 0.40}$ | $98.94_{\pm 0.16}$ | $90.41_{\pm 0.50}$ | $91.47_{\pm 0.45}$ | 95.87 |
| | DetectGPT | $96.86_{\pm 0.97}$ | $95.64_{\pm 0.45}$ | $44.40_{\pm 1.73}$ | $95.60_{\pm 0.62}$ | $92.55_{\pm 0.58}$ | $46.36_{\pm 0.71}$ | 78.57 |
| | F-DetectGPT | $83.86_{\pm 1.52}$ | $84.45_{\pm 1.58}$ | $94.12_{\pm 0.72}$ | $93.62_{\pm 0.73}$ | $55.86_{\pm 2.54}$ | $58.96_{\pm 1.52}$ | 78.48 |
| | ChatGPT-D | $99.38_{\pm 0.45}$ | $99.43_{\pm 0.65}$ | $91.96_{\pm 3.77}$ | $99.96_{\pm 0.03}$ | $94.81_{\pm 2.35}$ | $68.44_{\pm 12.85}$ | 92.33 |
| | **ChatGPT-STK** | $99.66_{\pm 0.12}$ | $99.66_{\pm 0.06}$ | $93.04_{\pm 1.65}$ | $99.92_{\pm 0.07}$ | $94.63_{\pm 1.05}$ | $72.64_{\pm 2.02}$ | 93.26 |
| | OpenAI-D | $99.72_{\pm 0.34}$ | $99.66_{\pm 0.26}$ | $98.51_{\pm 0.73}$ | $99.89_{\pm 0.14}$ | $98.39_{\pm 0.83}$ | $81.29_{\pm 4.23}$ | 96.24 |
| | **OpenAI-STK** | $99.84_{\pm 0.22}$ | $99.81_{\pm 0.16}$ | $99.15_{\pm 0.58}$ | $99.95_{\pm 0.04}$ | $98.44_{\pm 1.14}$ | $86.54_{\pm 5.25}$ | 97.29 |
| | MPU | $\underline{99.99}_{\pm 0.01}$ | $\mathbf{99.95}_{\pm 0.03}$ | $98.77_{\pm 0.34}$ | $\mathbf{99.99}_{\pm 0.01}$ | $98.02_{\pm 0.37}$ | $97.09_{\pm 0.60}$ | 98.97 |
| | **MPU-STK** | $\mathbf{99.99}_{\pm 0.01}$ | $99.91_{\pm 0.09}$ | $98.83_{\pm 0.21}$ | $\mathbf{99.99}_{\pm 0.01}$ | $98.15_{\pm 0.29}$ | $97.92_{\pm 0.57}$ | 99.13 |
| | RADAR | $99.92_{\pm 0.02}$ | $99.89_{\pm 0.05}$ | $\underline{99.89}_{\pm 0.10}$ | $99.98_{\pm 0.01}$ | $98.66_{\pm 0.24}$ | $98.98_{\pm 0.26}$ | 99.55 |
| | **RADAR-STK** | $99.93_{\pm 0.01}$ | $\underline{99.91}_{\pm 0.04}$ | $\mathbf{99.90}_{\pm 0.08}$ | $\underline{99.98}_{\pm 0.01}$ | $98.99_{\pm 0.23}$ | $99.09_{\pm 0.23}$ | **99.63** |
| Reuters | Log-Likelihood | $97.59_{\pm 0.37}$ | $74.85_{\pm 0.50}$ | $98.54_{\pm 0.38}$ | $99.54_{\pm 0.22}$ | $60.09_{\pm 2.13}$ | $85.66_{\pm 1.04}$ | 86.05 |
| | Rank | $84.00_{\pm 0.88}$ | $74.12_{\pm 1.24}$ | $91.82_{\pm 0.60}$ | $70.97_{\pm 1.65}$ | $54.88_{\pm 2.45}$ | $67.78_{\pm 1.14}$ | 73.93 |
| | Log-Rank | $97.86_{\pm 0.35}$ | $79.95_{\pm 0.29}$ | $98.82_{\pm 0.37}$ | $99.65_{\pm 0.20}$ | $61.24_{\pm 2.15}$ | $84.88_{\pm 1.13}$ | 87.07 |
| | DetectGPT | $92.78_{\pm 1.34}$ | $85.71_{\pm 2.02}$ | $49.24_{\pm 1.65}$ | $91.59_{\pm 1.67}$ | $83.41_{\pm 1.73}$ | $66.59_{\pm 2.53}$ | 78.22 |
| | F-DetectGPT | $96.23_{\pm 0.19}$ | $66.02_{\pm 1.53}$ | $98.85_{\pm 0.30}$ | $98.58_{\pm 0.35}$ | $40.38_{\pm 1.31}$ | $66.16_{\pm 2.58}$ | 77.70 |
| | ChatGPT-D | $99.59_{\pm 0.53}$ | $99.29_{\pm 0.78}$ | $99.37_{\pm 0.57}$ | $99.07_{\pm 0.76}$ | $90.45_{\pm 3.16}$ | $67.02_{\pm 7.87}$ | 92.47 |
| | **ChatGPT-STK** | $99.73_{\pm 0.17}$ | $99.56_{\pm 0.36}$ | $99.39_{\pm 0.51}$ | $99.65_{\pm 0.24}$ | $92.08_{\pm 0.68}$ | $74.16_{\pm 7.11}$ | 94.10 |
| | OpenAI-D | $99.65_{\pm 0.44}$ | $99.63_{\pm 0.44}$ | $99.63_{\pm 0.42}$ | $99.58_{\pm 0.48}$ | $98.51_{\pm 0.78}$ | $80.91_{\pm 1.35}$ | 96.32 |
| | **OpenAI-STK** | $\underline{99.76}_{\pm 0.25}$ | $99.82_{\pm 0.20}$ | $99.78_{\pm 0.25}$ | $99.75_{\pm 0.24}$ | $99.09_{\pm 0.43}$ | $83.85_{\pm 2.12}$ | 97.01 |
| | MPU | $\mathbf{100.00}_{\pm 0.00}$ | $\underline{99.94}_{\pm 0.05}$ | $\mathbf{100.00}_{\pm 0.00}$ | $99.99_{\pm 0.01}$ | $\underline{99.48}_{\pm 0.33}$ | $97.62_{\pm 0.97}$ | 99.51 |
| | **MPU-STK** | $\mathbf{100.00}_{\pm 0.00}$ | $99.90_{\pm 0.09}$ | $\mathbf{100.00}_{\pm 0.00}$ | $99.99_{\pm 0.01}$ | $99.39_{\pm 0.27}$ | $\underline{98.13}_{\pm 0.77}$ | $\underline{99.57}$ |
| | RADAR | $\mathbf{100.00}_{\pm 0.00}$ | $\mathbf{100.00}_{\pm 0.00}$ | $99.96_{\pm 0.02}$ | $\mathbf{100.00}_{\pm 0.01}$ | $99.53_{\pm 0.16}$ | $\mathbf{99.66}_{\pm 0.14}$ | **99.86** |
| | **RADAR-STK** | $\mathbf{100.00}_{\pm 0.00}$ | $\mathbf{100.00}_{\pm 0.00}$ | $\underline{99.96}_{\pm 0.02}$ | $\mathbf{100.00}_{\pm 0.01}$ | $99.53_{\pm 0.16}$ | $\mathbf{99.66}_{\pm 0.14}$ | **99.86** |
| SQuAD1 | Log-Likelihood | $69.62_{\pm 0.28}$ | $69.61_{\pm 1.06}$ | $67.18_{\pm 0.67}$ | $65.67_{\pm 0.56}$ | $67.24_{\pm 1.10}$ | $85.82_{\pm 1.96}$ | 70.86 |
| | Rank | $63.45_{\pm 0.91}$ | $64.71_{\pm 0.99}$ | $62.30_{\pm 0.77}$ | $61.56_{\pm 0.80}$ | $64.16_{\pm 0.86}$ | $73.52_{\pm 1.14}$ | 64.95 |
| | Log-Rank | $68.78_{\pm 0.34}$ | $69.52_{\pm 0.79}$ | $66.33_{\pm 0.59}$ | $64.99_{\pm 0.65}$ | $66.83_{\pm 1.04}$ | $85.50_{\pm 1.97}$ | 70.33 |
| | DetectGPT | $49.18_{\pm 3.68}$ | $52.08_{\pm 2.03}$ | $48.92_{\pm 2.56}$ | $50.32_{\pm 2.60}$ | $48.93_{\pm 1.94}$ | $49.87_{\pm 2.69}$ | 49.88 |
| | F-DetectGPT | $67.91_{\pm 0.23}$ | $63.84_{\pm 1.73}$ | $64.67_{\pm 0.20}$ | $63.36_{\pm 0.42}$ | $66.32_{\pm 1.36}$ | $79.62_{\pm 0.63}$ | 67.62 |
| | ChatGPT-D | $57.52_{\pm 0.82}$ | $61.68_{\pm 3.26}$ | $55.34_{\pm 0.48}$ | $54.90_{\pm 0.69}$ | $55.55_{\pm 1.19}$ | $66.07_{\pm 2.47}$ | 58.51 |
| | **ChatGPT-STK** | $57.38_{\pm 0.50}$ | $59.11_{\pm 0.81}$ | $55.74_{\pm 0.67}$ | $55.09_{\pm 0.41}$ | $55.50_{\pm 0.82}$ | $66.01_{\pm 1.32}$ | 58.14 |
| | OpenAI-D | $91.19_{\pm 1.69}$ | $\mathbf{99.72}_{\pm 0.16}$ | $89.44_{\pm 1.71}$ | $86.38_{\pm 1.76}$ | $88.06_{\pm 1.10}$ | $96.31_{\pm 1.27}$ | 91.85 |
| | **OpenAI-STK** | $\mathbf{99.45}_{\pm 0.23}$ | $99.05_{\pm 0.46}$ | $\underline{98.85}_{\pm 0.73}$ | $\mathbf{95.83}_{\pm 0.83}$ | $98.09_{\pm 0.25}$ | $\mathbf{99.71}_{\pm 0.17}$ | **98.50** |
| | MPU | $73.14_{\pm 5.54}$ | $\underline{99.11}_{\pm 0.50}$ | $69.43_{\pm 6.64}$ | $68.98_{\pm 5.69}$ | $71.34_{\pm 5.03}$ | $87.73_{\pm 2.94}$ | 78.29 |
| | **MPU-STK** | $\underline{99.11}_{\pm 0.19}$ | $98.96_{\pm 0.74}$ | $\mathbf{98.91}_{\pm 0.56}$ | $\underline{94.88}_{\pm 1.20}$ | $97.53_{\pm 0.47}$ | $\underline{99.10}_{\pm 0.23}$ | $\underline{98.08}$ |
| | RADAR | $82.32_{\pm 2.14}$ | $80.70_{\pm 0.82}$ | $81.86_{\pm 1.18}$ | $78.54_{\pm 0.60}$ | $78.42_{\pm 2.00}$ | $82.04_{\pm 2.07}$ | 80.65 |
| | **RADAR-STK** | $83.62_{\pm 1.57}$ | $82.09_{\pm 1.48}$ | $83.29_{\pm 1.39}$ | $79.50_{\pm 0.99}$ | $80.05_{\pm 1.83}$ | $82.79_{\pm 1.96}$ | 81.89 |

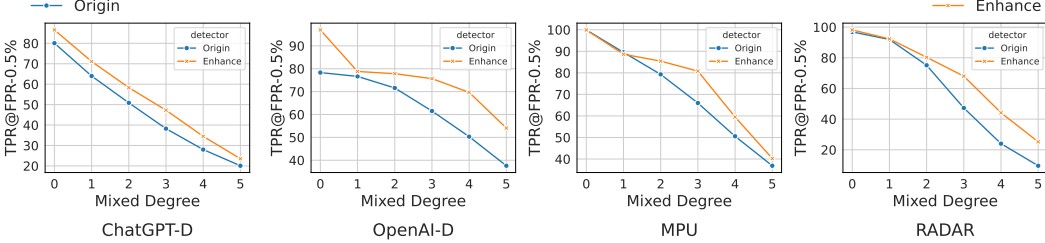

Figure 16: Performance concerning TPR@FPR-5% at different mixing levels. These detectors are trained on ChatGPT texts.

to the hard version, which aligns with many existing findings (Samdani et al., 2012; Parker & Yu, 2021; Wen et al., 2023) that the soft EM algorithm tends to allocate too much probability mass to the tail, wasting probability mass on unimportant hidden variables, thereby resulting in poorer performance. Additionally, the soft version is competitive on the poorer-performing ChatGPT-D but underperforms on the stronger detectors (OpenAI-D and MPU). This observation is consistent with existing findings (Spitkovsky et al., 2010) that for better parameter models, EM tends to drift further than hard EM (thus losing accuracy). In summary, the experimental results demonstrate the rationality of optimization based on the hard EM algorithm.

Table 5: Performance concerning FPR@TPR-0.5%. Detectors are trained on GPT-4 texts.

| Dataset | Method | ChatGPT | GPT-4 | GPT-turbo | ChatGLM | Dolly | Claude/StableLM | Avg. |
|---|---|---|---|---|---|---|---|---|
| Essay | Log-Likelihood | $24.08_{\pm22.67}$ | $37.70_{\pm30.63}$ | $23.12_{\pm24.38}$ | $5.86_{\pm7.02}$ | $12.45_{\pm12.04}$ | $2.48_{\pm3.06}$ | 17.62 |
| | Rank | $55.60_{\pm3.65}$ | $51.31_{\pm5.91}$ | $65.84_{\pm6.93}$ | $53.49_{\pm6.09}$ | $35.11_{\pm4.81}$ | $25.12_{\pm5.07}$ | 47.75 |
| | Log-Rank | $28.72_{\pm30.45}$ | $46.48_{\pm36.20}$ | $25.04_{\pm26.83}$ | $28.19_{\pm27.90}$ | $17.34_{\pm15.65}$ | $2.96_{\pm3.73}$ | 24.79 |
| | DetectGPT | $37.04_{\pm10.21}$ | $24.75_{\pm10.03}$ | $5.52_{\pm1.93}$ | $21.45_{\pm14.25}$ | $15.45_{\pm7.05}$ | $4.48_{\pm2.57}$ | 18.12 |
| | F-DetectGPT | $4.24_{\pm1.51}$ | $3.85_{\pm2.05}$ | $31.28_{\pm3.18}$ | $35.74_{\pm3.97}$ | $0.00_{\pm0.00}$ | $0.16_{\pm0.20}$ | 12.55 |
| | ChatGPT-D | $89.92_{\pm5.69}$ | $89.75_{\pm7.23}$ | $43.52_{\pm19.38}$ | $99.04_{\pm0.70}$ | $55.54_{\pm10.95}$ | $8.24_{\pm5.98}$ | 64.33 |
| | **ChatGPT-STK** | $90.88_{\pm1.39}$ | $93.61_{\pm1.77}$ | $48.80_{\pm3.50}$ | $99.04_{\pm1.03}$ | $56.65_{\pm3.69}$ | $8.40_{\pm1.29}$ | 66.23 |
| | OpenAI-D | $82.08_{\pm30.45}$ | $86.56_{\pm12.64}$ | $71.04_{\pm27.40}$ | $97.75_{\pm2.94}$ | $70.64_{\pm26.78}$ | $18.48_{\pm12.66}$ | 71.09 |
| | **OpenAI-STK** | $95.68_{\pm5.07}$ | $95.16_{\pm3.43}$ | $82.48_{\pm16.97}$ | $97.75_{\pm1.86}$ | $78.28_{\pm14.75}$ | $30.80_{\pm16.05}$ | 80.03 |
| | MPU | $\underline{99.44}_{\pm0.32}$ | $\mathbf{99.51}_{\pm0.40}$ | $73.04_{\pm6.10}$ | $\underline{99.44}_{\pm0.41}$ | $78.80_{\pm2.72}$ | $55.36_{\pm5.44}$ | 84.26 |
| | **MPU-STK** | $\mathbf{99.52}_{\pm0.39}$ | $\underline{99.43}_{\pm0.56}$ | $74.08_{\pm6.42}$ | $\mathbf{99.52}_{\pm0.30}$ | $79.66_{\pm4.67}$ | $67.36_{\pm12.94}$ | 86.59 |
| | RADAR | $97.28_{\pm1.32}$ | $95.98_{\pm2.54}$ | $\underline{96.80}_{\pm2.01}$ | $99.20_{\pm0.51}$ | $\underline{81.37}_{\pm7.64}$ | $\underline{77.44}_{\pm9.59}$ | $\underline{91.35}$ |
| | **RADAR-STK** | $97.92_{\pm0.69}$ | $96.23_{\pm2.62}$ | $\mathbf{97.36}_{\pm1.30}$ | $99.36_{\pm0.32}$ | $\mathbf{82.49}_{\pm6.18}$ | $\mathbf{78.80}_{\pm8.18}$ | $\mathbf{92.03}$ |
| Reuters | Log-Likelihood | $77.84_{\pm5.19}$ | $14.88_{\pm5.98}$ | $86.08_{\pm3.38}$ | $93.76_{\pm2.03}$ | $11.20_{\pm4.45}$ | $15.04_{\pm6.86}$ | 49.80 |
| | Rank | $48.88_{\pm1.59}$ | $35.92_{\pm2.88}$ | $58.40_{\pm3.94}$ | $40.56_{\pm1.85}$ | $18.56_{\pm2.27}$ | $6.24_{\pm1.87}$ | 34.76 |
| | Log-Rank | $82.40_{\pm5.24}$ | $25.92_{\pm7.08}$ | $90.96_{\pm4.12}$ | $96.80_{\pm0.88}$ | $14.00_{\pm4.82}$ | $17.60_{\pm8.29}$ | 54.61 |
| | DetectGPT | $4.40_{\pm2.62}$ | $0.64_{\pm0.54}$ | $2.32_{\pm1.87}$ | $2.56_{\pm2.80}$ | $0.48_{\pm0.47}$ | $3.04_{\pm1.61}$ | 2.24 |
| | F-DetectGPT | $48.00_{\pm9.48}$ | $6.80_{\pm1.88}$ | $92.96_{\pm1.65}$ | $88.96_{\pm4.80}$ | $0.00_{\pm0.00}$ | $0.48_{\pm0.39}$ | 39.53 |
| | ChatGPT-D | $85.76_{\pm23.50}$ | $82.08_{\pm30.65}$ | $81.92_{\pm29.00}$ | $83.04_{\pm31.12}$ | $54.96_{\pm20.38}$ | $9.28_{\pm6.55}$ | 66.17 |
| | **ChatGPT-STK** | $94.48_{\pm3.66}$ | $93.60_{\pm6.28}$ | $90.00_{\pm5.58}$ | $97.92_{\pm1.11}$ | $56.08_{\pm14.30}$ | $8.88_{\pm8.47}$ | 73.49 |
| | OpenAI-D | $79.84_{\pm38.52}$ | $83.84_{\pm27.78}$ | $79.68_{\pm37.85}$ | $79.36_{\pm38.68}$ | $61.92_{\pm23.84}$ | $16.96_{\pm7.37}$ | 66.93 |
| | **OpenAI-STK** | $86.48_{\pm25.04}$ | $94.00_{\pm9.82}$ | $87.04_{\pm24.12}$ | $96.08_{\pm5.45}$ | $72.24_{\pm13.89}$ | $26.16_{\pm6.47}$ | 77.00 |
| | MPU | $\mathbf{100.00}_{\pm0.00}$ | $\underline{99.12}_{\pm0.64}$ | $\underline{99.92}_{\pm0.16}$ | $99.84_{\pm0.20}$ | $\underline{88.40}_{\pm6.73}$ | $67.44_{\pm16.46}$ | 92.45 |
| | **MPU-STK** | $\mathbf{100.00}_{\pm0.00}$ | $\underline{99.12}_{\pm0.64}$ | $\mathbf{100.00}_{\pm0.00}$ | $99.68_{\pm0.30}$ | $86.64_{\pm4.03}$ | $65.28_{\pm13.06}$ | 91.79 |
| | RADAR | $\underline{99.92}_{\pm0.16}$ | $\mathbf{100.00}_{\pm0.00}$ | $99.04_{\pm1.12}$ | $\mathbf{99.92}_{\pm0.16}$ | $\mathbf{92.16}_{\pm2.66}$ | $\mathbf{90.96}_{\pm3.28}$ | $\mathbf{97.00}$ |
| | **RADAR-STK** | $\underline{99.92}_{\pm0.16}$ | $\mathbf{100.00}_{\pm0.00}$ | $99.04_{\pm1.12}$ | $\mathbf{99.92}_{\pm0.16}$ | $\mathbf{92.16}_{\pm2.66}$ | $\underline{90.88}_{\pm3.33}$ | $\underline{96.99}$ |
| SQuAD1 | Log-Likelihood | $1.04_{\pm0.57}$ | $5.00_{\pm2.93}$ | $2.11_{\pm0.29}$ | $1.85_{\pm1.12}$ | $5.93_{\pm3.44}$ | $21.85_{\pm10.14}$ | 6.30 |
| | Rank | $2.43_{\pm0.43}$ | $3.81_{\pm1.04}$ | $1.87_{\pm0.68}$ | $1.97_{\pm0.28}$ | $4.30_{\pm1.01}$ | $7.98_{\pm1.52}$ | 3.73 |
| | Log-Rank | $1.97_{\pm1.19}$ | $4.88_{\pm3.25}$ | $2.34_{\pm0.83}$ | $1.97_{\pm1.35}$ | $4.88_{\pm2.72}$ | $19.77_{\pm11.36}$ | 5.97 |
| | DetectGPT | $0.58_{\pm0.52}$ | $1.31_{\pm1.02}$ | $0.12_{\pm0.23}$ | $0.58_{\pm0.63}$ | $0.58_{\pm0.64}$ | $0.46_{\pm0.43}$ | 0.60 |
| | F-DetectGPT | $4.28_{\pm1.35}$ | $6.90_{\pm1.53}$ | $2.69_{\pm1.41}$ | $2.77_{\pm1.12}$ | $5.35_{\pm2.19}$ | $28.44_{\pm3.93}$ | 8.41 |
| | ChatGPT-D | $1.27_{\pm0.92}$ | $4.40_{\pm1.17}$ | $0.94_{\pm0.70}$ | $0.69_{\pm0.23}$ | $1.63_{\pm0.85}$ | $5.09_{\pm1.34}$ | 2.34 |
| | **ChatGPT-STK** | $1.04_{\pm0.67}$ | $4.05_{\pm1.38}$ | $1.17_{\pm0.52}$ | $1.04_{\pm0.57}$ | $1.86_{\pm1.19}$ | $7.75_{\pm4.24}$ | 2.82 |
| | OpenAI-D | $9.60_{\pm6.35}$ | $\underline{88.93}_{\pm13.08}$ | $10.88_{\pm8.84}$ | $7.63_{\pm4.41}$ | $20.58_{\pm7.94}$ | $52.83_{\pm23.86}$ | 31.74 |
| | **OpenAI-STK** | $\underline{49.02}_{\pm11.18}$ | $38.57_{\pm21.65}$ | $\underline{46.08}_{\pm16.91}$ | $\underline{35.26}_{\pm8.20}$ | $69.53_{\pm16.99}$ | $\mathbf{83.58}_{\pm7.79}$ | $\underline{53.67}$ |
| | MPU | $5.20_{\pm6.26}$ | $\mathbf{92.14}_{\pm9.53}$ | $2.34_{\pm1.57}$ | $4.51_{\pm5.26}$ | $9.77_{\pm2.79}$ | $27.05_{\pm13.30}$ | 23.50 |
| | **MPU-STK** | $\mathbf{54.22}_{\pm18.76}$ | $46.31_{\pm31.59}$ | $\mathbf{54.85}_{\pm15.39}$ | $\mathbf{43.70}_{\pm14.10}$ | $\underline{69.30}_{\pm22.98}$ | $\underline{58.73}_{\pm25.58}$ | $\mathbf{54.52}$ |
| | RADAR | $35.61_{\pm9.70}$ | $29.88_{\pm10.88}$ | $38.48_{\pm3.40}$ | $28.09_{\pm8.28}$ | $33.14_{\pm3.23}$ | $33.87_{\pm8.37}$ | 33.18 |
| | **RADAR-STK** | $39.77_{\pm6.43}$ | $31.55_{\pm11.45}$ | $42.22_{\pm2.32}$ | $31.79_{\pm5.32}$ | $36.28_{\pm2.46}$ | $36.30_{\pm7.33}$ | 36.32 |

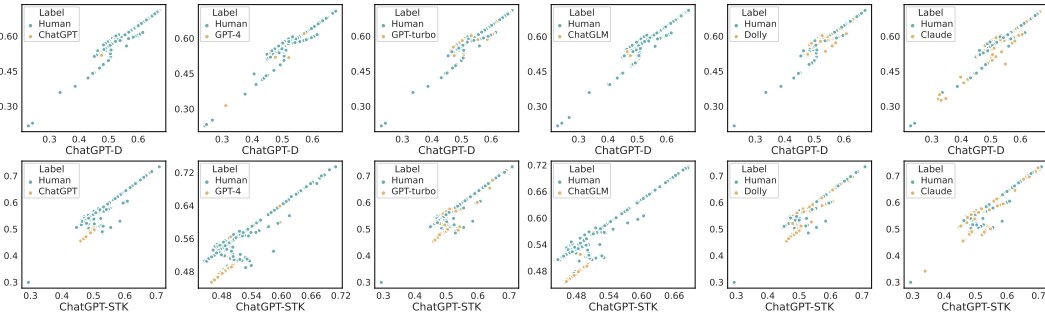

Figure 17: The t-SNE visualization of the last hidden state of ChatGPT-D (first line) and ChatGPT-STK (second line) when detecting different LLM texts (different columns). These detectors are trained on ChatGPT texts.

## H.12 ENHANCEMENTS TO SENTENCE-BASED DETECTOR

In this section, we apply the proposed strategy to the sentence-based detector SeqXGPT (Wang et al., 2023), and the results are shown in Table 18. First, the proposed framework significantly boosts SeqXGPT's detection performance, demonstrating the flexibility and broad effectiveness of our method. Second, while SeqXGPT outperforms feature-based approaches (Table 3), it is less

Table 6: Performance concerning AUROC. Detectors are trained on GPT-turbo texts.

| Dataset | Method | ChatGPT | GPT-4 | GPT-turbo | ChatGLM | Dolly | Claude/StableLM | Avg. |
|---|---|---|---|---|---|---|---|---|
| Essay | Log-Likelihood | $98.48_{\pm 0.43}$ | $96.46_{\pm 1.05}$ | $98.85_{\pm 0.42}$ | $98.78_{\pm 0.15}$ | $90.64_{\pm 0.35}$ | $92.16_{\pm 0.32}$ | 95.90 |
| | Rank | $92.84_{\pm 0.99}$ | $90.93_{\pm 0.74}$ | $98.19_{\pm 0.37}$ | $81.51_{\pm 1.32}$ | $80.45_{\pm 1.41}$ | $86.69_{\pm 1.06}$ | 88.44 |
| | Log-Rank | $98.64_{\pm 0.43}$ | $96.85_{\pm 0.89}$ | $98.92_{\pm 0.40}$ | $98.94_{\pm 0.16}$ | $90.41_{\pm 0.50}$ | $91.47_{\pm 0.45}$ | 95.87 |
| | DetectGPT | $96.86_{\pm 0.97}$ | $95.64_{\pm 0.45}$ | $44.40_{\pm 1.73}$ | $95.60_{\pm 0.62}$ | $92.55_{\pm 0.58}$ | $46.36_{\pm 0.71}$ | 78.57 |
| | F-DetectGPT | $83.86_{\pm 1.52}$ | $84.45_{\pm 1.58}$ | $94.12_{\pm 0.72}$ | $93.62_{\pm 0.73}$ | $55.86_{\pm 2.54}$ | $58.96_{\pm 1.52}$ | 78.48 |
| | ChatGPT-D | $97.11_{\pm 0.39}$ | $96.26_{\pm 2.23}$ | $96.66_{\pm 1.76}$ | $99.62_{\pm 0.15}$ | $80.88_{\pm 7.54}$ | $66.73_{\pm 17.76}$ | 89.54 |
| | **ChatGPT-STK** | $96.86_{\pm 1.34}$ | $96.37_{\pm 0.90}$ | $96.53_{\pm 1.33}$ | $99.48_{\pm 0.31}$ | $83.37_{\pm 5.26}$ | $67.94_{\pm 15.24}$ | 90.09 |
| | OpenAI-D | $99.77_{\pm 0.31}$ | $\mathbf{99.87}_{\pm 0.08}$ | $99.83_{\pm 0.32}$ | $99.88_{\pm 0.16}$ | $\mathbf{98.51}_{\pm 0.81}$ | $97.62_{\pm 0.92}$ | 99.25 |
| | **OpenAI-STK** | $99.73_{\pm 0.15}$ | $99.75_{\pm 0.14}$ | $99.91_{\pm 0.15}$ | $99.89_{\pm 0.11}$ | $\underline{97.93}_{\pm 0.66}$ | $97.82_{\pm 0.85}$ | 99.17 |
| | MPU | $99.86_{\pm 0.05}$ | $99.79_{\pm 0.10}$ | $\mathbf{99.97}_{\pm 0.02}$ | $99.93_{\pm 0.04}$ | $94.38_{\pm 0.53}$ | $\mathbf{98.55}_{\pm 0.19}$ | 98.75 |
| | **MPU-STK** | $99.85_{\pm 0.04}$ | $99.74_{\pm 0.13}$ | $99.96_{\pm 0.03}$ | $99.91_{\pm 0.05}$ | $93.57_{\pm 0.80}$ | $98.03_{\pm 0.33}$ | 98.51 |
| | RADAR | $\mathbf{99.88}_{\pm 0.05}$ | $\underline{99.82}_{\pm 0.05}$ | $\mathbf{99.97}_{\pm 0.02}$ | $\mathbf{99.98}_{\pm 0.00}$ | $97.03_{\pm 0.64}$ | $\underline{98.13}_{\pm 0.26}$ | 99.14 |
| | **RADAR-STK** | $\underline{99.87}_{\pm 0.07}$ | $99.81_{\pm 0.06}$ | $\mathbf{99.97}_{\pm 0.02}$ | $\mathbf{99.98}_{\pm 0.00}$ | $97.15_{\pm 0.95}$ | $98.13_{\pm 0.65}$ | **99.15** |
| Reuters | Log-Likelihood | $97.59_{\pm 0.37}$ | $74.85_{\pm 0.50}$ | $98.54_{\pm 0.38}$ | $99.54_{\pm 0.22}$ | $60.09_{\pm 2.13}$ | $85.66_{\pm 1.04}$ | 86.05 |
| | Rank | $84.00_{\pm 0.88}$ | $74.12_{\pm 1.24}$ | $91.82_{\pm 0.60}$ | $70.97_{\pm 1.65}$ | $54.88_{\pm 2.45}$ | $67.78_{\pm 1.14}$ | 73.93 |
| | Log-Rank | $97.86_{\pm 0.35}$ | $79.95_{\pm 0.29}$ | $98.82_{\pm 0.37}$ | $99.65_{\pm 0.20}$ | $61.24_{\pm 2.15}$ | $84.88_{\pm 1.13}$ | 87.07 |
| | DetectGPT | $74.86_{\pm 34.84}$ | $69.99_{\pm 29.66}$ | $48.45_{\pm 0.95}$ | $74.21_{\pm 33.87}$ | $68.96_{\pm 27.57}$ | $58.45_{\pm 14.49}$ | 65.82 |
| | F-DetectGPT | $96.23_{\pm 0.19}$ | $66.02_{\pm 1.53}$ | $98.85_{\pm 0.30}$ | $98.58_{\pm 0.35}$ | $40.38_{\pm 1.31}$ | $66.16_{\pm 2.58}$ | 77.70 |
| | ChatGPT-D | $99.12_{\pm 0.82}$ | $98.01_{\pm 0.98}$ | $99.52_{\pm 0.48}$ | $99.02_{\pm 0.50}$ | $81.10_{\pm 6.36}$ | $70.01_{\pm 19.60}$ | 91.13 |
| | **ChatGPT-STK** | $99.19_{\pm 0.78}$ | $98.11_{\pm 0.72}$ | $99.65_{\pm 0.42}$ | $99.11_{\pm 0.47}$ | $82.17_{\pm 6.04}$ | $75.57_{\pm 17.69}$ | 92.30 |
| | OpenAI-D | $99.78_{\pm 0.37}$ | $99.49_{\pm 0.45}$ | $99.79_{\pm 0.36}$ | $99.72_{\pm 0.30}$ | $96.88_{\pm 0.80}$ | $86.58_{\pm 3.28}$ | 97.04 |
| | **OpenAI-STK** | $\underline{99.96}_{\pm 0.01}$ | $99.58_{\pm 0.11}$ | $\underline{99.99}_{\pm 0.00}$ | $99.78_{\pm 0.14}$ | $95.84_{\pm 0.91}$ | $92.77_{\pm 1.47}$ | 97.99 |
| | MPU | $\mathbf{100.00}_{\pm 0.00}$ | $\underline{99.74}_{\pm 0.17}$ | $\mathbf{100.00}_{\pm 0.00}$ | $99.97_{\pm 0.01}$ | $\underline{97.30}_{\pm 0.68}$ | $99.29_{\pm 0.29}$ | 99.38 |
| | **MPU-STK** | $\mathbf{100.00}_{\pm 0.00}$ | $99.71_{\pm 0.22}$ | $\mathbf{100.00}_{\pm 0.00}$ | $99.95_{\pm 0.03}$ | $97.11_{\pm 0.57}$ | $\underline{99.68}_{\pm 0.21}$ | 99.41 |
| | RADAR | $\mathbf{100.00}_{\pm 0.00}$ | $\mathbf{100.00}_{\pm 0.00}$ | $99.99_{\pm 0.01}$ | $99.99_{\pm 0.01}$ | $\mathbf{99.61}_{\pm 0.16}$ | $\mathbf{99.84}_{\pm 0.09}$ | $\underline{99.90}$ |
| | **RADAR-STK** | $\mathbf{100.00}_{\pm 0.00}$ | $\mathbf{100.00}_{\pm 0.00}$ | $99.99_{\pm 0.01}$ | $\underline{99.99}_{\pm 0.01}$ | $\mathbf{99.61}_{\pm 0.16}$ | $\mathbf{99.84}_{\pm 0.09}$ | **99.91** |
| SQuAD1 | Log-Likelihood | $69.62_{\pm 0.28}$ | $69.61_{\pm 1.06}$ | $67.18_{\pm 0.67}$ | $65.67_{\pm 0.56}$ | $67.24_{\pm 1.10}$ | $85.82_{\pm 1.96}$ | 70.86 |
| | Rank | $63.45_{\pm 0.91}$ | $64.71_{\pm 0.99}$ | $62.30_{\pm 0.77}$ | $61.56_{\pm 0.80}$ | $64.16_{\pm 0.86}$ | $73.52_{\pm 1.14}$ | 64.95 |
| | Log-Rank | $68.78_{\pm 0.34}$ | $69.52_{\pm 0.79}$ | $66.33_{\pm 0.59}$ | $64.99_{\pm 0.65}$ | $66.83_{\pm 1.04}$ | $85.50_{\pm 1.97}$ | 70.33 |
| | DetectGPT | $49.94_{\pm 3.77}$ | $48.22_{\pm 2.29}$ | $50.12_{\pm 2.77}$ | $51.15_{\pm 2.35}$ | $50.92_{\pm 2.02}$ | $49.83_{\pm 2.68}$ | 50.03 |
| | F-DetectGPT | $67.91_{\pm 0.23}$ | $63.84_{\pm 1.73}$ | $64.67_{\pm 0.20}$ | $63.36_{\pm 0.42}$ | $66.32_{\pm 1.36}$ | $79.62_{\pm 0.63}$ | 67.62 |
| | ChatGPT-D | $58.69_{\pm 0.60}$ | $41.91_{\pm 2.79}$ | $57.44_{\pm 0.65}$ | $55.98_{\pm 0.35}$ | $55.61_{\pm 0.43}$ | $65.41_{\pm 1.86}$ | 55.84 |
| | **ChatGPT-STK** | $57.78_{\pm 1.56}$ | $59.87_{\pm 1.17}$ | $56.56_{\pm 1.09}$ | $56.06_{\pm 1.24}$ | $56.23_{\pm 0.85}$ | $66.53_{\pm 1.93}$ | 58.84 |
| | OpenAI-D | $\mathbf{99.43}_{\pm 0.33}$ | $\underline{99.30}_{\pm 0.38}$ | $\mathbf{98.51}_{\pm 0.81}$ | $\underline{95.88}_{\pm 1.18}$ | $\mathbf{98.06}_{\pm 0.40}$ | $\underline{99.58}_{\pm 0.23}$ | $\underline{98.46}$ |
| | **OpenAI-STK** | $\underline{99.43}_{\pm 0.42}$ | $\mathbf{99.32}_{\pm 0.32}$ | $\underline{98.45}_{\pm 0.86}$ | $\mathbf{95.98}_{\pm 1.13}$ | $\underline{98.04}_{\pm 0.41}$ | $\mathbf{99.60}_{\pm 0.24}$ | **98.47** |
| | MPU | $98.59_{\pm 0.82}$ | $98.80_{\pm 0.68}$ | $98.43_{\pm 0.94}$ | $94.27_{\pm 1.94}$ | $96.61_{\pm 0.68}$ | $98.60_{\pm 0.87}$ | 97.55 |
| | **MPU-STK** | $98.63_{\pm 0.81}$ | $99.23_{\pm 0.68}$ | $98.38_{\pm 0.94}$ | $94.28_{\pm 1.92}$ | $96.53_{\pm 0.73}$ | $98.65_{\pm 0.91}$ | 97.62 |
| | RADAR | $82.20_{\pm 2.31}$ | $80.49_{\pm 1.02}$ | $81.25_{\pm 1.39}$ | $78.44_{\pm 0.74}$ | $77.80_{\pm 1.91}$ | $81.56_{\pm 2.58}$ | 80.29 |
| | **RADAR-STK** | $82.95_{\pm 2.70}$ | $80.98_{\pm 1.16}$ | $82.09_{\pm 1.42}$ | $79.03_{\pm 0.53}$ | $78.67_{\pm 2.06}$ | $81.97_{\pm 2.97}$ | 80.95 |

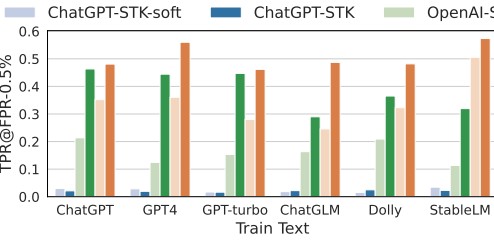
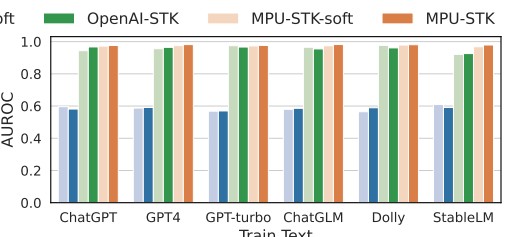

Figure 18: The performance (TPR@FPR-0.5% or AUROC) of detectors modeled and optimized by the traditional (soft) EM algorithm and the hard EM algorithm. The suffix "soft" indicates the traditional EM algorithm; otherwise, the default hard EM algorithm. The detector is trained on different LLM texts (x-axis) and reports the average performance tested on various LLM texts (six LLM texts).

competitive among model-based methods. We suspect that its direct approach of aggregating sentence labels may not be optimal for paragraph-level detection. For instance, it is likely to classify mixed text as human text, which might be inappropriate.

Table 7: Performance concerning TPR@FPR-0.5%. Detectors are trained on GPT-turbo texts.

| Dataset | Method | ChatGPT | GPT-4 | GPT-turbo | ChatGLM | Dolly | Claude/StableLM | Avg. |
|---|---|---|---|---|---|---|---|---|
| Essay | Log-Likelihood | $24.08_{\pm22.67}$ | $37.70_{\pm30.63}$ | $23.12_{\pm24.38}$ | $5.86_{\pm7.02}$ | $12.45_{\pm12.04}$ | $2.48_{\pm3.06}$ | 17.62 |
| | Rank | $55.60_{\pm3.65}$ | $51.31_{\pm5.91}$ | $65.84_{\pm6.93}$ | $53.49_{\pm6.09}$ | $35.11_{\pm4.81}$ | $25.12_{\pm5.07}$ | 47.75 |
| | Log-Rank | $28.72_{\pm26.90}$ | $46.48_{\pm36.20}$ | $25.04_{\pm26.83}$ | $28.19_{\pm27.90}$ | $17.34_{\pm15.65}$ | $2.96_{\pm3.73}$ | 24.79 |
| | DetectGPT | $37.04_{\pm10.21}$ | $24.75_{\pm10.03}$ | $5.52_{\pm1.93}$ | $21.45_{\pm14.25}$ | $15.45_{\pm7.05}$ | $4.48_{\pm2.57}$ | 18.12 |
| | F-DetectGPT | $4.24_{\pm1.51}$ | $3.85_{\pm2.05}$ | $31.28_{\pm3.18}$ | $35.74_{\pm3.97}$ | $0.00_{\pm0.00}$ | $0.16_{\pm0.20}$ | 12.55 |
| | ChatGPT-D | $60.72_{\pm14.49}$ | $71.48_{\pm4.55}$ | $65.36_{\pm17.74}$ | $85.78_{\pm7.09}$ | $23.18_{\pm4.53}$ | $1.68_{\pm0.89}$ | 51.37 |
| | **ChatGPT-STK** | $63.68_{\pm15.32}$ | $66.39_{\pm7.37}$ | $65.92_{\pm13.67}$ | $88.92_{\pm6.98}$ | $20.86_{\pm10.25}$ | $1.68_{\pm1.02}$ | 51.24 |
| | OpenAI-D | $79.12_{\pm37.59}$ | $\mathbf{98.11}_{\pm1.12}$ | $79.76_{\pm39.29}$ | $98.55_{\pm1.21}$ | $\underline{70.90}_{\pm32.85}$ | $59.28_{\pm31.57}$ | 80.96 |
| | **OpenAI-STK** | $95.44_{\pm2.82}$ | $96.56_{\pm1.90}$ | $\mathbf{99.76}_{\pm0.48}$ | $98.23_{\pm1.21}$ | $70.99_{\pm13.03}$ | $\mathbf{74.00}_{\pm15.92}$ | **89.16** |
| | MPU | $95.36_{\pm1.94}$ | $\mathbf{96.97}_{\pm1.61}$ | $98.24_{\pm0.82}$ | $98.63_{\pm1.07}$ | $57.34_{\pm6.19}$ | $66.72_{\pm14.28}$ | 85.54 |
| | **MPU-STK** | $\mathbf{95.92}_{\pm1.98}$ | $95.98_{\pm2.60}$ | $98.24_{\pm0.90}$ | $98.63_{\pm0.94}$ | $58.54_{\pm5.27}$ | $65.36_{\pm13.32}$ | 85.45 |
| | RADAR | $\underline{95.76}_{\pm1.57}$ | $94.67_{\pm2.06}$ | $98.08_{\pm1.39}$ | $\underline{98.96}_{\pm0.54}$ | $63.26_{\pm4.51}$ | $62.32_{\pm8.83}$ | 85.51 |
| | **RADAR-STK** | $95.76_{\pm1.87}$ | $95.57_{\pm1.59}$ | $\underline{98.32}_{\pm1.11}$ | $\mathbf{99.12}_{\pm0.64}$ | $64.64_{\pm5.05}$ | $61.52_{\pm10.33}$ | $\underline{85.82}$ |
| Reuters | Log-Likelihood | $77.84_{\pm5.19}$ | $14.88_{\pm5.98}$ | $86.08_{\pm3.38}$ | $93.76_{\pm2.03}$ | $11.20_{\pm4.45}$ | $15.04_{\pm6.86}$ | 49.80 |
| | Rank | $48.88_{\pm1.59}$ | $35.92_{\pm2.88}$ | $58.40_{\pm3.94}$ | $40.56_{\pm1.85}$ | $18.56_{\pm2.27}$ | $6.24_{\pm1.87}$ | 34.76 |
| | Log-Rank | $82.40_{\pm5.24}$ | $25.92_{\pm7.08}$ | $90.96_{\pm4.12}$ | $96.80_{\pm0.88}$ | $14.00_{\pm4.82}$ | $17.60_{\pm8.29}$ | 54.61 |
| | DetectGPT | $2.64_{\pm1.53}$ | $0.32_{\pm0.30}$ | $1.20_{\pm1.07}$ | $0.96_{\pm0.82}$ | $0.88_{\pm1.06}$ | $2.88_{\pm1.85}$ | 1.48 |
| | F-DetectGPT | $48.00_{\pm9.48}$ | $6.80_{\pm1.88}$ | $92.96_{\pm1.65}$ | $88.96_{\pm4.80}$ | $0.00_{\pm0.00}$ | $0.48_{\pm0.39}$ | 39.53 |
| | ChatGPT-D | $77.60_{\pm24.67}$ | $74.48_{\pm24.64}$ | $76.24_{\pm28.15}$ | $76.00_{\pm29.72}$ | $37.52_{\pm15.02}$ | $24.72_{\pm22.20}$ | 61.09 |
| | **ChatGPT-STK** | $93.60_{\pm4.55}$ | $88.96_{\pm5.73}$ | $96.64_{\pm3.56}$ | $97.68_{\pm1.20}$ | $39.76_{\pm10.49}$ | $25.44_{\pm16.75}$ | 73.68 |
| | OpenAI-D | $82.88_{\pm32.04}$ | $81.12_{\pm18.73}$ | $81.84_{\pm34.33}$ | $88.08_{\pm20.85}$ | $48.08_{\pm11.99}$ | $22.24_{\pm5.85}$ | 67.37 |
| | **OpenAI-STK** | $\underline{98.32}_{\pm0.59}$ | $83.84_{\pm3.66}$ | $99.52_{\pm0.30}$ | $98.16_{\pm0.48}$ | $38.08_{\pm6.02}$ | $37.36_{\pm2.91}$ | 75.88 |
| | MPU | $\mathbf{100.00}_{\pm0.00}$ | $\underline{98.40}_{\pm0.76}$ | $\mathbf{100.00}_{\pm0.00}$ | $\underline{99.20}_{\pm0.36}$ | $72.72_{\pm4.80}$ | $91.44_{\pm2.13}$ | 93.63 |
| | **MPU-STK** | $\mathbf{100.00}_{\pm0.00}$ | $98.40_{\pm0.80}$ | $\mathbf{100.00}_{\pm0.00}$ | $98.96_{\pm0.41}$ | $70.80_{\pm6.48}$ | $\underline{94.72}_{\pm3.76}$ | 93.81 |
| | RADAR | $\mathbf{100.00}_{\pm0.00}$ | $\mathbf{100.00}_{\pm0.00}$ | $99.68_{\pm0.30}$ | $\mathbf{99.92}_{\pm0.16}$ | $\underline{91.36}_{\pm2.30}$ | $\mathbf{95.84}_{\pm1.73}$ | $\underline{97.80}$ |
| | **RADAR-STK** | $\mathbf{100.00}_{\pm0.00}$ | $\mathbf{100.00}_{\pm0.00}$ | $\underline{99.68}_{\pm0.30}$ | $\mathbf{99.92}_{\pm0.16}$ | $\mathbf{91.44}_{\pm2.31}$ | $\mathbf{95.84}_{\pm1.73}$ | **97.81** |
| SQuAD1 | Log-Likelihood | $1.04_{\pm0.57}$ | $5.00_{\pm2.93}$ | $2.11_{\pm0.29}$ | $1.85_{\pm1.12}$ | $5.93_{\pm3.44}$ | $21.85_{\pm10.14}$ | 6.30 |
| | Rank | $2.43_{\pm0.43}$ | $3.81_{\pm1.04}$ | $1.87_{\pm0.68}$ | $1.97_{\pm0.28}$ | $4.30_{\pm1.01}$ | $7.98_{\pm1.52}$ | 3.73 |
| | Log-Rank | $1.97_{\pm1.19}$ | $4.88_{\pm3.25}$ | $2.34_{\pm0.83}$ | $1.97_{\pm1.35}$ | $4.88_{\pm2.72}$ | $19.77_{\pm11.36}$ | 5.97 |
| | DetectGPT | $0.46_{\pm0.67}$ | $0.48_{\pm0.45}$ | $0.47_{\pm0.44}$ | $0.58_{\pm0.73}$ | $0.81_{\pm0.59}$ | $0.69_{\pm0.43}$ | 0.58 |
| | F-DetectGPT | $4.28_{\pm1.35}$ | $6.90_{\pm1.53}$ | $2.69_{\pm1.41}$ | $2.77_{\pm1.12}$ | $5.35_{\pm2.19}$ | $28.44_{\pm3.93}$ | 8.41 |
| | ChatGPT-D | $1.27_{\pm0.43}$ | $1.07_{\pm0.87}$ | $1.17_{\pm0.83}$ | $0.81_{\pm0.59}$ | $1.16_{\pm0.74}$ | $5.32_{\pm3.88}$ | 1.80 |
| | **ChatGPT-STK** | $1.73_{\pm1.75}$ | $3.81_{\pm1.53}$ | $1.87_{\pm0.44}$ | $1.50_{\pm1.58}$ | $1.51_{\pm1.31}$ | $6.59_{\pm4.18}$ | 2.84 |
| | OpenAI-D | $\underline{51.21}_{\pm23.65}$ | $51.07_{\pm22.43}$ | $32.28_{\pm19.01}$ | $\underline{39.88}_{\pm18.98}$ | $\mathbf{62.67}_{\pm26.90}$ | $\mathbf{76.18}_{\pm12.19}$ | **52.22** |
| | **OpenAI-STK** | $\mathbf{53.41}_{\pm26.65}$ | $\underline{51.90}_{\pm20.23}$ | $32.98_{\pm18.26}$ | $\mathbf{41.39}_{\pm20.40}$ | $60.58_{\pm27.86}$ | $\underline{72.25}_{\pm14.07}$ | $\underline{52.09}$ |
| | MPU | $41.16_{\pm29.06}$ | $17.02_{\pm6.63}$ | $\underline{46.78}_{\pm23.16}$ | $33.06_{\pm23.42}$ | $41.86_{\pm25.14}$ | $51.33_{\pm23.66}$ | 38.54 |
| | **MPU-STK** | $46.71_{\pm29.55}$ | $\mathbf{66.43}_{\pm24.57}$ | $\mathbf{48.07}_{\pm22.73}$ | $37.23_{\pm24.65}$ | $41.40_{\pm25.34}$ | $55.84_{\pm22.94}$ | 49.28 |
| | RADAR | $28.32_{\pm13.89}$ | $30.48_{\pm9.01}$ | $33.45_{\pm10.05}$ | $21.97_{\pm11.86}$ | $29.88_{\pm5.56}$ | $26.94_{\pm13.33}$ | 28.51 |
| | **RADAR-STK** | $28.90_{\pm15.02}$ | $32.14_{\pm9.55}$ | $36.37_{\pm7.51}$ | $21.27_{\pm13.05}$ | $31.63_{\pm5.01}$ | $30.06_{\pm13.26}$ | 30.06 |

## H.13 SENSITIVITY ANALYSIS OF HYPERPARAMETERS

**Sensitivity of Sequence Length** $k$. Fig. 19 illustrates the detection performance comparison under different sequence lengths (number of sentences per sequence). Notably, on the x-axis, 0 indicates the original detector without the proposed strategy configured. From the figure, we observe that within a certain range of sequence lengths, detection performance improves as the sequence length increases. This is because short sequences are difficult to detect, and longer sequences allow for better identification of human-generated text within mixed texts, thus enhancing detection accuracy. However, when the sequence length becomes excessively long, performance starts to decline. We suspect that when the sequence length increases too much, human text is more likely to be mixed with machine-generated text, which causes the proportion of pure human text to decrease and, in turn, become more difficult to detect accurately. Therefore, for most detectors, $k = 3$ or $4$ serves as a good compromise. Furthermore, we observe that when detector performance is poor—such as on SQuAD1—a smaller $k$ is necessary for better exploration. In such cases, we recommend $k = 1$ or 2.

**Sensitivity of Filter Ratio** $\tau$. Fig. 20 presents the comparison of detection performance across different filter ratios $\tau$. The results show that as the filtering ratio increases, detection performance initially improves and then declines. This supports our motivation for introducing this hyperparameter. Specifically, excessively removing human-generated text can result in short sequences dominating

Table 8: Performance concerning AUROC. Detectors are trained on ChatGLM texts.

| Dataset | Method | ChatGPT | GPT-4 | GPT-turbo | ChatGLM | Dolly | Claude/StableLM | Avg. |
|---|---|---|---|---|---|---|---|---|
| Essay | Log-Likelihood | $98.48_{\pm0.43}$ | $96.46_{\pm1.05}$ | $98.85_{\pm0.42}$ | $98.78_{\pm0.15}$ | $90.64_{\pm0.35}$ | $92.16_{\pm0.32}$ | 95.90 |
| | Rank | $92.84_{\pm0.99}$ | $90.93_{\pm0.74}$ | $98.19_{\pm0.37}$ | $81.51_{\pm1.32}$ | $80.45_{\pm1.41}$ | $86.69_{\pm1.06}$ | 88.44 |
| | Log-Rank | $98.64_{\pm0.43}$ | $96.85_{\pm0.89}$ | $98.92_{\pm0.40}$ | $98.94_{\pm0.16}$ | $90.41_{\pm0.50}$ | $91.47_{\pm0.45}$ | 95.87 |
| | DetectGPT | $96.86_{\pm0.97}$ | $95.64_{\pm0.45}$ | $44.40_{\pm1.73}$ | $95.60_{\pm0.62}$ | $92.55_{\pm0.58}$ | $46.36_{\pm0.71}$ | 78.57 |
| | F-DetectGPT | $83.86_{\pm1.52}$ | $84.45_{\pm1.58}$ | $94.12_{\pm0.72}$ | $93.62_{\pm0.73}$ | $55.86_{\pm2.54}$ | $58.96_{\pm1.52}$ | 78.48 |
| | ChatGPT-D | $97.86_{\pm0.63}$ | $95.40_{\pm2.44}$ | $90.65_{\pm1.69}$ | $99.90_{\pm0.16}$ | $79.10_{\pm6.61}$ | $33.03_{\pm7.16}$ | 82.66 |
| | **ChatGPT-STK** | $98.45_{\pm0.40}$ | $97.08_{\pm0.99}$ | $88.03_{\pm3.10}$ | $99.92_{\pm0.11}$ | $85.81_{\pm4.37}$ | $46.70_{\pm6.20}$ | 86.00 |
| | OpenAI-D | $99.11_{\pm0.53}$ | $99.77_{\pm0.23}$ | $94.01_{\pm1.32}$ | $99.89_{\pm0.13}$ | $96.03_{\pm1.37}$ | $62.51_{\pm1.96}$ | 91.89 |
| | **OpenAI-STK** | $99.72_{\pm0.14}$ | $99.67_{\pm0.18}$ | $96.37_{\pm0.49}$ | $99.84_{\pm0.17}$ | $\mathbf{97.79}_{\pm0.43}$ | $71.36_{\pm1.53}$ | 94.13 |
| | MPU | $\underline{99.86}_{\pm0.12}$ | $\mathbf{99.93}_{\pm0.04}$ | $96.75_{\pm0.25}$ | $\mathbf{99.99}_{\pm0.01}$ | $96.72_{\pm0.53}$ | $86.57_{\pm1.95}$ | 96.64 |
| | **MPU-STK** | $\mathbf{99.87}_{\pm0.12}$ | $\mathbf{99.93}_{\pm0.04}$ | $96.64_{\pm0.24}$ | $\mathbf{99.99}_{\pm0.01}$ | $96.99_{\pm0.29}$ | $91.69_{\pm1.39}$ | 97.52 |
| | RADAR | $99.79_{\pm0.11}$ | $\underline{99.80}_{\pm0.09}$ | $\underline{99.75}_{\pm0.19}$ | $99.98_{\pm0.02}$ | $97.43_{\pm0.46}$ | $\mathbf{97.41}_{\pm0.39}$ | 99.03 |
| | **RADAR-STK** | $99.80_{\pm0.11}$ | $99.76_{\pm0.09}$ | $\mathbf{99.76}_{\pm0.20}$ | $99.98_{\pm0.03}$ | $\underline{97.67}_{\pm0.36}$ | $\underline{97.36}_{\pm0.43}$ | **99.05** |
| Reuters | Log-Likelihood | $97.59_{\pm0.37}$ | $74.85_{\pm0.50}$ | $98.54_{\pm0.38}$ | $99.54_{\pm0.22}$ | $60.09_{\pm2.13}$ | $85.66_{\pm1.04}$ | 86.05 |
| | Rank | $84.00_{\pm0.88}$ | $74.12_{\pm1.24}$ | $91.82_{\pm0.60}$ | $70.97_{\pm1.65}$ | $54.88_{\pm2.45}$ | $67.78_{\pm1.14}$ | 73.93 |
| | Log-Rank | $97.86_{\pm0.35}$ | $79.95_{\pm0.29}$ | $98.82_{\pm0.37}$ | $99.65_{\pm0.20}$ | $61.24_{\pm2.15}$ | $84.88_{\pm1.13}$ | 87.07 |
| | DetectGPT | $92.78_{\pm1.34}$ | $85.71_{\pm2.02}$ | $49.24_{\pm1.65}$ | $91.59_{\pm1.67}$ | $83.41_{\pm1.73}$ | $66.59_{\pm2.53}$ | 78.22 |
| | F-DetectGPT | $96.23_{\pm0.19}$ | $66.02_{\pm1.53}$ | $98.85_{\pm0.30}$ | $98.58_{\pm0.35}$ | $40.38_{\pm1.31}$ | $66.16_{\pm2.58}$ | 77.70 |
| | ChatGPT-D | $98.80_{\pm0.71}$ | $97.95_{\pm0.83}$ | $98.73_{\pm0.79}$ | $99.72_{\pm0.31}$ | $80.34_{\pm5.32}$ | $40.49_{\pm6.96}$ | 86.00 |
| | **ChatGPT-STK** | $98.98_{\pm0.60}$ | $98.56_{\pm0.43}$ | $99.23_{\pm0.39}$ | $99.79_{\pm0.23}$ | $84.47_{\pm4.73}$ | $59.35_{\pm5.00}$ | 90.06 |
| | OpenAI-D | $99.71_{\pm0.22}$ | $99.69_{\pm0.29}$ | $99.74_{\pm0.16}$ | $99.99_{\pm0.01}$ | $97.83_{\pm0.82}$ | $66.96_{\pm2.61}$ | 93.99 |
| | **OpenAI-STK** | $\underline{99.77}_{\pm0.17}$ | $99.61_{\pm0.11}$ | $99.78_{\pm0.09}$ | $99.98_{\pm0.01}$ | $96.91_{\pm0.53}$ | $78.19_{\pm2.14}$ | 95.71 |
| | MPU | $\mathbf{100.00}_{\pm0.00}$ | $\underline{99.79}_{\pm0.11}$ | $99.97_{\pm0.04}$ | $\underline{99.99}_{\pm0.01}$ | $96.62_{\pm0.53}$ | $94.78_{\pm1.36}$ | 98.52 |
| | **MPU-STK** | $\mathbf{100.00}_{\pm0.00}$ | $99.75_{\pm0.17}$ | $\mathbf{99.99}_{\pm0.01}$ | $\underline{99.99}_{\pm0.01}$ | $97.10_{\pm0.52}$ | $\underline{98.00}_{\pm0.22}$ | 99.14 |
| | RADAR | $\mathbf{100.00}_{\pm0.00}$ | $\mathbf{100.00}_{\pm0.00}$ | $99.97_{\pm0.03}$ | $\mathbf{100.00}_{\pm0.00}$ | $\underline{99.44}_{\pm0.19}$ | $\mathbf{99.79}_{\pm0.09}$ | **99.87** |
| | **RADAR-STK** | $\mathbf{100.00}_{\pm0.00}$ | $\mathbf{100.00}_{\pm0.00}$ | $\underline{99.98}_{\pm0.03}$ | $\mathbf{100.00}_{\pm0.00}$ | $\mathbf{99.46}_{\pm0.20}$ | $\mathbf{99.79}_{\pm0.09}$ | **99.87** |
| SQuAD1 | Log-Likelihood | $69.62_{\pm0.28}$ | $69.61_{\pm1.06}$ | $67.18_{\pm0.67}$ | $65.67_{\pm0.56}$ | $67.24_{\pm1.10}$ | $85.82_{\pm1.96}$ | 70.86 |
| | Rank | $63.45_{\pm0.91}$ | $64.71_{\pm0.99}$ | $62.30_{\pm0.77}$ | $61.56_{\pm0.80}$ | $64.16_{\pm0.86}$ | $73.52_{\pm1.14}$ | 64.95 |
| | Log-Rank | $68.78_{\pm0.34}$ | $69.52_{\pm0.79}$ | $66.33_{\pm0.59}$ | $64.99_{\pm0.65}$ | $66.83_{\pm1.04}$ | $85.50_{\pm1.97}$ | 70.33 |
| | DetectGPT | $52.60_{\pm2.74}$ | $47.47_{\pm1.42}$ | $52.05_{\pm1.87}$ | $50.88_{\pm2.47}$ | $50.24_{\pm2.20}$ | $51.12_{\pm2.44}$ | 50.73 |
| | F-DetectGPT | $67.91_{\pm0.23}$ | $63.84_{\pm1.73}$ | $64.67_{\pm0.20}$ | $63.36_{\pm0.42}$ | $66.32_{\pm1.36}$ | $79.62_{\pm0.63}$ | 67.62 |
| | ChatGPT-D | $58.03_{\pm1.19}$ | $42.38_{\pm4.85}$ | $56.33_{\pm1.02}$ | $55.71_{\pm0.92}$ | $55.98_{\pm1.41}$ | $67.19_{\pm2.64}$ | 55.94 |
| | **ChatGPT-STK** | $57.96_{\pm1.30}$ | $59.49_{\pm0.74}$ | $56.25_{\pm0.99}$ | $55.62_{\pm1.04}$ | $55.92_{\pm0.93}$ | $65.95_{\pm1.72}$ | 58.53 |
| | OpenAI-D | $\underline{98.77}_{\pm0.34}$ | $\mathbf{98.99}_{\pm0.57}$ | $98.33_{\pm0.66}$ | $95.32_{\pm0.93}$ | $\mathbf{97.71}_{\pm0.39}$ | $\underline{99.25}_{\pm0.31}$ | 98.06 |
| | **OpenAI-STK** | $\mathbf{98.90}_{\pm0.16}$ | $\underline{98.98}_{\pm0.40}$ | $98.31_{\pm0.64}$ | $95.29_{\pm0.93}$ | $\underline{97.62}_{\pm0.51}$ | $\mathbf{99.34}_{\pm0.19}$ | **98.07** |
| | MPU | $98.60_{\pm0.34}$ | $98.47_{\pm1.08}$ | $\mathbf{98.72}_{\pm0.54}$ | $\mathbf{96.32}_{\pm1.34}$ | $97.56_{\pm0.47}$ | $98.72_{\pm0.29}$ | 98.06 |
| | **MPU-STK** | $98.50_{\pm0.50}$ | $98.96_{\pm0.88}$ | $\underline{98.62}_{\pm0.64}$ | $\underline{96.16}_{\pm1.36}$ | $97.41_{\pm0.40}$ | $98.65_{\pm0.41}$ | 98.05 |
| | RADAR | $81.33_{\pm1.72}$ | $80.23_{\pm0.80}$ | $80.91_{\pm1.51}$ | $77.72_{\pm0.95}$ | $77.52_{\pm1.95}$ | $80.95_{\pm1.89}$ | 79.78 |
| | **RADAR-STK** | $81.59_{\pm1.86}$ | $80.69_{\pm0.71}$ | $81.37_{\pm1.39}$ | $77.88_{\pm0.76}$ | $77.91_{\pm1.90}$ | $81.03_{\pm1.95}$ | 80.08 |

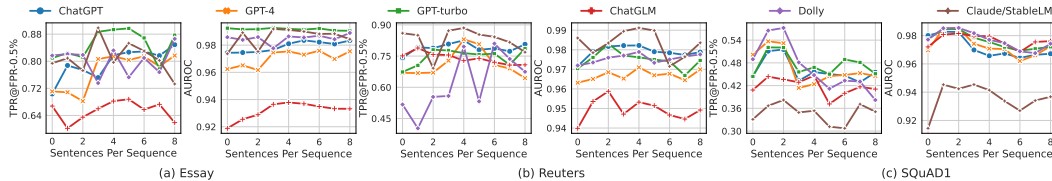

Figure 19: The average detection performance (TPR@FPR-0.5% or AUROC) of OpenAI-STK under different sequence lengths (sentences per sequence). We train the detector on various texts (different lines) and report the average performance tested on various LLM texts. An x-coordinate of 0 indicates the original detector OpenAI-D.

the prediction of the entire text, which can lead to mispredictions. Consequently, 0.5 serves as a suitable compromise.

**Sensitivity of Confidence Threshold** $r_s$. Fig. 21 illustrates the comparison of detection performance under different confidence thresholds $r_s$. In practical classification tasks, a threshold of 0.5 is commonly used. However, as discussed in Section 3.2, due to the difficulty of detecting short texts and the unknown proportion of human-generated text in the mixed text, a smaller threshold is necessary. This can effectively mitigate the misclassification, prevent critical MGT from being filtered out, and thereby avoid a decline in detection performance. Sensitivity analysis of this parameter

Table 9: Performance concerning TPR@FPR-0.5%. Detectors are trained on ChatGLM texts.

| Dataset | Method | ChatGPT | GPT-4 | GPT-turbo | ChatGLM | Dolly | Claude/StableLM | Avg. |
|---|---|---|---|---|---|---|---|---|
| Essay | Log-Likelihood | $24.08_{\pm22.67}$ | $37.70_{\pm30.63}$ | $23.12_{\pm24.38}$ | $5.86_{\pm7.02}$ | $12.45_{\pm12.04}$ | $2.48_{\pm3.06}$ | 17.62 |
| | Rank | $55.60_{\pm3.65}$ | $51.31_{\pm5.91}$ | $65.84_{\pm6.93}$ | $53.49_{\pm6.09}$ | $35.11_{\pm4.81}$ | $25.12_{\pm5.07}$ | 47.75 |
| | Log-Rank | $28.72_{\pm26.90}$ | $46.48_{\pm36.20}$ | $25.04_{\pm26.83}$ | $28.19_{\pm27.90}$ | $17.34_{\pm15.65}$ | $2.96_{\pm3.73}$ | 24.79 |
| | DetectGPT | $37.04_{\pm10.21}$ | $24.75_{\pm10.03}$ | $5.52_{\pm1.93}$ | $21.45_{\pm14.25}$ | $15.45_{\pm7.05}$ | $4.48_{\pm2.57}$ | 18.12 |
| | F-DetectGPT | $4.24_{\pm1.51}$ | $3.85_{\pm2.05}$ | $31.28_{\pm3.18}$ | $35.74_{\pm3.97}$ | $0.00_{\pm0.00}$ | $0.16_{\pm0.20}$ | 12.55 |
| | ChatGPT-D | $80.08_{\pm8.29}$ | $79.51_{\pm2.47}$ | $34.48_{\pm8.86}$ | $96.71_{\pm5.80}$ | $34.76_{\pm8.77}$ | $0.48_{\pm0.30}$ | 54.34 |
| | **ChatGPT-STK** | $85.36_{\pm3.83}$ | $83.36_{\pm4.43}$ | $39.20_{\pm7.94}$ | $97.99_{\pm3.27}$ | $43.43_{\pm4.46}$ | $2.48_{\pm1.09}$ | 58.64 |
| | OpenAI-D | $83.84_{\pm23.24}$ | $85.66_{\pm21.19}$ | $57.36_{\pm15.97}$ | $95.74_{\pm8.11}$ | $\underline{72.19}_{\pm19.00}$ | $5.84_{\pm4.57}$ | 66.77 |
| | **OpenAI-STK** | $93.12_{\pm1.55}$ | $94.51_{\pm2.60}$ | $61.68_{\pm7.31}$ | $98.63_{\pm1.07}$ | $61.89_{\pm10.59}$ | $9.68_{\pm5.32}$ | 69.92 |
| | MPU | $\underline{96.80}_{\pm1.29}$ | $\underline{98.69}_{\pm0.66}$ | $51.52_{\pm7.75}$ | $\mathbf{99.60}_{\pm0.44}$ | $70.39_{\pm6.73}$ | $18.16_{\pm11.12}$ | 72.53 |
| | **MPU-STK** | $\mathbf{97.28}_{\pm0.89}$ | $\mathbf{98.85}_{\pm0.40}$ | $52.48_{\pm7.96}$ | $\mathbf{99.60}_{\pm0.44}$ | $72.88_{\pm5.59}$ | $27.84_{\pm12.28}$ | 74.82 |
| | RADAR | $89.92_{\pm8.96}$ | $94.34_{\pm2.28}$ | $\underline{86.72}_{\pm14.46}$ | $99.12_{\pm1.20}$ | $58.28_{\pm16.43}$ | $\underline{47.68}_{\pm19.59}$ | 79.34 |
| | **RADAR-STK** | $90.24_{\pm8.37}$ | $94.10_{\pm2.97}$ | $\mathbf{87.04}_{\pm14.19}$ | $99.04_{\pm1.36}$ | $58.63_{\pm16.56}$ | $\mathbf{49.68}_{\pm19.93}$ | **79.79** |
| Reuters | Log-Likelihood | $77.84_{\pm5.19}$ | $14.88_{\pm5.98}$ | $86.08_{\pm3.38}$ | $93.76_{\pm2.03}$ | $11.20_{\pm4.45}$ | $15.04_{\pm6.86}$ | 49.80 |
| | Rank | $48.88_{\pm1.59}$ | $35.92_{\pm2.88}$ | $58.40_{\pm3.94}$ | $40.56_{\pm1.85}$ | $18.56_{\pm2.27}$ | $6.24_{\pm1.87}$ | 34.76 |
| | Log-Rank | $82.40_{\pm5.24}$ | $25.92_{\pm7.08}$ | $90.96_{\pm4.12}$ | $96.80_{\pm0.88}$ | $14.00_{\pm4.82}$ | $17.60_{\pm8.29}$ | 54.61 |
| | DetectGPT | $4.40_{\pm2.62}$ | $0.64_{\pm0.54}$ | $2.32_{\pm1.87}$ | $2.56_{\pm2.80}$ | $0.48_{\pm0.47}$ | $3.04_{\pm1.61}$ | 2.24 |
| | F-DetectGPT | $48.00_{\pm9.48}$ | $6.80_{\pm1.88}$ | $92.96_{\pm1.65}$ | $88.96_{\pm4.80}$ | $0.00_{\pm0.00}$ | $0.48_{\pm0.39}$ | 39.53 |
| | ChatGPT-D | $91.28_{\pm6.11}$ | $85.20_{\pm7.05}$ | $89.12_{\pm6.82}$ | $97.92_{\pm1.46}$ | $41.28_{\pm16.63}$ | $2.80_{\pm2.83}$ | 67.93 |
| | **ChatGPT-STK** | $94.64_{\pm2.52}$ | $90.16_{\pm2.97}$ | $93.92_{\pm2.57}$ | $98.88_{\pm0.93}$ | $50.40_{\pm11.41}$ | $8.88_{\pm0.89}$ | 72.81 |
| | OpenAI-D | $96.80_{\pm2.65}$ | $86.96_{\pm13.16}$ | $96.24_{\pm2.02}$ | $99.60_{\pm0.44}$ | $60.64_{\pm12.43}$ | $10.32_{\pm3.91}$ | 75.09 |
| | **OpenAI-STK** | $97.44_{\pm0.41}$ | $87.92_{\pm2.92}$ | $97.36_{\pm0.86}$ | $99.44_{\pm0.20}$ | $50.08_{\pm6.35}$ | $15.36_{\pm3.96}$ | 74.60 |
| | MPU | $\mathbf{100.00}_{\pm0.00}$ | $\underline{97.12}_{\pm1.32}$ | $\mathbf{99.76}_{\pm0.20}$ | $\underline{99.76}_{\pm0.20}$ | $67.60_{\pm5.76}$ | $55.92_{\pm10.37}$ | 86.69 |
| | **MPU-STK** | $\underline{99.92}_{\pm0.16}$ | $96.96_{\pm1.40}$ | $\underline{99.76}_{\pm0.32}$ | $99.60_{\pm0.44}$ | $63.44_{\pm7.23}$ | $69.36_{\pm8.86}$ | 88.17 |
| | RADAR | $99.92_{\pm0.16}$ | $\mathbf{99.92}_{\pm0.16}$ | $99.28_{\pm0.85}$ | $\mathbf{99.92}_{\pm0.16}$ | $\mathbf{90.16}_{\pm2.82}$ | $\mathbf{93.92}_{\pm2.62}$ | **97.19** |
| | **RADAR-STK** | $99.92_{\pm0.16}$ | $\mathbf{99.92}_{\pm0.16}$ | $99.28_{\pm0.85}$ | $\mathbf{99.92}_{\pm0.16}$ | $\underline{89.84}_{\pm2.66}$ | $\underline{93.84}_{\pm2.62}$ | $\underline{97.12}$ |
| SQuAD1 | Log-Likelihood | $1.04_{\pm0.57}$ | $5.00_{\pm2.93}$ | $2.11_{\pm0.29}$ | $1.85_{\pm1.12}$ | $5.93_{\pm3.44}$ | $21.85_{\pm10.14}$ | 6.30 |
| | Rank | $2.43_{\pm0.43}$ | $3.81_{\pm1.04}$ | $1.87_{\pm0.68}$ | $1.97_{\pm0.28}$ | $4.30_{\pm1.01}$ | $7.98_{\pm1.52}$ | 3.73 |
| | Log-Rank | $1.97_{\pm1.19}$ | $4.88_{\pm3.25}$ | $2.34_{\pm0.83}$ | $1.97_{\pm1.35}$ | $4.88_{\pm2.72}$ | $19.77_{\pm11.36}$ | 5.97 |
| | DetectGPT | $0.35_{\pm0.69}$ | $0.36_{\pm0.48}$ | $0.47_{\pm0.44}$ | $0.58_{\pm0.73}$ | $0.58_{\pm0.37}$ | $0.81_{\pm0.28}$ | 0.52 |
| | F-DetectGPT | $4.28_{\pm1.35}$ | $6.90_{\pm1.53}$ | $2.69_{\pm1.41}$ | $2.77_{\pm1.12}$ | $5.35_{\pm2.19}$ | $28.44_{\pm3.93}$ | 8.41 |
| | ChatGPT-D | $1.04_{\pm0.99}$ | $1.31_{\pm0.69}$ | $1.40_{\pm0.95}$ | $1.16_{\pm0.82}$ | $1.74_{\pm1.27}$ | $6.01_{\pm3.26}$ | 2.11 |
| | **ChatGPT-STK** | $2.66_{\pm1.96}$ | $3.81_{\pm1.28}$ | $1.40_{\pm0.60}$ | $1.62_{\pm1.69}$ | $1.05_{\pm0.44}$ | $3.82_{\pm1.49}$ | 2.39 |
| | OpenAI-D | $36.53_{\pm17.45}$ | $41.79_{\pm19.02}$ | $36.96_{\pm16.77}$ | $28.67_{\pm13.27}$ | $50.58_{\pm24.47}$ | $\underline{63.24}_{\pm19.04}$ | 42.96 |
| | **OpenAI-STK** | $\underline{39.77}_{\pm15.28}$ | $\underline{47.26}_{\pm22.70}$ | $31.70_{\pm14.38}$ | $29.94_{\pm10.70}$ | $51.86_{\pm26.87}$ | $\mathbf{65.66}_{\pm16.99}$ | $\underline{44.37}$ |
| | MPU | $\mathbf{41.85}_{\pm26.11}$ | $23.69_{\pm33.19}$ | $\mathbf{45.03}_{\pm14.50}$ | $\mathbf{33.64}_{\pm20.00}$ | $\underline{54.19}_{\pm29.09}$ | $53.29_{\pm18.57}$ | 41.95 |
| | **MPU-STK** | $38.73_{\pm19.95}$ | $\mathbf{53.33}_{\pm24.68}$ | $\underline{42.69}_{\pm16.63}$ | $\underline{30.17}_{\pm16.73}$ | $\mathbf{55.81}_{\pm29.23}$ | $51.33_{\pm14.03}$ | **45.34** |
| | RADAR | $28.09_{\pm13.16}$ | $31.19_{\pm11.73}$ | $35.32_{\pm6.78}$ | $22.66_{\pm11.61}$ | $30.93_{\pm5.47}$ | $27.40_{\pm10.59}$ | 29.27 |
| | **RADAR-STK** | $30.52_{\pm11.76}$ | $31.07_{\pm11.41}$ | $37.43_{\pm4.62}$ | $23.82_{\pm10.28}$ | $31.40_{\pm5.95}$ | $26.82_{\pm9.64}$ | 30.17 |

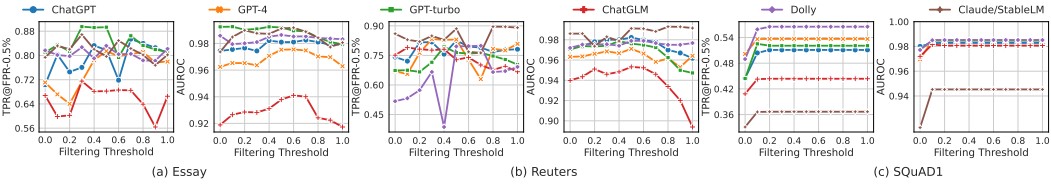

Figure 20: The average detection performance (TPR@FPR-0.5% or AUROC) of OpenAI-STK under different filtering thresholds. We train the detector on various texts (different lines) and report the average performance tested on various LLM texts. An x-coordinate of 0 indicates the original detector OpenAI-D.

supports this observation, indicating that 0.5 is not an optimal choice and a smaller value is needed. Besides, the result shows that the proposed strategy is relatively insensitive to smaller thresholds, making 0.005–0.05 a promising range in practice.

Table 10: Performance concerning AUROC. Detectors are trained on Dolly texts.

| Dataset | Method | ChatGPT | GPT-4 | GPT-turbo | ChatGLM | Dolly | Claude/StableLM | Avg. |
|---|---|---|---|---|---|---|---|---|
| Essay | Log-Likelihood | $98.48_{\pm0.43}$ | $96.46_{\pm1.05}$ | $98.85_{\pm0.42}$ | $98.78_{\pm0.15}$ | $90.64_{\pm0.35}$ | $92.16_{\pm0.32}$ | 95.90 |
| | Rank | $92.84_{\pm0.99}$ | $90.93_{\pm0.74}$ | $98.19_{\pm0.37}$ | $81.51_{\pm1.32}$ | $80.45_{\pm1.41}$ | $86.69_{\pm1.06}$ | 88.44 |
| | Log-Rank | $98.64_{\pm0.43}$ | $96.85_{\pm0.89}$ | $98.92_{\pm0.40}$ | $98.94_{\pm0.16}$ | $90.41_{\pm0.50}$ | $91.47_{\pm0.45}$ | 95.87 |
| | DetectGPT | $96.86_{\pm0.97}$ | $95.64_{\pm0.45}$ | $44.40_{\pm1.73}$ | $95.60_{\pm0.62}$ | $92.55_{\pm0.58}$ | $46.36_{\pm0.71}$ | 78.57 |
| | F-DetectGPT | $83.86_{\pm1.52}$ | $84.45_{\pm1.58}$ | $94.12_{\pm0.72}$ | $93.62_{\pm0.73}$ | $55.86_{\pm2.54}$ | $58.96_{\pm1.52}$ | 78.48 |
| | ChatGPT-D | $99.08_{\pm0.84}$ | $98.90_{\pm1.45}$ | $89.54_{\pm2.35}$ | $99.86_{\pm0.22}$ | $93.15_{\pm6.08}$ | $62.74_{\pm13.69}$ | 90.54 |
| | **ChatGPT-STK** | $99.08_{\pm0.45}$ | $98.89_{\pm1.07}$ | $90.05_{\pm2.28}$ | $99.89_{\pm0.11}$ | $93.32_{\pm4.06}$ | $65.73_{\pm12.96}$ | 91.16 |
| | OpenAI-D | $99.81_{\pm0.21}$ | $99.86_{\pm0.15}$ | $99.21_{\pm0.40}$ | $99.98_{\pm0.03}$ | $99.48_{\pm0.46}$ | $93.16_{\pm2.83}$ | 98.58 |
| | **OpenAI-STK** | $\underline{99.90}_{\pm0.14}$ | $\underline{99.95}_{\pm0.04}$ | $99.22_{\pm0.20}$ | $\underline{99.99}_{\pm0.01}$ | $\underline{99.68}_{\pm0.25}$ | $92.71_{\pm0.80}$ | 98.58 |
| | MPU | $\mathbf{100.00}_{\pm0.00}$ | $\mathbf{99.98}_{\pm0.03}$ | $97.88_{\pm0.43}$ | $\mathbf{100.00}_{\pm0.00}$ | $\mathbf{99.86}_{\pm0.04}$ | $98.74_{\pm0.46}$ | 99.41 |
| | **MPU-STK** | $\mathbf{100.00}_{\pm0.00}$ | $\mathbf{99.98}_{\pm0.03}$ | $98.16_{\pm0.39}$ | $\mathbf{100.00}_{\pm0.00}$ | $\mathbf{99.86}_{\pm0.04}$ | $98.80_{\pm0.32}$ | 99.47 |
| | RADAR | $99.69_{\pm0.10}$ | $99.89_{\pm0.07}$ | $\underline{99.39}_{\pm0.23}$ | $99.96_{\pm0.02}$ | $99.36_{\pm0.13}$ | $\mathbf{99.69}_{\pm0.05}$ | $\mathbf{99.66}$ |
| | **RADAR-STK** | $99.70_{\pm0.11}$ | $99.89_{\pm0.07}$ | $\mathbf{99.39}_{\pm0.31}$ | $99.97_{\pm0.02}$ | $99.35_{\pm0.16}$ | $\underline{99.68}_{\pm0.08}$ | $\mathbf{99.66}$ |
| Reuters | Log-Likelihood | $21.47_{\pm38.08}$ | $35.27_{\pm20.02}$ | $20.75_{\pm38.74}$ | $20.35_{\pm39.69}$ | $44.75_{\pm8.87}$ | $28.28_{\pm28.30}$ | 28.48 |
| | Rank | $16.00_{\pm0.88}$ | $25.88_{\pm1.24}$ | $8.18_{\pm0.60}$ | $29.03_{\pm1.65}$ | $45.12_{\pm2.45}$ | $32.22_{\pm1.14}$ | 26.07 |
| | Log-Rank | $2.14_{\pm0.35}$ | $20.05_{\pm0.29}$ | $1.18_{\pm0.37}$ | $0.35_{\pm0.20}$ | $38.76_{\pm2.15}$ | $15.12_{\pm1.13}$ | 12.93 |
| | DetectGPT | $92.78_{\pm1.34}$ | $85.71_{\pm2.02}$ | $49.24_{\pm1.65}$ | $91.59_{\pm1.67}$ | $83.41_{\pm1.73}$ | $66.59_{\pm2.53}$ | 78.22 |
| | F-DetectGPT | $96.23_{\pm0.19}$ | $66.02_{\pm1.53}$ | $98.85_{\pm0.30}$ | $98.58_{\pm0.35}$ | $40.38_{\pm1.31}$ | $66.16_{\pm2.58}$ | 77.70 |
| | ChatGPT-D | $99.67_{\pm0.16}$ | $99.26_{\pm0.68}$ | $99.28_{\pm0.50}$ | $99.13_{\pm0.97}$ | $94.75_{\pm2.38}$ | $78.91_{\pm2.78}$ | 95.16 |
| | **ChatGPT-STK** | $\underline{99.53}_{\pm0.50}$ | $98.96_{\pm0.89}$ | $99.36_{\pm0.65}$ | $98.84_{\pm1.48}$ | $94.58_{\pm1.73}$ | $80.78_{\pm8.15}$ | 95.34 |
| | OpenAI-D | $99.09_{\pm1.02}$ | $99.17_{\pm0.96}$ | $99.08_{\pm1.00}$ | $99.04_{\pm1.03}$ | $98.61_{\pm0.81}$ | $88.19_{\pm1.27}$ | 97.20 |
| | **OpenAI-STK** | $99.45_{\pm0.60}$ | $99.55_{\pm0.52}$ | $99.44_{\pm0.58}$ | $99.43_{\pm0.61}$ | $99.23_{\pm0.49}$ | $89.64_{\pm1.88}$ | 97.79 |
| | MPU | $\mathbf{100.00}_{\pm0.00}$ | $\underline{99.99}_{\pm0.01}$ | $\mathbf{100.00}_{\pm0.00}$ | $\mathbf{100.00}_{\pm0.00}$ | $\mathbf{99.95}_{\pm0.04}$ | $99.55_{\pm0.11}$ | 99.91 |
| | **MPU-STK** | $\mathbf{100.00}_{\pm0.00}$ | $99.98_{\pm0.02}$ | $\underline{99.99}_{\pm0.01}$ | $\mathbf{100.00}_{\pm0.00}$ | $\underline{99.93}_{\pm0.07}$ | $\underline{99.62}_{\pm0.26}$ | $\underline{99.92}$ |
| | RADAR | $\mathbf{100.00}_{\pm0.00}$ | $\mathbf{100.00}_{\pm0.00}$ | $99.96_{\pm0.03}$ | $\underline{99.99}_{\pm0.01}$ | $99.86_{\pm0.04}$ | $\mathbf{99.82}_{\pm0.09}$ | $\mathbf{99.94}$ |
| | **RADAR-STK** | $\mathbf{100.00}_{\pm0.00}$ | $\mathbf{100.00}_{\pm0.00}$ | $99.96_{\pm0.03}$ | $\underline{99.99}_{\pm0.01}$ | $99.85_{\pm0.04}$ | $\mathbf{99.82}_{\pm0.09}$ | $\mathbf{99.94}$ |
| SQuAD1 | Log-Likelihood | $69.62_{\pm0.28}$ | $69.61_{\pm1.06}$ | $67.18_{\pm0.67}$ | $65.67_{\pm0.56}$ | $67.24_{\pm1.10}$ | $85.82_{\pm1.96}$ | 70.86 |
| | Rank | $63.45_{\pm0.91}$ | $64.71_{\pm0.99}$ | $62.30_{\pm0.77}$ | $61.56_{\pm0.80}$ | $64.16_{\pm0.86}$ | $73.52_{\pm1.14}$ | 64.95 |
| | Log-Rank | $68.78_{\pm0.34}$ | $69.52_{\pm0.79}$ | $66.33_{\pm0.59}$ | $64.99_{\pm0.65}$ | $66.83_{\pm1.04}$ | $85.50_{\pm1.97}$ | 70.33 |
| | DetectGPT | $52.60_{\pm2.74}$ | $47.47_{\pm1.42}$ | $52.05_{\pm1.87}$ | $50.88_{\pm2.47}$ | $50.24_{\pm2.20}$ | $51.12_{\pm2.44}$ | 50.73 |
| | F-DetectGPT | $67.91_{\pm0.23}$ | $63.84_{\pm1.73}$ | $64.67_{\pm0.20}$ | $63.36_{\pm0.42}$ | $66.32_{\pm1.36}$ | $79.62_{\pm0.63}$ | 67.62 |
| | ChatGPT-D | $57.10_{\pm1.04}$ | $42.02_{\pm4.36}$ | $55.48_{\pm1.28}$ | $54.90_{\pm1.09}$ | $54.82_{\pm0.91}$ | $65.11_{\pm2.34}$ | 54.90 |
| | **ChatGPT-STK** | $57.87_{\pm1.35}$ | $59.11_{\pm1.82}$ | $55.87_{\pm1.31}$ | $55.28_{\pm1.11}$ | $55.70_{\pm1.53}$ | $66.15_{\pm3.96}$ | 58.33 |
| | OpenAI-D | $\underline{99.34}_{\pm0.45}$ | $\underline{99.39}_{\pm0.34}$ | $\underline{98.57}_{\pm0.81}$ | $95.94_{\pm1.16}$ | $\underline{97.81}_{\pm0.51}$ | $\underline{99.58}_{\pm0.27}$ | $\underline{98.44}$ |
| | **OpenAI-STK** | $\mathbf{99.39}_{\pm0.41}$ | $\mathbf{99.48}_{\pm0.30}$ | $\mathbf{98.61}_{\pm0.83}$ | $\mathbf{96.13}_{\pm1.05}$ | $\mathbf{97.88}_{\pm0.51}$ | $\mathbf{99.63}_{\pm0.22}$ | $\mathbf{98.52}$ |
| | MPU | $98.73_{\pm0.43}$ | $99.00_{\pm0.92}$ | $98.46_{\pm0.77}$ | $95.68_{\pm0.92}$ | $97.43_{\pm0.39}$ | $98.97_{\pm0.48}$ | 98.04 |
| | **MPU-STK** | $98.80_{\pm0.55}$ | $99.01_{\pm0.98}$ | $98.41_{\pm0.77}$ | $95.51_{\pm0.96}$ | $97.16_{\pm0.39}$ | $99.00_{\pm0.53}$ | 97.98 |
| | RADAR | $80.32_{\pm2.05}$ | $79.27_{\pm1.43}$ | $79.72_{\pm1.69}$ | $77.01_{\pm0.57}$ | $77.44_{\pm0.79}$ | $80.57_{\pm2.72}$ | 79.05 |
| | **RADAR-STK** | $81.23_{\pm2.56}$ | $80.21_{\pm2.10}$ | $80.53_{\pm2.46}$ | $77.56_{\pm1.28}$ | $78.69_{\pm1.27}$ | $81.17_{\pm3.01}$ | 79.90 |

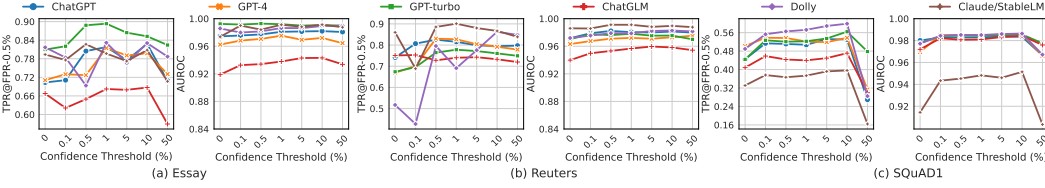

Figure 21: The average detection performance (TPR@FPR-0.5% or AUROC) of OpenAI-STK under different confidence thresholds. We train the detector on various texts (different lines) and report the average performance tested on various LLM texts. An x-coordinate of 0 indicates the original detector OpenAI-D.

## H.14 COMPUTATION EFFICIENCY EVALUATION

In this section, we will discuss the computational efficiency of the framework from the perspectives of runtime and memory analysis.

First, Table 19 compares the running time (training time and inference time) between the original detectors and the enhanced versions, fine-tuned for 5 epochs. The results are consistent with the complexity analysis discussed in Section F.6, indicating that the actual running time of the pro-

Table 11: Performance concerning TPR@FPR-0.5%. Detectors are trained on Dolly texts.

| Dataset | Method | ChatGPT | GPT-4 | GPT-turbo | ChatGLM | Dolly | Claude/StableLM | Avg. |
|---|---|---|---|---|---|---|---|---|
| Essay | Log-Likelihood | $24.08_{\pm22.67}$ | $37.70_{\pm30.63}$ | $23.12_{\pm24.38}$ | $5.86_{\pm7.02}$ | $12.45_{\pm12.04}$ | $2.48_{\pm3.06}$ | 17.62 |
| | Rank | $55.60_{\pm3.65}$ | $51.31_{\pm5.91}$ | $65.84_{\pm6.93}$ | $53.49_{\pm6.09}$ | $35.11_{\pm4.81}$ | $25.12_{\pm5.07}$ | 47.75 |
| | Log-Rank | $28.72_{\pm26.90}$ | $46.48_{\pm36.20}$ | $25.04_{\pm26.83}$ | $28.19_{\pm27.90}$ | $17.34_{\pm15.65}$ | $2.96_{\pm3.73}$ | 24.79 |
| | DetectGPT | $37.04_{\pm10.21}$ | $24.75_{\pm10.03}$ | $5.52_{\pm1.93}$ | $21.45_{\pm14.25}$ | $15.45_{\pm7.05}$ | $4.48_{\pm2.57}$ | 18.12 |
| | F-DetectGPT | $4.24_{\pm1.51}$ | $3.85_{\pm2.05}$ | $31.28_{\pm3.18}$ | $35.74_{\pm3.97}$ | $0.00_{\pm0.00}$ | $0.16_{\pm0.20}$ | 12.55 |
| | ChatGPT-D | $82.08_{\pm5.30}$ | $83.28_{\pm14.28}$ | $32.00_{\pm6.21}$ | $97.75_{\pm2.99}$ | $46.87_{\pm10.12}$ | $1.92_{\pm1.32}$ | 57.32 |
| | **ChatGPT-STK** | $69.44_{\pm15.68}$ | $83.52_{\pm8.05}$ | $26.00_{\pm9.13}$ | $95.58_{\pm5.06}$ | $39.74_{\pm14.81}$ | $3.52_{\pm4.97}$ | 52.97 |
| | OpenAI-D | $92.88_{\pm13.04}$ | $\underline{98.61}_{\pm1.46}$ | $79.36_{\pm16.76}$ | $99.44_{\pm0.60}$ | $85.41_{\pm16.99}$ | $34.40_{\pm18.17}$ | 81.68 |
| | **OpenAI-STK** | $\underline{98.88}_{\pm0.82}$ | $98.52_{\pm1.20}$ | $\underline{79.76}_{\pm8.15}$ | $\underline{99.60}_{\pm0.51}$ | $92.02_{\pm4.04}$ | $29.76_{\pm11.57}$ | 83.09 |
| | MPU | $\mathbf{99.84}_{\pm0.20}$ | $\mathbf{99.67}_{\pm0.31}$ | $42.88_{\pm12.55}$ | $\mathbf{100.00}_{\pm0.00}$ | $\underline{95.02}_{\pm1.40}$ | $51.36_{\pm8.25}$ | 81.46 |
| | **MPU-STK** | $\mathbf{99.84}_{\pm0.20}$ | $\mathbf{99.67}_{\pm0.31}$ | $46.48_{\pm15.65}$ | $\mathbf{100.00}_{\pm0.00}$ | $\mathbf{95.28}_{\pm2.38}$ | $56.88_{\pm7.73}$ | 83.03 |
| | RADAR | $87.20_{\pm5.92}$ | $96.31_{\pm3.07}$ | $79.20_{\pm9.10}$ | $98.15_{\pm1.36}$ | $78.71_{\pm6.05}$ | $\underline{84.48}_{\pm7.31}$ | $\underline{87.34}$ |
| | **RADAR-STK** | $87.92_{\pm5.68}$ | $95.57_{\pm3.04}$ | $\mathbf{80.88}_{\pm9.28}$ | $98.15_{\pm1.41}$ | $78.54_{\pm6.01}$ | $\mathbf{84.72}_{\pm7.49}$ | **87.63** |
| Reuters | Log-Likelihood | $15.76_{\pm31.52}$ | $4.16_{\pm6.53}$ | $16.96_{\pm33.92}$ | $18.88_{\pm37.76}$ | $8.24_{\pm1.94}$ | $2.80_{\pm5.60}$ | 11.13 |
| | Rank | $0.08_{\pm0.16}$ | $0.64_{\pm0.90}$ | $0.00_{\pm0.00}$ | $0.56_{\pm1.12}$ | $2.40_{\pm4.40}$ | $0.00_{\pm0.00}$ | 0.61 |
| | Log-Rank | $0.00_{\pm0.00}$ | $0.80_{\pm0.36}$ | $0.00_{\pm0.00}$ | $0.00_{\pm0.00}$ | $7.04_{\pm2.31}$ | $0.00_{\pm0.00}$ | 1.31 |
| | DetectGPT | $4.40_{\pm2.62}$ | $0.64_{\pm0.54}$ | $2.32_{\pm1.87}$ | $2.56_{\pm2.80}$ | $0.48_{\pm0.47}$ | $3.04_{\pm1.61}$ | 2.24 |
| | F-DetectGPT | $48.00_{\pm9.48}$ | $6.80_{\pm1.88}$ | $92.96_{\pm1.65}$ | $88.96_{\pm4.80}$ | $0.00_{\pm0.00}$ | $0.48_{\pm0.39}$ | 39.53 |
| | ChatGPT-D | $94.72_{\pm3.88}$ | $92.00_{\pm10.07}$ | $91.52_{\pm3.28}$ | $92.08_{\pm13.06}$ | $73.12_{\pm3.95}$ | $17.52_{\pm9.95}$ | 76.83 |
| | **ChatGPT-STK** | $91.04_{\pm10.00}$ | $88.72_{\pm13.24}$ | $90.72_{\pm8.98}$ | $88.00_{\pm18.04}$ | $70.88_{\pm16.85}$ | $22.32_{\pm13.71}$ | 75.28 |
| | OpenAI-D | $59.84_{\pm48.53}$ | $61.60_{\pm43.63}$ | $59.44_{\pm47.89}$ | $59.44_{\pm48.04}$ | $51.36_{\pm32.85}$ | $18.64_{\pm10.58}$ | 51.72 |
| | **OpenAI-STK** | $60.00_{\pm48.50}$ | $65.52_{\pm41.11}$ | $59.92_{\pm47.62}$ | $59.60_{\pm48.17}$ | $58.56_{\pm33.66}$ | $24.72_{\pm14.98}$ | 54.72 |
| | MPU | $\mathbf{100.00}_{\pm0.00}$ | $\underline{99.68}_{\pm0.16}$ | $\mathbf{99.84}_{\pm0.20}$ | $\underline{99.84}_{\pm0.20}$ | $\mathbf{98.32}_{\pm1.25}$ | $90.72_{\pm1.72}$ | 98.07 |
| | **MPU-STK** | $\mathbf{100.00}_{\pm0.00}$ | $99.28_{\pm0.64}$ | $\underline{99.76}_{\pm0.20}$ | $\underline{99.84}_{\pm0.20}$ | $97.76_{\pm1.38}$ | $93.04_{\pm2.43}$ | 98.28 |
| | RADAR | $\underline{99.92}_{\pm0.16}$ | $\mathbf{100.00}_{\pm0.00}$ | $99.36_{\pm0.48}$ | $\mathbf{99.92}_{\pm0.16}$ | $96.32_{\pm0.93}$ | $\underline{96.80}_{\pm1.75}$ | $\underline{98.72}$ |
| | **RADAR-STK** | $\mathbf{100.00}_{\pm0.00}$ | $\mathbf{100.00}_{\pm0.00}$ | $99.60_{\pm0.44}$ | $\mathbf{99.92}_{\pm0.16}$ | $96.24_{\pm0.86}$ | $\mathbf{97.04}_{\pm1.65}$ | **98.80** |
| SQuAD1 | Log-Likelihood | $1.04_{\pm0.57}$ | $5.00_{\pm2.93}$ | $2.11_{\pm0.29}$ | $1.85_{\pm1.12}$ | $5.93_{\pm3.44}$ | $21.85_{\pm10.14}$ | 6.30 |
| | Rank | $2.43_{\pm0.43}$ | $3.81_{\pm1.04}$ | $1.87_{\pm0.68}$ | $1.97_{\pm0.28}$ | $4.30_{\pm1.01}$ | $7.98_{\pm1.52}$ | 3.73 |
| | Log-Rank | $1.97_{\pm1.19}$ | $4.88_{\pm3.25}$ | $2.34_{\pm0.83}$ | $1.97_{\pm1.35}$ | $4.88_{\pm2.72}$ | $19.77_{\pm11.36}$ | 5.97 |
| | DetectGPT | $0.35_{\pm0.69}$ | $0.36_{\pm0.48}$ | $0.47_{\pm0.44}$ | $0.58_{\pm0.73}$ | $0.58_{\pm0.37}$ | $0.81_{\pm0.28}$ | 0.52 |
| | F-DetectGPT | $4.28_{\pm1.35}$ | $6.90_{\pm1.53}$ | $2.69_{\pm1.41}$ | $2.77_{\pm1.12}$ | $5.35_{\pm2.19}$ | $28.44_{\pm3.93}$ | 8.41 |
| | ChatGPT-D | $1.73_{\pm0.97}$ | $1.55_{\pm0.97}$ | $1.29_{\pm0.68}$ | $0.46_{\pm0.23}$ | $1.05_{\pm0.57}$ | $4.16_{\pm3.17}$ | 1.71 |
| | **ChatGPT-STK** | $1.16_{\pm0.97}$ | $2.98_{\pm1.36}$ | $1.17_{\pm0.83}$ | $0.58_{\pm0.37}$ | $1.05_{\pm0.85}$ | $4.16_{\pm2.60}$ | 1.85 |
| | OpenAI-D | $\mathbf{53.87}_{\pm26.50}$ | $\underline{60.36}_{\pm10.30}$ | $\underline{43.27}_{\pm23.38}$ | $40.58_{\pm20.72}$ | $\mathbf{54.19}_{\pm25.62}$ | $\mathbf{76.42}_{\pm14.26}$ | $\underline{54.78}$ |
| | **OpenAI-STK** | $\underline{53.64}_{\pm26.69}$ | $\mathbf{70.95}_{\pm5.64}$ | $\mathbf{46.08}_{\pm24.36}$ | $\mathbf{40.69}_{\pm21.89}$ | $\underline{51.74}_{\pm22.89}$ | $\underline{76.07}_{\pm16.06}$ | **56.53** |
| | MPU | $43.12_{\pm23.37}$ | $54.17_{\pm31.77}$ | $37.89_{\pm22.45}$ | $32.14_{\pm19.43}$ | $47.33_{\pm28.64}$ | $68.09_{\pm12.98}$ | 47.12 |
| | **MPU-STK** | $44.16_{\pm30.43}$ | $59.05_{\pm27.99}$ | $40.23_{\pm23.59}$ | $32.49_{\pm24.94}$ | $42.21_{\pm27.72}$ | $67.98_{\pm14.68}$ | 47.69 |
| | RADAR | $34.45_{\pm7.07}$ | $31.07_{\pm7.41}$ | $33.10_{\pm3.96}$ | $27.17_{\pm6.57}$ | $27.91_{\pm4.92}$ | $35.72_{\pm8.20}$ | 31.57 |
| | **RADAR-STK** | $36.18_{\pm6.63}$ | $32.74_{\pm7.63}$ | $35.32_{\pm3.89}$ | $28.90_{\pm6.89}$ | $29.53_{\pm4.00}$ | $35.14_{\pm8.28}$ | 32.97 |

posed enhancement framework does not exceed twice that of the original detector. We believe that achieving superior detection enhancement performance in an acceptable time frame is valuable.

Second, regarding memory usage, our stacked framework aims for efficient memory utilization. Specifically, the proposed framework uses the same detection model in both the E-step (filtering) and the M-step (final detection). Therefore, the peak memory usage is primarily determined by the detector model itself and is almost identical to the memory usage of the original benchmark detector, as no new model or parameters are introduced. Table 20 below also empirically validates this, showing that the additional memory introduced by the enhanced framework is negligible.

## H.15 CASE STUDY

We illustrate examples that can be correctly detected by the enhancement framework but not by the original detector, as shown in Figs. 22-26. The proposed enhancement method does not entirely input these difficult samples but identifies and filters the potential human parts (highlighted in green), focusing more on the difficult-to-distinguish sequences, thereby making it easier for the detector to learn. It is worth noting that since there is a lack of ground truths of sentences, we employed a voting mechanism of five detection models (ChatGPT-D, OpenAI-D, MPU, GPTZero, and ZeroGPT) to get the pseudo labels, and the pseudo labels are consistently identified as human-generated. This demonstrates its capability to recognize high-confidence human text sequences.

Table 12: Performance concerning AUROC. Detectors are trained on Claude/StableLM texts.

| Dataset | Method | ChatGPT | GPT-4 | GPT-turbo | ChatGLM | Dolly | Claude/StableLM | Avg. |
|---|---|---|---|---|---|---|---|---|
| Essay | Log-Likelihood | $98.48_{\pm0.43}$ | $96.46_{\pm1.05}$ | $98.85_{\pm0.42}$ | $98.78_{\pm0.15}$ | $90.64_{\pm0.35}$ | $92.16_{\pm0.32}$ | 95.90 |
| | Rank | $92.84_{\pm0.99}$ | $90.93_{\pm0.74}$ | $98.19_{\pm0.37}$ | $81.51_{\pm1.32}$ | $80.45_{\pm1.41}$ | $86.69_{\pm1.06}$ | 88.44 |
| | Log-Rank | $98.64_{\pm0.43}$ | $96.85_{\pm0.89}$ | $98.92_{\pm0.40}$ | $98.94_{\pm0.16}$ | $90.41_{\pm0.50}$ | $91.47_{\pm0.45}$ | 95.87 |
| | DetectGPT | $96.86_{\pm0.97}$ | $95.64_{\pm0.45}$ | $44.40_{\pm1.73}$ | $95.60_{\pm0.62}$ | $92.55_{\pm0.58}$ | $46.36_{\pm0.71}$ | 78.57 |
| | F-DetectGPT | $83.86_{\pm1.52}$ | $84.45_{\pm1.58}$ | $94.12_{\pm0.72}$ | $93.62_{\pm0.73}$ | $55.86_{\pm2.54}$ | $58.96_{\pm1.52}$ | 78.48 |
| | ChatGPT-D | $95.91_{\pm2.81}$ | $96.62_{\pm2.16}$ | $93.96_{\pm4.28}$ | $98.72_{\pm1.11}$ | $86.14_{\pm7.72}$ | $81.87_{\pm14.51}$ | 92.20 |
| | **ChatGPT-STK** | $96.89_{\pm2.07}$ | $97.78_{\pm1.79}$ | $95.41_{\pm4.00}$ | $99.49_{\pm0.42}$ | $89.16_{\pm3.99}$ | $88.89_{\pm9.25}$ | 94.60 |
| | OpenAI-D | $97.02_{\pm3.00}$ | $97.11_{\pm3.91}$ | $\underline{99.89}_{\pm0.13}$ | $98.99_{\pm1.10}$ | $91.43_{\pm9.06}$ | $99.77_{\pm0.19}$ | 97.37 |
| | **OpenAI-STK** | $98.77_{\pm0.50}$ | $99.18_{\pm0.35}$ | $\mathbf{99.93}_{\pm0.08}$ | $99.17_{\pm0.38}$ | $97.74_{\pm0.69}$ | $99.93_{\pm0.07}$ | 99.12 |
| | MPU | $\underline{99.92}_{\pm0.09}$ | $\underline{99.84}_{\pm0.11}$ | $97.68_{\pm0.54}$ | $99.88_{\pm0.10}$ | $96.72_{\pm0.48}$ | $\mathbf{100.00}_{\pm0.00}$ | 99.01 |
| | **MPU-STK** | $\mathbf{99.99}_{\pm0.01}$ | $\mathbf{99.86}_{\pm0.13}$ | $97.87_{\pm0.26}$ | $\mathbf{99.98}_{\pm0.02}$ | $96.46_{\pm0.35}$ | $\mathbf{100.00}_{\pm0.00}$ | 99.03 |
| | RADAR | $99.61_{\pm0.23}$ | $99.76_{\pm0.10}$ | $99.66_{\pm0.26}$ | $\mathbf{99.96}_{\pm0.03}$ | $\mathbf{98.72}_{\pm0.40}$ | $\underline{99.95}_{\pm0.04}$ | 99.61 |
| | **RADAR-STK** | $99.48_{\pm0.30}$ | $99.62_{\pm0.17}$ | $99.43_{\pm0.42}$ | $99.93_{\pm0.05}$ | $\underline{98.52}_{\pm0.27}$ | $99.89_{\pm0.06}$ | 99.48 |
| Reuters | Log-Likelihood | $97.59_{\pm0.37}$ | $74.85_{\pm0.50}$ | $98.54_{\pm0.38}$ | $99.54_{\pm0.22}$ | $60.09_{\pm2.13}$ | $85.66_{\pm1.04}$ | 86.05 |
| | Rank | $84.00_{\pm0.88}$ | $74.12_{\pm1.24}$ | $91.82_{\pm0.60}$ | $70.97_{\pm1.65}$ | $54.88_{\pm2.45}$ | $67.78_{\pm1.14}$ | 73.93 |
| | Log-Rank | $97.86_{\pm0.35}$ | $79.95_{\pm0.29}$ | $98.82_{\pm0.37}$ | $99.65_{\pm0.20}$ | $61.24_{\pm2.15}$ | $84.88_{\pm1.13}$ | 87.07 |
| | DetectGPT | $92.78_{\pm1.34}$ | $85.71_{\pm2.02}$ | $49.24_{\pm1.65}$ | $91.59_{\pm1.67}$ | $83.41_{\pm1.73}$ | $66.59_{\pm2.53}$ | 78.22 |
| | F-DetectGPT | $96.23_{\pm0.19}$ | $66.02_{\pm1.53}$ | $98.85_{\pm0.30}$ | $98.58_{\pm0.35}$ | $40.38_{\pm1.31}$ | $66.16_{\pm2.58}$ | 77.70 |
| | ChatGPT-D | $96.84_{\pm2.58}$ | $95.96_{\pm2.06}$ | $96.66_{\pm2.88}$ | $95.94_{\pm4.73}$ | $85.23_{\pm4.30}$ | $94.56_{\pm5.33}$ | 94.20 |
| | **ChatGPT-STK** | $99.45_{\pm0.43}$ | $97.98_{\pm1.02}$ | $99.53_{\pm0.48}$ | $98.62_{\pm0.76}$ | $84.74_{\pm3.26}$ | $97.98_{\pm1.67}$ | 96.38 |
| | OpenAI-D | $99.87_{\pm0.08}$ | $98.42_{\pm0.73}$ | $99.98_{\pm0.02}$ | $99.72_{\pm0.17}$ | $93.79_{\pm1.90}$ | $99.83_{\pm0.12}$ | 98.60 |
| | **OpenAI-STK** | $\underline{99.99}_{\pm0.01}$ | $99.38_{\pm0.38}$ | $\underline{99.99}_{\pm0.01}$ | $99.68_{\pm0.14}$ | $95.02_{\pm1.77}$ | $\underline{99.86}_{\pm0.07}$ | 98.99 |
| | MPU | $\mathbf{100.00}_{\pm0.00}$ | $99.71_{\pm0.12}$ | $99.96_{\pm0.07}$ | $\underline{99.99}_{\pm0.01}$ | $97.25_{\pm0.76}$ | $\mathbf{100.00}_{\pm0.00}$ | 99.48 |
| | **MPU-STK** | $\mathbf{100.00}_{\pm0.00}$ | $\underline{99.74}_{\pm0.08}$ | $\mathbf{100.00}_{\pm0.01}$ | $\underline{99.99}_{\pm0.01}$ | $97.53_{\pm0.90}$ | $\mathbf{100.00}_{\pm0.00}$ | 99.54 |
| | RADAR | $99.98_{\pm0.01}$ | $\mathbf{99.99}_{\pm0.01}$ | $99.96_{\pm0.02}$ | $\mathbf{100.00}_{\pm0.00}$ | $\underline{99.09}_{\pm0.28}$ | $\mathbf{100.00}_{\pm0.00}$ | 99.84 |
| | **RADAR-STK** | $99.98_{\pm0.01}$ | $\mathbf{99.99}_{\pm0.01}$ | $99.96_{\pm0.03}$ | $\mathbf{100.00}_{\pm0.00}$ | $\underline{99.09}_{\pm0.27}$ | $\mathbf{100.00}_{\pm0.00}$ | 99.84 |
| SQuAD1 | Log-Likelihood | $69.62_{\pm0.28}$ | $69.61_{\pm1.06}$ | $67.18_{\pm0.67}$ | $65.67_{\pm0.56}$ | $67.24_{\pm1.10}$ | $85.82_{\pm1.96}$ | 70.86 |
| | Rank | $63.45_{\pm0.91}$ | $64.71_{\pm0.99}$ | $62.30_{\pm0.77}$ | $61.56_{\pm0.80}$ | $64.16_{\pm0.86}$ | $73.52_{\pm1.14}$ | 64.95 |
| | Log-Rank | $68.78_{\pm0.34}$ | $69.52_{\pm0.79}$ | $66.33_{\pm0.59}$ | $64.99_{\pm0.65}$ | $66.83_{\pm1.04}$ | $85.50_{\pm1.97}$ | 70.33 |
| | DetectGPT | $50.42_{\pm3.75}$ | $50.02_{\pm2.90}$ | $49.98_{\pm2.78}$ | $48.63_{\pm2.23}$ | $51.14_{\pm1.90}$ | $49.48_{\pm2.64}$ | 49.95 |
| | F-DetectGPT | $67.91_{\pm0.23}$ | $63.84_{\pm1.73}$ | $64.67_{\pm0.20}$ | $63.36_{\pm0.42}$ | $66.32_{\pm1.36}$ | $79.62_{\pm0.63}$ | 67.62 |
| | ChatGPT-D | $62.28_{\pm4.56}$ | $47.85_{\pm7.29}$ | $59.93_{\pm3.25}$ | $58.84_{\pm3.58}$ | $60.37_{\pm4.80}$ | $74.56_{\pm7.18}$ | 60.64 |
| | **ChatGPT-STK** | $58.67_{\pm0.90}$ | $60.80_{\pm1.05}$ | $56.43_{\pm0.95}$ | $55.73_{\pm0.80}$ | $56.82_{\pm0.84}$ | $68.54_{\pm1.98}$ | 59.50 |
| | OpenAI-D | $94.38_{\pm1.82}$ | $96.55_{\pm1.12}$ | $94.34_{\pm1.96}$ | $89.07_{\pm2.04}$ | $92.74_{\pm1.26}$ | $97.66_{\pm0.92}$ | 94.13 |
| | **OpenAI-STK** | $95.09_{\pm1.97}$ | $96.69_{\pm1.30}$ | $94.61_{\pm2.54}$ | $89.69_{\pm2.42}$ | $93.14_{\pm2.05}$ | $97.96_{\pm0.93}$ | 94.53 |
| | MPU | $\underline{99.13}_{\pm0.24}$ | $\underline{99.16}_{\pm1.07}$ | $\mathbf{98.70}_{\pm0.71}$ | $93.53_{\pm1.48}$ | $96.60_{\pm0.46}$ | $\underline{99.37}_{\pm0.24}$ | $\underline{97.75}$ |
| | **MPU-STK** | $\mathbf{99.17}_{\pm0.35}$ | $\mathbf{99.57}_{\pm0.33}$ | $\underline{98.66}_{\pm0.75}$ | $94.20_{\pm1.12}$ | $96.66_{\pm0.34}$ | $99.51_{\pm0.19}$ | $\mathbf{97.96}$ |
| | RADAR | $81.75_{\pm2.16}$ | $79.98_{\pm0.95}$ | $80.59_{\pm2.09}$ | $77.20_{\pm0.49}$ | $76.98_{\pm1.58}$ | $81.90_{\pm2.36}$ | 79.73 |
| | **RADAR-STK** | $84.33_{\pm1.45}$ | $82.81_{\pm1.30}$ | $83.57_{\pm1.95}$ | $79.78_{\pm0.54}$ | $80.51_{\pm1.96}$ | $84.23_{\pm2.09}$ | 82.54 |

Besides, Through our case analysis, we have observed several distinctions between human-like and machine-generated text:

- **Sentence structure**. Human-like sequences tend to exhibit relatively short and succinct sentences (e.g., "A randomized control trial found no difference..."). In contrast, machine-generated text often features more complex sentence structures, including frequent subordinate clauses and detailed arguments. For instance: "The use of time is fundamental... However, ... For example, in The Winter's Tale... These visual transitions... In contrast, the Novel conveys... While the reader experiences... This gradual progression... Overall, time in the Novel... The use of time also impacts... For instance, in The Winter's Tale... Similarly, Hermione is reanimated... In the Novel, character development... For example, Emma Woodhouse...".

- **Sentence length**. Human-like sequences generally favor short to medium-length sentences (around 12–15 words). On the contrary, machine-generated text averages around 20–25 words per sentence, and their shorter sentences typically serve as transitions or summaries (e.g., "Overall, time in the Novel is more mimetic than in Drama.").

- **Linguistic style**. Human-like sequences typically employ concise analytical language, whereas machine-generated text tends to adopt a more formal style and frequently incorporates technical terms such as "mimetic", "interiority", "staging", and "pacing".

Table 13: Performance concerning TPR@FPR-0.5%. Detectors are trained on Claude/StableLM texts.

| Dataset | Method | ChatGPT | GPT-4 | GPT-turbo | ChatGLM | Dolly | Claude/StableLM | Avg. |
|---|---|---|---|---|---|---|---|---|
| Essay | Log-Likelihood | $24.08_{\pm22.67}$ | $37.70_{\pm30.63}$ | $23.12_{\pm24.38}$ | $5.86_{\pm7.02}$ | $12.45_{\pm12.04}$ | $2.48_{\pm3.06}$ | 17.62 |
| | Rank | $55.60_{\pm3.65}$ | $51.31_{\pm5.91}$ | $65.84_{\pm6.93}$ | $53.49_{\pm6.09}$ | $35.11_{\pm4.81}$ | $25.12_{\pm5.07}$ | 47.75 |
| | Log-Rank | $28.72_{\pm26.90}$ | $46.48_{\pm36.20}$ | $25.04_{\pm26.83}$ | $28.19_{\pm27.90}$ | $17.34_{\pm15.65}$ | $2.96_{\pm3.73}$ | 24.79 |
| | DetectGPT | $37.04_{\pm10.21}$ | $24.75_{\pm10.03}$ | $5.52_{\pm1.93}$ | $21.45_{\pm14.25}$ | $15.45_{\pm7.05}$ | $4.48_{\pm2.57}$ | 18.12 |
| | F-DetectGPT | $4.24_{\pm1.51}$ | $3.85_{\pm2.05}$ | $31.28_{\pm3.18}$ | $35.74_{\pm3.97}$ | $0.00_{\pm0.00}$ | $0.16_{\pm0.20}$ | 12.55 |
| | ChatGPT-D | $65.44_{\pm18.70}$ | $68.44_{\pm17.30}$ | $59.28_{\pm23.21}$ | $85.70_{\pm12.69}$ | $32.02_{\pm13.80}$ | $26.40_{\pm20.62}$ | 56.21 |
| | **ChatGPT-STK** | $74.48_{\pm8.81}$ | $82.54_{\pm7.64}$ | $63.68_{\pm21.91}$ | $92.69_{\pm3.25}$ | $38.80_{\pm11.53}$ | $31.12_{\pm25.30}$ | 63.89 |
| | OpenAI-D | $69.60_{\pm6.44}$ | $81.72_{\pm6.15}$ | $\underline{97.68}_{\pm2.46}$ | $91.24_{\pm2.66}$ | $43.18_{\pm9.42}$ | $92.88_{\pm8.45}$ | 79.38 |
| | **OpenAI-STK** | $75.84_{\pm9.73}$ | $89.59_{\pm3.65}$ | $\mathbf{97.92}_{\pm2.30}$ | $78.63_{\pm12.05}$ | $64.46_{\pm11.10}$ | $\underline{97.60}_{\pm2.65}$ | 84.01 |
| | MPU | $\underline{99.60}_{\pm0.62}$ | $\underline{96.64}_{\pm1.45}$ | $31.68_{\pm12.80}$ | $\underline{97.43}_{\pm3.19}$ | $\underline{71.76}_{\pm2.48}$ | $\mathbf{100.00}_{\pm0.00}$ | 82.85 |
| | **MPU-STK** | $\mathbf{99.92}_{\pm0.16}$ | $\mathbf{98.20}_{\pm0.61}$ | $52.72_{\pm15.55}$ | $\mathbf{99.28}_{\pm0.53}$ | $72.53_{\pm5.33}$ | $\mathbf{100.00}_{\pm0.00}$ | **87.11** |
| | RADAR | $84.16_{\pm10.35}$ | $92.05_{\pm3.28}$ | $84.64_{\pm12.36}$ | $97.35_{\pm2.36}$ | $66.87_{\pm11.34}$ | $96.56_{\pm3.29}$ | $\underline{86.94}$ |
| | **RADAR-STK** | $79.60_{\pm11.00}$ | $87.70_{\pm6.06}$ | $78.64_{\pm12.45}$ | $96.63_{\pm2.10}$ | $60.94_{\pm11.00}$ | $94.16_{\pm3.81}$ | 82.95 |
| Reuters | Log-Likelihood | $77.84_{\pm5.19}$ | $14.88_{\pm5.98}$ | $86.08_{\pm3.38}$ | $93.76_{\pm2.03}$ | $11.20_{\pm4.45}$ | $15.04_{\pm6.86}$ | 49.80 |
| | Rank | $48.88_{\pm1.59}$ | $35.92_{\pm2.88}$ | $58.40_{\pm3.94}$ | $40.56_{\pm1.85}$ | $18.56_{\pm2.27}$ | $6.24_{\pm1.87}$ | 34.76 |
| | Log-Rank | $82.40_{\pm5.24}$ | $25.92_{\pm7.08}$ | $90.96_{\pm4.12}$ | $96.80_{\pm0.88}$ | $14.00_{\pm4.82}$ | $17.60_{\pm8.29}$ | 54.61 |
| | DetectGPT | $4.40_{\pm2.62}$ | $0.64_{\pm0.54}$ | $2.32_{\pm1.87}$ | $2.56_{\pm2.80}$ | $0.48_{\pm0.47}$ | $3.04_{\pm1.61}$ | 2.24 |
| | F-DetectGPT | $48.00_{\pm9.48}$ | $6.80_{\pm1.88}$ | $92.96_{\pm1.65}$ | $88.96_{\pm4.80}$ | $0.00_{\pm0.00}$ | $0.48_{\pm0.39}$ | 39.53 |
| | ChatGPT-D | $54.08_{\pm42.25}$ | $53.84_{\pm35.75}$ | $53.60_{\pm42.42}$ | $55.44_{\pm44.73}$ | $30.56_{\pm10.00}$ | $36.80_{\pm27.02}$ | 47.39 |
| | **ChatGPT-STK** | $68.00_{\pm36.41}$ | $67.52_{\pm28.17}$ | $69.52_{\pm37.70}$ | $62.80_{\pm41.59}$ | $39.44_{\pm9.68}$ | $63.76_{\pm17.12}$ | 61.84 |
| | OpenAI-D | $97.60_{\pm1.68}$ | $75.92_{\pm6.09}$ | $99.36_{\pm0.41}$ | $98.00_{\pm0.67}$ | $45.68_{\pm5.87}$ | $\underline{99.44}_{\pm0.78}$ | 86.00 |
| | **OpenAI-STK** | $99.36_{\pm0.60}$ | $87.52_{\pm10.37}$ | $\mathbf{99.84}_{\pm0.20}$ | $98.00_{\pm0.51}$ | $47.92_{\pm9.88}$ | $96.40_{\pm2.90}$ | 88.17 |
| | MPU | $\underline{99.84}_{\pm0.32}$ | $92.32_{\pm8.06}$ | $98.48_{\pm3.04}$ | $99.52_{\pm0.16}$ | $68.72_{\pm13.13}$ | $\mathbf{100.00}_{\pm0.00}$ | 93.15 |
| | **MPU-STK** | $\mathbf{100.00}_{\pm0.00}$ | $94.40_{\pm5.72}$ | $\underline{99.68}_{\pm0.64}$ | $\underline{99.60}_{\pm0.00}$ | $72.48_{\pm11.30}$ | $\mathbf{100.00}_{\pm0.00}$ | 94.36 |
| | RADAR | $99.44_{\pm0.48}$ | $\mathbf{99.76}_{\pm0.32}$ | $98.72_{\pm0.89}$ | $\mathbf{100.00}_{\pm0.00}$ | $\underline{87.44}_{\pm3.16}$ | $\mathbf{100.00}_{\pm0.00}$ | $\underline{97.56}$ |
| | **RADAR-STK** | $99.44_{\pm0.48}$ | $\mathbf{99.76}_{\pm0.32}$ | $98.80_{\pm0.98}$ | $\mathbf{100.00}_{\pm0.00}$ | $87.68_{\pm3.07}$ | $\mathbf{100.00}_{\pm0.00}$ | **97.61** |
| SQuAD1 | Log-Likelihood | $1.04_{\pm0.57}$ | $5.00_{\pm2.93}$ | $2.11_{\pm0.29}$ | $1.85_{\pm1.12}$ | $5.93_{\pm3.44}$ | $21.85_{\pm10.14}$ | 6.30 |
| | Rank | $2.43_{\pm0.43}$ | $3.81_{\pm1.04}$ | $1.87_{\pm0.68}$ | $1.97_{\pm0.28}$ | $4.30_{\pm1.01}$ | $7.98_{\pm1.52}$ | 3.73 |
| | Log-Rank | $1.97_{\pm1.19}$ | $4.88_{\pm3.25}$ | $2.34_{\pm0.83}$ | $1.97_{\pm1.35}$ | $4.88_{\pm2.72}$ | $19.77_{\pm11.36}$ | 5.97 |
| | DetectGPT | $0.46_{\pm0.67}$ | $0.48_{\pm0.58}$ | $0.58_{\pm0.52}$ | $0.35_{\pm0.46}$ | $0.81_{\pm0.59}$ | $0.58_{\pm0.00}$ | 0.54 |
| | F-DetectGPT | $4.28_{\pm1.35}$ | $6.90_{\pm1.53}$ | $2.69_{\pm1.41}$ | $2.77_{\pm1.12}$ | $5.35_{\pm2.19}$ | $28.44_{\pm3.93}$ | 8.41 |
| | ChatGPT-D | $1.62_{\pm0.57}$ | $2.50_{\pm0.79}$ | $1.40_{\pm0.79}$ | $0.92_{\pm0.46}$ | $1.63_{\pm0.77}$ | $15.72_{\pm10.57}$ | 3.97 |
| | **ChatGPT-STK** | $1.39_{\pm1.35}$ | $4.76_{\pm2.10}$ | $1.40_{\pm0.29}$ | $0.69_{\pm0.57}$ | $1.16_{\pm0.97}$ | $4.97_{\pm3.07}$ | 2.40 |
| | OpenAI-D | $30.17_{\pm13.82}$ | $41.90_{\pm13.95}$ | $25.85_{\pm20.68}$ | $22.31_{\pm10.85}$ | $29.53_{\pm6.08}$ | $65.55_{\pm6.79}$ | 35.89 |
| | **OpenAI-STK** | $30.40_{\pm14.41}$ | $43.45_{\pm14.41}$ | $23.98_{\pm16.43}$ | $21.16_{\pm8.81}$ | $31.98_{\pm7.23}$ | $69.02_{\pm7.46}$ | 36.66 |
| | MPU | $\underline{51.33}_{\pm28.22}$ | $\underline{68.10}_{\pm20.61}$ | $\mathbf{46.32}_{\pm25.93}$ | $\underline{36.07}_{\pm24.10}$ | $\underline{64.07}_{\pm22.40}$ | $\underline{78.38}_{\pm9.28}$ | 57.38 |
| | **MPU-STK** | $\mathbf{56.88}_{\pm29.08}$ | $\mathbf{69.52}_{\pm20.85}$ | $\underline{42.34}_{\pm24.31}$ | $\mathbf{40.58}_{\pm27.05}$ | $65.00_{\pm29.61}$ | $78.96_{\pm7.64}$ | **58.88** |
| | RADAR | $32.25_{\pm6.87}$ | $26.19_{\pm9.21}$ | $29.12_{\pm4.17}$ | $23.58_{\pm6.11}$ | $25.70_{\pm5.11}$ | $35.26_{\pm5.79}$ | 28.68 |
| | **RADAR-STK** | $37.34_{\pm6.02}$ | $30.95_{\pm8.19}$ | $34.04_{\pm2.26}$ | $27.98_{\pm5.58}$ | $29.53_{\pm6.44}$ | $38.96_{\pm5.02}$ | 33.13 |

Table 14: Performance concerning AUROC on Chinese texts and code texts. Detectors are trained on GPT3.5 (on Chinese texts) and ChatGPT (on code texts).

| Method | Chinese Text | | | | | Code Text |
|---|---|---|---|---|---|---|
| | GPT3.5 | Baichuan | ChatGLM | Qwen | Avg. | ChatGPT |
| ChatGPT-D | $92.45_{\pm2.18}$ | $94.60_{\pm2.15}$ | $76.70_{\pm5.47}$ | $92.91_{\pm2.82}$ | 89.16 | $99.35_{\pm0.46}$ |
| **ChatGPT-STK** | $94.12_{\pm2.15}$ | $95.58_{\pm1.96}$ | $80.11_{\pm2.93}$ | $94.16_{\pm1.54}$ | 91.00 | $99.63_{\pm0.27}$ |
| OpenAI-D | $92.75_{\pm1.33}$ | $96.57_{\pm1.11}$ | $87.51_{\pm1.71}$ | $93.83_{\pm1.98}$ | 92.66 | $99.50_{\pm0.51}$ |
| **OpenAI-STK** | $95.76_{\pm0.63}$ | $97.93_{\pm0.85}$ | $88.55_{\pm2.00}$ | $95.04_{\pm1.36}$ | 94.32 | $99.53_{\pm0.70}$ |
| MPU | $96.22_{\pm0.64}$ | $98.08_{\pm0.49}$ | $80.12_{\pm1.87}$ | $95.38_{\pm1.26}$ | 92.45 | $99.80_{\pm0.14}$ |
| **MPU-STK** | $96.02_{\pm0.75}$ | $97.44_{\pm1.03}$ | $83.77_{\pm1.75}$ | $95.30_{\pm1.01}$ | 93.13 | $99.90_{\pm0.08}$ |
| RADAR | $92.43_{\pm1.54}$ | $93.26_{\pm0.55}$ | $85.41_{\pm1.97}$ | $87.55_{\pm0.98}$ | 89.66 | $99.56_{\pm0.46}$ |
| **RADAR-STK** | $92.65_{\pm1.26}$ | $93.28_{\pm0.68}$ | $85.48_{\pm1.38}$ | $88.51_{\pm0.98}$ | 89.98 | $99.64_{\pm0.39}$ |

Table 15: Expected cost in Essay dataset. Detectors are trained on ChatGPT text.

| Method | 10% | 20% | 30% | 40% | 50% |
|---|---|---|---|---|---|
| ChatGPT-D | 0.0635 | 0.0723 | 0.0939 | 0.0951 | 0.1070 |
| **ChatGPT-STK** | 0.0450 | 0.0529 | 0.0574 | 0.0517 | 0.0550 |
| OpenAI-D | 0.0561 | 0.0497 | 0.0490 | 0.0493 | 0.0430 |
| **OpenAI-STK** | 0.0450 | 0.0400 | 0.0406 | 0.0348 | 0.0330 |
| MPU | **0.0450** | 0.0400 | 0.0350 | 0.0300 | 0.0250 |
| **MPU-STK** | 0.0450 | 0.0400 | 0.0350 | 0.0300 | 0.0250 |
| RADAR | 0.0524 | 0.0561 | 0.0546 | 0.0637 | 0.0630 |
| **RADAR-STK** | 0.0450 | 0.0400 | 0.0462 | 0.0469 | 0.0450 |

Table 16: Detection performance (AUROC) on Essay dataset when the text length is at most 64. Detectors are trained on ChatGPT texts.

| Method | ChatGPT | GPT4All | ChatGPT-turbo | ChatGLM | Dolly | Claude | Avg. |
|---|---|---|---|---|---|---|---|
| ChatGPT-D | $89.94_{\pm1.13}$ | $86.29_{\pm2.93}$ | $85.13_{\pm2.60}$ | $96.66_{\pm0.99}$ | $70.23_{\pm2.41}$ | $64.79_{\pm3.25}$ | 82.17 |
| **ChatGPT-STK** | $91.33_{\pm1.01}$ | $88.17_{\pm3.78}$ | $87.69_{\pm0.53}$ | $97.15_{\pm0.59}$ | $75.16_{\pm2.71}$ | $69.44_{\pm5.13}$ | 84.82 |
| OpenAI-D | $93.50_{\pm1.22}$ | $93.10_{\pm1.94}$ | $89.93_{\pm1.52}$ | $98.09_{\pm0.33}$ | $85.36_{\pm1.86}$ | $72.74_{\pm2.85}$ | 88.79 |
| **OpenAI-STK** | $95.68_{\pm0.49}$ | $94.21_{\pm1.13}$ | $93.69_{\pm0.80}$ | $98.59_{\pm0.40}$ | $88.84_{\pm0.60}$ | $81.67_{\pm2.05}$ | 92.11 |
| MPU | $96.28_{\pm0.42}$ | $95.77_{\pm0.37}$ | $96.01_{\pm0.81}$ | $99.24_{\pm0.14}$ | $81.00_{\pm1.07}$ | $85.57_{\pm0.67}$ | 92.32 |
| **MPU-STK** | $96.13_{\pm0.27}$ | $95.66_{\pm0.40}$ | $95.81_{\pm0.80}$ | $99.29_{\pm0.12}$ | $83.38_{\pm1.43}$ | $86.11_{\pm0.57}$ | 92.72 |
| RADAR | $98.63_{\pm0.17}$ | $96.25_{\pm0.81}$ | $97.87_{\pm0.40}$ | $99.30_{\pm0.11}$ | $92.29_{\pm0.95}$ | $93.31_{\pm0.84}$ | 96.28 |
| **RADAR-STK** | $98.77_{\pm0.17}$ | $96.39_{\pm0.64}$ | $98.07_{\pm0.44}$ | $99.38_{\pm0.05}$ | $92.58_{\pm1.02}$ | $93.63_{\pm1.10}$ | 96.47 |

Table 17: Performance concerning Accuracy. Detectors are trained on ChatGPT texts.

| Dataset | Method | ChatGPT | GPT4All | GPT-turbo | ChatGLM | Dolly | Claude/StableLM | Avg. |
|---|---|---|---|---|---|---|---|---|
| Essay | Log-Likelihood | $95.16_{\pm0.41}$ | $91.80_{\pm0.91}$ | $\underline{97.24}_{\pm0.29}$ | $97.39_{\pm0.18}$ | $79.40_{\pm1.06}$ | $77.80_{\pm1.52}$ | 89.80 |
| | Rank | $85.84_{\pm1.12}$ | $84.30_{\pm1.09}$ | $91.36_{\pm1.27}$ | $80.00_{\pm1.02}$ | $75.71_{\pm1.18}$ | $78.04_{\pm1.43}$ | 82.54 |
| | Log-Rank | $95.68_{\pm0.27}$ | $93.28_{\pm0.81}$ | $\mathbf{97.28}_{\pm0.10}$ | $\underline{97.63}_{\pm0.45}$ | $79.91_{\pm0.84}$ | $77.12_{\pm1.49}$ | 90.15 |
| | DetectGPT | $92.08_{\pm1.58}$ | $89.06_{\pm0.46}$ | $50.76_{\pm1.30}$ | $90.04_{\pm1.42}$ | $82.96_{\pm1.77}$ | $51.92_{\pm0.43}$ | 76.14 |
| | F-DetectGPT | $0.84_{\pm0.29}$ | $1.11_{\pm0.31}$ | $0.72_{\pm0.27}$ | $1.24_{\pm0.53}$ | $2.10_{\pm0.32}$ | $1.64_{\pm0.51}$ | 1.28 |
| | ChatGPT-D | $90.00_{\pm3.63}$ | $89.39_{\pm4.07}$ | $83.56_{\pm2.98}$ | $90.76_{\pm3.48}$ | $79.27_{\pm5.40}$ | $51.04_{\pm4.02}$ | 80.67 |
| | **ChatGPT-STK** | $91.12_{\pm3.20}$ | $90.90_{\pm3.94}$ | $85.52_{\pm2.20}$ | $91.89_{\pm3.29}$ | $82.70_{\pm5.81}$ | $57.12_{\pm5.54}$ | 83.21 |
| | OpenAI-D | $94.88_{\pm1.07}$ | $94.88_{\pm0.45}$ | $94.40_{\pm0.90}$ | $95.38_{\pm0.87}$ | $\underline{93.09}_{\pm1.27}$ | $78.56_{\pm3.28}$ | 91.86 |
| | **OpenAI-STK** | $95.24_{\pm0.83}$ | $95.41_{\pm1.21}$ | $94.76_{\pm0.43}$ | $96.02_{\pm0.91}$ | $\mathbf{93.13}_{\pm1.75}$ | $81.28_{\pm2.75}$ | 92.64 |
| | MPU | $\underline{96.44}_{\pm2.30}$ | $96.56_{\pm2.79}$ | $93.88_{\pm1.49}$ | $97.39_{\pm2.44}$ | $92.02_{\pm1.11}$ | $\underline{92.56}_{\pm1.34}$ | $\underline{94.81}$ |
| | **MPU-STK** | $\mathbf{96.48}_{\pm0.82}$ | $\mathbf{96.68}_{\pm1.36}$ | $94.24_{\pm0.61}$ | $\mathbf{97.75}_{\pm0.89}$ | $92.75_{\pm0.32}$ | $\mathbf{95.60}_{\pm1.18}$ | **95.58** |
| | RADAR | $91.80_{\pm2.52}$ | $92.83_{\pm2.73}$ | $91.80_{\pm2.52}$ | $92.77_{\pm2.87}$ | $89.96_{\pm1.46}$ | $90.60_{\pm1.95}$ | 91.63 |
| | **RADAR-STK** | $93.76_{\pm1.46}$ | $94.43_{\pm1.44}$ | $93.76_{\pm1.46}$ | $94.54_{\pm1.71}$ | $91.12_{\pm1.46}$ | $92.12_{\pm0.89}$ | 93.29 |
| Reuters | Log-Likelihood | $91.72_{\pm0.98}$ | $66.60_{\pm1.60}$ | $94.20_{\pm0.69}$ | $95.20_{\pm1.41}$ | $59.88_{\pm1.68}$ | $71.36_{\pm1.41}$ | 79.83 |
| | Rank | $73.80_{\pm0.84}$ | $68.96_{\pm1.11}$ | $80.44_{\pm0.82}$ | $67.92_{\pm0.55}$ | $58.08_{\pm1.37}$ | $62.36_{\pm1.73}$ | 68.59 |
| | Log-Rank | $93.24_{\pm0.74}$ | $72.24_{\pm1.39}$ | $95.08_{\pm0.65}$ | $96.36_{\pm1.13}$ | $60.84_{\pm0.93}$ | $69.64_{\pm1.45}$ | 81.23 |
| | DetectGPT | $88.36_{\pm1.68}$ | $70.68_{\pm1.71}$ | $48.88_{\pm1.26}$ | $86.56_{\pm0.98}$ | $66.40_{\pm2.64}$ | $63.84_{\pm2.24}$ | 70.79 |
| | F-DetectGPT | $1.68_{\pm0.57}$ | $4.00_{\pm0.74}$ | $1.72_{\pm0.56}$ | $1.96_{\pm0.64}$ | $9.96_{\pm1.35}$ | $2.60_{\pm0.72}$ | 3.65 |
| | ChatGPT-D | $96.20_{\pm4.11}$ | $95.28_{\pm3.89}$ | $95.56_{\pm4.00}$ | $96.00_{\pm4.03}$ | $82.68_{\pm5.53}$ | $56.88_{\pm9.25}$ | 87.10 |
| | **ChatGPT-STK** | $95.08_{\pm2.01}$ | $94.52_{\pm2.03}$ | $95.12_{\pm2.03}$ | $94.68_{\pm2.01}$ | $86.92_{\pm3.59}$ | $73.12_{\pm5.73}$ | 89.91 |
| | OpenAI-D | $95.48_{\pm0.88}$ | $95.12_{\pm0.93}$ | $95.48_{\pm0.88}$ | $95.08_{\pm0.90}$ | $90.48_{\pm1.09}$ | $75.12_{\pm0.95}$ | 91.13 |
| | **OpenAI-STK** | $97.56_{\pm0.51}$ | $97.04_{\pm0.64}$ | $97.56_{\pm0.51}$ | $97.00_{\pm0.64}$ | $91.28_{\pm0.57}$ | $80.48_{\pm0.68}$ | 93.49 |
| | MPU | $\underline{98.12}_{\pm0.60}$ | $97.68_{\pm0.45}$ | $\underline{98.08}_{\pm0.68}$ | $98.00_{\pm0.54}$ | $90.80_{\pm0.81}$ | $93.96_{\pm0.98}$ | 96.11 |
| | **MPU-STK** | $\mathbf{98.44}_{\pm0.81}$ | $\mathbf{98.00}_{\pm0.77}$ | $\mathbf{98.44}_{\pm0.81}$ | $\mathbf{98.32}_{\pm0.86}$ | $91.48_{\pm1.29}$ | $\underline{96.44}_{\pm1.02}$ | 96.85 |
| | RADAR | $97.84_{\pm1.39}$ | $\underline{97.84}_{\pm1.39}$ | $97.80_{\pm1.34}$ | $97.80_{\pm1.41}$ | $\mathbf{96.00}_{\pm0.62}$ | $97.40_{\pm1.23}$ | **97.45** |
| | **RADAR-STK** | $97.84_{\pm1.39}$ | $\underline{97.84}_{\pm1.39}$ | $97.80_{\pm1.34}$ | $97.80_{\pm1.41}$ | $95.96_{\pm0.65}$ | $97.40_{\pm1.23}$ | $\underline{97.44}$ |
| SQuAD1 | Log-Likelihood | $63.12_{\pm1.38}$ | $63.39_{\pm2.15}$ | $60.94_{\pm1.06}$ | $60.29_{\pm1.23}$ | $61.16_{\pm1.42}$ | $73.93_{\pm1.89}$ | 63.81 |
| | Rank | $58.67_{\pm0.68}$ | $60.36_{\pm1.93}$ | $57.89_{\pm1.21}$ | $57.51_{\pm0.88}$ | $59.30_{\pm1.12}$ | $65.32_{\pm1.17}$ | 59.84 |
| | Log-Rank | $63.29_{\pm0.98}$ | $63.69_{\pm1.61}$ | $60.18_{\pm0.88}$ | $60.29_{\pm0.98}$ | $60.29_{\pm0.98}$ | $73.87_{\pm1.73}$ | 63.60 |
| | DetectGPT | $52.43_{\pm2.58}$ | $48.81_{\pm1.37}$ | $50.76_{\pm2.16}$ | $50.23_{\pm2.90}$ | $49.77_{\pm1.29}$ | $50.40_{\pm1.83}$ | 50.40 |
| | F-DetectGPT | $4.91_{\pm1.44}$ | $5.71_{\pm0.74}$ | $6.20_{\pm1.04}$ | $6.42_{\pm2.06}$ | $5.58_{\pm1.08}$ | $3.41_{\pm0.81}$ | 5.37 |
| | ChatGPT-D | $53.01_{\pm1.41}$ | $46.37_{\pm2.25}$ | $51.99_{\pm1.79}$ | $52.49_{\pm1.45}$ | $52.21_{\pm1.73}$ | $56.71_{\pm2.31}$ | 52.13 |
| | **ChatGPT-STK** | $54.51_{\pm0.47}$ | $55.00_{\pm1.91}$ | $53.04_{\pm1.53}$ | $53.12_{\pm1.07}$ | $53.31_{\pm1.23}$ | $57.69_{\pm0.81}$ | 54.45 |
| | OpenAI-D | $96.99_{\pm0.60}$ | $97.02_{\pm0.60}$ | $96.90_{\pm1.26}$ | $89.65_{\pm1.41}$ | $92.67_{\pm2.12}$ | $97.28_{\pm1.01}$ | 95.09 |
| | **OpenAI-STK** | $97.28_{\pm0.43}$ | $97.50_{\pm0.67}$ | $97.31_{\pm1.07}$ | $88.90_{\pm0.83}$ | $92.44_{\pm2.01}$ | $\mathbf{97.98}_{\pm0.73}$ | $\underline{95.24}$ |
| | MPU | $\mathbf{97.69}_{\pm0.58}$ | $\mathbf{98.39}_{\pm0.85}$ | $\underline{97.60}_{\pm0.62}$ | $91.21_{\pm1.87}$ | $95.29_{\pm0.72}$ | $\underline{97.63}_{\pm0.53}$ | **96.30** |
| | **MPU-STK** | $\underline{97.69}_{\pm0.75}$ | $\mathbf{98.39}_{\pm1.02}$ | $\mathbf{97.66}_{\pm0.83}$ | $91.21_{\pm1.63}$ | $95.29_{\pm1.08}$ | $97.57_{\pm0.62}$ | **96.30** |
| | RADAR | $70.35_{\pm1.47}$ | $69.40_{\pm1.07}$ | $69.88_{\pm1.42}$ | $67.86_{\pm0.87}$ | $67.50_{\pm1.53}$ | $70.12_{\pm1.46}$ | 69.19 |
| | **RADAR-STK** | $71.56_{\pm1.74}$ | $70.06_{\pm0.91}$ | $71.17_{\pm0.71}$ | $68.55_{\pm0.64}$ | $68.14_{\pm1.45}$ | $70.92_{\pm2.18}$ | 70.07 |

Table 18: Enhancements to the AUROC metric for sentence-based detector. Detector are trained on ChatGPT text.

| Dataset | Method | ChatGPT | GPT4All | ChatGPT-turbo | ChatGLM | Dolly | Claude | Avg. |
|---|---|---|---|---|---|---|---|---|
| Essay | SeqXGPT | **100** | 99.99 | 52.18 | 99.95 | 99.91 | 54.67 | 84.45 |
| | **SeqXGPT-STK** | **100** | **100** | **53.17** | **100** | 99.99 | 58.75 | 85.32 |
| Reuters | SeqXGPT | 99.34 | **99.48** | 61 | 99.54 | **99.51** | 60.78 | 86.61 |
| | **SeqXGPT-STK** | **99.44** | 99.25 | **65.69** | 99.75 | 99.3 | **76.85** | 90.05 |
| SQuAD1 | SeqXGPT | 50.96 | 46.89 | 45.62 | 47.17 | 51.18 | 49.99 | 48.64 |
| | **SeqXGPT-STK** | **53.89** | **49.24** | **52.17** | 50.2 | 54.06 | 53.19 | 52.12 |

Table 19: Running time comparison.

| Method | Training Time (s) | | | Inference Time (s) | | |
|---|---|---|---|---|---|---|
| | Essay | Reuters | SQuAD1 | Essay | Reuters | SQuAD1 |
| Log-Likelihood | 42.17 | 56.29 | 20.43 | 19.82 | 22.34 | 11.39 |
| Rank | 68.93 | 89.60 | 28.31 | 32.92 | 44.61 | 15.27 |
| GLTR | 72.04 | 81.32 | 31.69 | 35.13 | 39.32 | 17.98 |
| DetectGPT | 3829.10 | 4356.37 | 442.74 | 1960.20 | 2269.46 | 858.28 |
| Fast-DetectGPT | 51.39 | 66.77 | 20.10 | 113.69 | 126.78 | 39.60 |
| ChatGPT-D | 151.72 | 167.33 | 118.65 | 5.79 | 5.91 | 2.41 |
| OpenAI-D | 156.23 | 169.09 | 129.37 | 5.81 | 5.99 | 2.57 |
| MPU | 172.83 | 189.46 | 141.20 | 5.83 | 5.87 | 2.62 |
| RADAR | 326.79 | 331.63 | 254.16 | 26.54 | 27.03 | 22.49 |
| **ChatGPT-STK** | 269.85 | 252.18 | 139.54 | 10.75 | 10.89 | 5.67 |
| **OpenAI-STK** | 286.79 | 255.36 | 148.24 | 11.17 | 10.92 | 6.08 |
| **MPU-STK** | 301.51 | 289.77 | 151.92 | 11.26 | 11.69 | 6.37 |
| **RADAR-STK** | 587.16 | 590.11 | 530.20 | 45.38 | 46.81 | 40.56 |

Table 20: Memory consumption.

| Method | Essay | Reuters | SQuAD1 |
|---|---|---|---|
| ChatGPT-D/OpenAI-D/MPU | 9.36G | 9.36G | 9.15G |
| **ChatGPT-STK/OpenAI-STK/MPU-STK** | 9.38G | 9.38G | 9.20G |
| RADAR | 10.45G | 10.45G | 10.38G |
| **RADAR-STK** | 10.47G | 10.47G | 10.42G |

**PROMPT:** Write an essay in 300 words to the prompt "healthcare organization and the personal characteristics and abilities they possess. The sources of a manager's power base can be categorized into personal and organizational aspects. Personal sources include leadership skills, knowledge, experience, and the ability to influence through charisma and charm. Organizational sources, on the other hand, involve the authority granted by the position held, the ability to make decisions and influence through rewards and punishments, and the manager's connections within the organization. Consider the various sources of power for healthcare managers and discuss how they can effectively utilize them to achieve organizational goals.."

**LLM RESPONSE:** There are several approaches used to address the issue of truancy within schools. These include parental prosecution, placing an Education Social Worker within schools, home visits, mentoring programs, and rewarding good attendance. The effectiveness of these interventions has been evaluated through various research methodologies, including randomized control trials, qualitative studies, and longitudinal analyses. Parental prosecution involves taking legal action against parents for their child's truancy, including fining or even jailing parents. This approach is controversial, however, and there is little evidence to suggest it effectively reduces truancy. A randomized control trial found no difference in attendance for students whose parents were prosecuted versus those who were not (Maguire, 2010). Qualitative research also found parental prosecution damaged the school-family relationship and parents' trust in the school (Gazeley, 2012). Placing an Education Social Worker within a school had more promising results. A longitudinal study found attendance improved by 12% over 2 years in schools with a social worker, compared to only 3% improvement in schools without this intervention (Williams et al., 2015). Social workers were able to address the root causes of truancy by providing counseling and connecting families to community resources. Students reported feeling more supported and motivated to attend. Three other studies analyzed various interventions. A mentoring program that matched truant students with teacher mentors found attendance increased during the mentoring but dropped again once the program ended (Thompson & Kelly, 2011). The temporary support was inadequate. In contrast, a program providing small rewards for improved attendance, such as movie passes or snack coupons, was effective in a qualitative study. Students said the rewards motivated them to keep attending to continue earning prizes (Parker et al., 2013). Finally, a randomized control trial found that home visits from a truancy officer increased attendance by an average of 5% among participants compared to control students (Donaldson, 2019)...

[The essay would continue on for 1250 words to fully analyze the effectiveness and methodology of the research studies on the various anti-truancy approaches and provide a summary of three additional studies from the literature review].

**PROMPT:** Write an essay in 200 words to the prompt "Compare and contrast the pandemics caused by the Typhoid and COVID-19 viruses, focusing on the cases of Typhoid Mary and Novak Djokovic. How did these two individuals become exposed to the illnesses, and what impact did their infections have on their families and associates? Additionally, discuss the advancements in medical counter-measures and healthcare system preparedness over the past 100 years that have helped in handling pandemics.."

**LLM RESPONSE:** Typhoid fever, caused by the Salmonella typhi bacteria, became a concern in the early 20th century. Typhoid Mary, an infamous case, was a carrier of the disease without exhibiting any symptoms herself. She worked as a cook and unknowingly infected numerous individuals with the bacteria. It was through her lack of proper hygiene practices and contamination of food that Typhoid Mary exposed others to the illness. Her case underscored the importance of public health measures and raised awareness about asymptomatic carriers. On the other hand, in the current era, the COVID-19 pandemic has captured the world's attention. Novak Djokovic, a renowned tennis player, became exposed to the virus during the ill-fated Adria Tour, an exhibition tournament he organized. Despite the rising number of COVID-19 cases and precautions advised by health authorities, the tournament proceeded without sufficient safety measures. Djokovic, along with several other players, contracted the virus, further spreading it within their circles. This incident highlighted the importance of adherence to preventive measures and the role of individuals in avoiding the transmission of the virus. In both cases, Typhoid Mary and Novak Djokovic unintentionally became exposed to the illnesses. Typhoid Mary's lack of awareness about her carrier status resulted in the transmission of typhoid fever, while Djokovic's disregard for public health guidelines led to his infection with COVID-19. The impact of these infections on the families and associates of Typhoid Mary and Novak Djokovic varied significantly. In the case of Typhoid Mary, several people succumbed to the illness, tarnishing her reputation and causing immense suffering to their families. Djokovic's infection, in contrast, led to widespread criticism and backlash, highlighting the potential consequences of one's actions during a pandemic. The family and associates of both individuals were affected by the indirect consequences of their infections. Over the past 100 years, remarkable advancements have been made in medical counter-measures and healthcare system preparedness. The development of vaccines, antibiotics, and antiviral drugs has revolutionized the way pandemics are managed. Additionally, public health systems have become more robust in terms of contact tracing, testing capabilities, and public awareness campaigns. These improvements have contributed to effectively controlling and treating pandemics. In conclusion, comparing and contrasting the pandemics caused by the Typhoid and COVID-19 viruses reveals the different modes of transmission and impacts on individuals and their circles. Typhoid Mary and Novak Djokovic's cases underscore the importance of personal responsibility and adherence to public health guidelines. Furthermore, advancements in medical counter-measures and healthcare system preparedness over the past century have played a pivotal role in mitigating the effects of pandemics and improving global health outcomes. It is crucial to remember the lessons learned from these pandemics and continue to invest in preventive measures and healthcare infrastructure to safeguard humanity from future outbreaks.

**PROMPT:** Write an essay in 300 words to the prompt "Explore the use of contrast and semantic imagery in John Milton's Paradise Lost and William Shakespeare's Othello. How do the authors utilize dark and light imagery to convey meanings and reinforce boundaries between good and evil? Discuss how these works challenge traditional interpretations and showcase the complexity of these semantic symbols.."

**LLM RESPONSE:** Both Charles Dickens' Great Expectations and Alan Warner's Morvern Callar construct social identities through the politics of space and place in imaginative ways. By depicting protagonists navigating fraught social spaces and places in search of self-identity, these novels meditate upon the contemporary dilemma of self-identity in literature and society. In Great Expectations, Pip's imagination is powerfully shaped by his experience of place and space. The novel opens in a churchyard, where Pip encounters the terrifying convict Magwitch. This gothic space fills Pip with dread and establishes a prevailing sense of danger and uncertainty. As Pip is "brought up by hand" by his sister in their home, the claustrophobic and humorless domestic space compounds Pip's discomfort. His imaginative escape is found in his notion of becoming a "gentleman" in London. Pip's expectations of London as a space of opportunity and status are soon dashed. In London, strict rules of etiquette and class boundaries rigidly define social identities in spatial terms. Pip struggles to navigate these spaces, uncomfortably lodging with the eccentric character Herbert Pocket. However, through encounters with the upper-class Estella and Miss Havisham, Pip begins constructing a fantasy of belonging to a higher social class. Morvern Callar similarly portrays its eponymous protagonist navigating the politics of space and place. In her small port town in Scotland, Morvern finds little opportunity to explore her identity. However, after her boyfriend commits suicide, Morvern conceals his death to claim the proceeds from his unpublished novel. She uses this money to escape on holiday to Spain with her friend Lanna. In Spain, Morvern experiments with new identities that would be impossible in her hometown. She pursues romance on her own terms, presenting herself as worldly and enigmatic. She observes, "Nobody knew me. I could be whatever I wanted to be." However, Morvern struggles to integrate these ephemeral holiday identities into a coherent sense of self. Upon returning home, Morvern struggles to reconcile her transformational experiences in Spain with the mundane familiarity of place in Scotland. However, she starts to re-imagine the possibilities afforded by her own locality. The novel ends with Morvern dispersing her boyfriend's ashes in the sea, signalling her reconciliation with place and growing self-assurance. In conclusion, Great Expectations and Morvern Callar are novels profoundly concerned with self-identity, which they explore through the imaginative politics of space and place. By following their protagonists' journeys through fraught spaces and places, both novels suggest how localities can be both confining and liberating. They show how one's sense of identity emerges through the dialectical and transformative relationship between the familiar and unfamiliar in space and place. Overall, these novels present a compelling vision of how self-identity develops through navigating the spaces and places that shape our lives.

Figure 22: MGTs that can be correctly detected by the proposed framework but not by the original detector. The identified human-generated parts are highlighted in green.

**PROMPT:** Write an essay in 600 words to the prompt "Explain the benefits and features of using a front load washing machine. Describe the parts of a front load washer and how they contribute to its efficiency. Provide instructions and guidelines for using a front load washer effectively.."

**LLM RESPONSE:** The use of time is fundamental in structuring Drama and the Novel. However, time serves distinct purposes across these two forms of fiction, enhancing the reader or audience's experience in different ways. In Drama, the use of time is portrayed visually through staging, lighting, costumes, and the physical aging of the actors. These visual elements give the audience an acute sense of the passage of time over the course of the play. For example, in The Winter's Tale, Hermione's 16-year separation from her daughter Perdita is conveyed through her transition from youth to middle age on stage, demonstrated through aging makeup and costuming. The gap in time between Acts 3 and 4, signified by the Chorus, is reinforced by a complete set change. These visual transitions in time, though abrupt, give the play a sweeping, epic quality as whole lifetimes pass over the course of a few hours. In contrast, the Novel conveys the passage of time through descriptive language and pacing. While the reader experiences multiple gaps in time over the course of Emma, time passes more gradually. The four main sections of the novel span two years altogether, but the reader gains a sense of the passing seasons, holidays, and daily rhythms in the village of Highbury. Important events like Frank Churchill and Jane Fairfax's clandestine engagement unfold over months. This gradual progression of time, depicted through subtle cues in narration and description, gives the novel a leisurely pace that reflects the steady and unremarkable rhythm of life in Regency-era England. Overall, time in the Novel is more mimetic than in Drama. The use of time also impacts character development differently across the forms. In Drama, characters visibly age and mature on stage, enabling abrupt transformations. For instance, in The Winter's Tale, the teenage Perdita blossoms into a young woman before the audience's eyes after the gap in time between Acts 3 and 4. Similarly, Hermione is reanimated after 16 years as a living statue, transformed from middle age back to youth. These physical transformations, especially Hermione's de-aging, create a sense of characters emerging from the layers of time to reconcile their past and present selves. In the Novel, character development unfolds more gradually through interiority. For example, Emma Woodhouse matures over two years through a series of humbling insights into her own flaws and misunderstandings. *The reader gains access to Emma's developing self-awareness through free indirect discourse. While Emma's essential spirit remains unchanged, by the end of the novel she has gained a sense of compassion and moral duty that she lacked at its start. Rather than visibly transforming, Emma ripens into her best self through steady growth over time. In conclusion, time serves distinct narrative purposes across Drama and the Novel.* In The Winter's Tale and Emma, time is conveyed visually through staging and descriptive language, respectively, impacting the reader's experience and sense of pacing in each form. Time also enables different trajectories of character development, with abrupt transformations occurring in Drama and gradual evolutions unfolding in the Novel. Ultimately, time shapes storytelling in diverse ways across these two genres of fiction.

**PROMPT:** Write an essay in 500 words to the prompt "What are the effects of Dyslexia on children and how can specialized interventions help improve their performance in school?."

**LLM RESPONSE:** The early states that emerged in mainland Greece during the Bronze Age, from around 2000 BCE to 1200 BCE, displayed several defining features. They were primarily based around fortified palatial centers that were the seats of power for emerging elite classes. The power of these elites was based on control of resources, especially agricultural surplus and trade networks. At the same time, the character and power bases of these states impacted their organization and form. They lacked aspects of centralization that would emerge in later states. Power was concentrated at the local level in the hands of palace elites, and there were limited bureaucratic institutions. The Mycenaean states that ruled mainland Greece were organized around fortified palace centers, with the most prominent at Mycenae, Tiryns and Pylos. These palatial centers served as the seats of power for an emerging elite class. The power and control of resources by these elites allowed the accumulation of wealth and the mobilization of resources required to construct the massive "cyclopean" walls and public works projects for which the Mycenaeans are known. Power and status were based on control of resources, especially agricultural surplus and trade. Linear B tablets provide evidence that the palaces exercised economic control over agricultural land and oversaw the collection and distribution of staple goods. Maritime trade also supported the rise of the Mycenaean elites. However, the Mycenaean states lacked major institutions of centralization that would emerge in later states. Power was concentrated at the local level, in the hands of the palace elites. There is little evidence for Mycenaeanstate-controlled centralized institutions, bureaucracy, or military forces. The palaces appear organized to meet their own local needs without strong connections to a broader state apparatus. Most interactions appear to have occurred at a regional and local level. The Minoan civilization on Crete emerged around 2000 BCE, centering around the palaces at Knossos, Phaistos, and Mallia. As in Mycenaean Greece, power was concentrated in the control of agricultural surplus and trade networks by palace elites. The Minoans relied heavily on maritime trade, exchanging luxury goods with Egypt and the Near East. *Frescoes at Knossos depict the storage of jars, presumably filled with agricultural surplus like grain or olive oil—allowing palace control of these critical resources. Although more centralized than Mycenaean palaces, there is little evidence for strong state institutions in Minoan Crete. Power remained concentrated among the palatial elites, with a reliance on personal contact and loyalty. The growth and fall of the Minoan palaces, and broader shifts in Minoan culture seem regional or tied to particular palatial centers, rather than reflecting a coherent state policy or institutions. The lack of Minoan bureaucracy and records in Linear A script point to limits in political centralization and state institutions in Minoan society. In conclusion, the early Bronze Age states that emerged in Mycenaean Greece and Minoan Crete were characterized by fortified palatial centers as the base of power for an emerging elite class. Control of agricultural surplus and trade networks allowed these elites to amass wealth and political power. However, power remained concentrated locally in the hands of palace elites.* There is little evidence for the bureaucratic institutions or military organizations that would typify later states. Political centralization and state formation were limited, with regional and local dynamics remaining predominant. Socio-political structures were built around personal loyalty, kinship and clientage rather than strong state institutions.

**PROMPT:** Write an essay in 300 words to the prompt "Discuss the potential risks and strategies for mitigating them in the establishment of Centers for Diabetes, focusing on community health needs and the targets set by the Affordable Care Act (ACA).."

**LLM RESPONSE:** Wallaroo Wines should adopt a premium, high-end product differentiation market entry strategy for Hong Kong and mainland China. Given the brand's focus on high-quality premium wines, it should capitalize on Chinese consumers' growing taste for luxury imported wines. A high-end strategy is compelling given Hong Kong's large population of high-income consumers, and the growing upper classes in major mainland cities. For its product strategy, Wallaroo Wines should maintain its focus on high-end red wines like its premium Cabernet Sauvignon and Shiraz varieties. *These wines should be priced at a premium, leveraging their status as imported luxury goods. The labeling and packaging should also convey an upscale premium image to appeal to status-conscious Chinese consumers. In terms of place, Wallaroo Wines should focus distribution in Hong Kong first before expanding to mainland China. Hong Kong provides an ideal test market given its sizable population of wine consumers, lower taxes/tariffs, stronger intellectual property protection, and simpler logistics.* Once established in Hong Kong, Wallaroo can pursue partnerships with prestige importers and distributors in mainland cities like Beijing, Shanghai and Shenzhen. For promotion, Wallaroo should invest heavily in social media, sponsoring events, influencer marketing and traditional media advertising to raise brand awareness and appeal to target consumers. Premium positioning should be reinforced through marketing that conveys heritage, quality and indulgence. Sponsoring prestigious events like art galleries, film festivals or golf tournaments would effectively reach high-end consumers. Working with influencers like luxury lifestyle bloggers or celebrity wine aficionados can also boost brand buzz. Key opportunities in this market entry strategy include tapping into China's fast-growing demand for imported wine, particularly at the luxury end; leveraging Hong Kong as a launch pad; and strengthening brand positioning as a premium lifestyle brand. However, there are also challenges such as intense competition from other imported and domestic wine brands; complex regulatory environments; counterfeiting; and price sensitivity, even among higher-income consumers. Overall, a premium differentiation strategy targeting high-end consumers in Hong Kong and China's major cities can be advantageous for Wallaroo Wines. By focusing on a niche, underserved segment and emphasizing quality and status, Wallaroo can build strong brand positioning that sets it apart in a crowded market. With the right partnerships and marketing, Wallaroo Wines can make substantial inroads into this attractive export market.

Figure 23: MGTs that can be correctly detected by the proposed framework but not by the original detector. The identified human-generated parts are highlighted in green.

**PROMPT:** Write an essay in 300 words to the prompt "Discuss the influence of Charles Bolden on the history of jazz, including his musical innovations, impact on future musicians, and the events and people that shaped his life and career.."

**LLM RESPONSE:** The city of Oxford is home to four major bookstores—Blackwell's, Borders, Waterstone's, and WH Smith—that attract a wide range of customers due to their varying locations, layouts, inventory, and atmospheres. In this report, I will analyze the key attributes and retail strategies of each bookshop to determine how they appeal to different target markets. Blackwell's, located on Broad Street in the heart of Oxford city centre, is the oldest and largest bookshop, occupying multiple floors of an imposing historic building. Its vast selection of over 200,000 new, used, and rare books—especially academic texts and secondary literature—attracts serious readers and students. The multi-level labyrinthine layout of small rooms creates an intimate, almost private browsing experience conducive to serendipitous discoveries. The scholastic ambiance, overlooking the courtyard of Balliol College, appeals to intellectually curious customers seeking a quintessential Oxford book-buying experience. In contrast, Waterstone's on St. Giles Street has a more modern open-plan design spread over two floors. Its front tables feature prominent displays of popular fiction and non-fiction, especially the latest bestsellers, aimed at casual readers and tourists. While also carrying a wide range of books, Waterstone's focuses on highly commercial mainstream titles in an attempt to draw in a larger customer base seeking trendy and accessible reads. Its central location, glass storefront, and sleeker décor give it a hip and contemporary feel that contrasts with the traditional atmosphere of Blackwell's, appealing to younger and more popular audiences. Borders, located in the Clarendon Centre shopping mall, closed down in 2019 due to the pressures of online retail competitors and changing reader trends. When it was open, it carried a range of commercial fiction and non-fiction, especially in popular genres like crime, thrillers, and romance. Its generic big-box layout and chain brand identity attracted deal-seeking customers in search of discounted bestsellers and impulse buys. The mall location suggested an attempt to capture weekend shoppers and families in addition to dedicated readers. However, its formulaic design and inventory failed to establish a distinctive brand identity and loyal customer base in the competitive Oxford book market. Finally, WH Smith on Cornmarket Street focuses on convenience items like stationery, magazines, and entry-level fiction and non-fiction. While also selling a modest selection of books, especially Oxford-themed gift titles aimed at tourists, the dominant range of general merchandise indicates its primary customer base comprises passersby looking for quick impulse purchases and everyday essentials rather than serious book readers. The small ground-floor layout and limited book range lend it the feel of a neighborhood shop meant for practical community use rather than an inviting space for extended browsing or in-depth discovery. In summary, the major bookshops of Oxford have carved out distinct niches by catering to different readers and purposes. Blackwell's and Waterstone's dominate the dedicated book market by offering a choice between traditional or contemporary ambiances and stocking either academic or popular titles. The now-defunct Borders occupied a middle ground as a casual mainstream outlet. And WH Smith serves local needs as a convenient spot for essentials and small gifts rather than substantive book browsing. Through their locations, layouts, inventory selections, and general atmospheres, each store has adopted a retail strategy aimed at matching a particular set of customers and their book-buying motivations.

**PROMPT:** Write an essay in 900 words to the prompt "The Prompt:Discuss the benefits and challenges of the Internet of Things (IoT) in our daily lives, including its impact on efficiency, automation, and privacy concerns.."

**LLM RESPONSE:** Sitcoms often utilize pragmatic frameworks, including Grice's conversational maxims, Brown and Levinson's Politeness theory, and Leech's Politeness Maxims, to achieve the expected humorous effect in their scripts. The popular American sitcom "Friends" is an excellent example of how these pragmatic tools are employed creatively and strategically to elicit audience laughter. Grice's conversational maxims relate to the cooperative principle, whereby conversational contributions should be purposeful, truthful, relevant, and clear. Flouting these maxims in sitcoms can create comedic irony and absurdity. In a scene from "Friends", Monica, Rachel and Phoebe are lambasting their friend Ross about saving a mouse from a glue trap but then accidentally killing it. Ross defends that he "was just trying to be a good friend" to the mouse. Phoebe quips: "Aw, you're like a cute, fuzzy little unintentional kitten killer." This flouts the maxim of relevance, juxtaposing the irrelevant concept of "kitten killer" for comedic effect. The non sequitur also adds to the absurdity and irony, making the audience laugh at Phoebe's exaggerated comparison. Politeness theory focuses on the conflict between two speakers' needs to be efficient and indirect. Character harassment and teasing are common mechanisms by which this conflict elicits humor in sitcoms. In another "Friends" episode, Joey and Chandler harass Ross by singing "I'm Bein' Kind", a song mocking Ross's failed relationships. Although intended as a joke, the singing also flouts the tact maxim by embarrassing Ross and highlighting his romantic inadequacies. The rudeness is softened by the casual, friendly dynamic between the characters, conveying that the insults are said in jest and with the goal of amusing rather than offending Ross – and the audience. Leech's Politeness Maxims relate to tact, generosity, approbation, modesty, agreement, and sympathy. Flouting these maxims in clever or ironic ways is a common source of humor in sitcoms. In one episode, Rachel makes her famous English trifle for Thanksgiving, but the recipe pages get stuck together, creating a disastrous combination of beef and custard. When asked how it tastes, Joey replies: "It tastes like feet!", exploiting the comedy in flouting the tact and approbation maxims. However, because Rachel acknowledges the trifle tastes awful, Joey's comment also reinforces the friendship and honesty between the characters, softening the rudeness and making the audience laugh with Joey rather than at Rachel. In conclusion, the sitcom "Friends" utilizes pragmatic frameworks like Grice's maxims, Politeness theory, and Leech's maxims in innovative ways to craft comedic scenarios and witty dialogue. Flouting these pragmatic principles and manipulating the conflict between efficiency and politeness create situations and interactions that elicit laughs from the audience. A close analysis of "Friends" demonstrates how these tools can be strategically combined for comedic effect.

**PROMPT:** Write an essay in 300 words to the prompt "Essay Discuss the potential benefits of implementing a decentralized parking system using blockchain technology, and outline the business goals and strategies for introducing this innovative solution.."

**LLM RESPONSE:** To properly assess the status of 65-year-old Deirdre after undergoing electro-convulsive therapy for severe depression and risk of relapse, several evaluation steps should be taken by her health professionals. First, a thorough medical and psychiatric history review should be conducted, including discussion of her mental and physical health before and after her husband's death, the severity and symptoms of her depression that warranted ECT treatment, how she responded to the ECT, any side effects, and her current state of mood, cognition, and daily functioning. Standardized depression screening tools, like the Patient Health Questionnaire-9, that Deirdre fills out and clinician-administered scales such as the Hamilton Depression Rating Scale provide quantitative measures of her current depression severity and risk of relapse. Deirdre reported that ECT helped lift her depression but left her with memory gaps and concentration difficulties, common side effects, so cognitive testing is recommended. Simple screening tests include the Mini-Mental State Exam, clock drawing test, and verbal fluency to check for significant cognitive impairment. More in-depth neuropsychological testing may also be needed to identify specific memory, attention, and executive function deficits. These cognitive baselines will help determine if further ECT treatments are suitable and monitor her progress. Discussion about Deirdre's activities of daily living, social interactions, sleep, and eating habits provides insight into her overall wellbeing and recovery. Talking to her close ones, with her consent, can give another perspective on her day-to-day functioning. It is important that Deirdre maintains a routine, sticks to a healthy diet and exercise, and continues social engagement to avoid isolation and support her rehabilitation and remission. Ongoing follow-ups and management are required to monitor for depression recurrence. Deirdre should schedule regular visits with her psychiatrist and therapist, and be aware of potential trigger events that may exacerbate her depression. Medication may be introduced to sustain her mood stability and adjunct therapies like psychotherapy sought if needed. Community support groups can also aid her long-term coping. In summary, a comprehensive assessment and management plan considering Deirdre's medical history, current symptoms, cognitive and functional status, and relapse prevention strategies is key to evaluating her progress, maintaining stability, and optimizing her wellbeing after loss and depression. With the right care and her personal motivation, Deirdre has a good chance of sustained remission despite facing a difficult life event. But vigilance for recurrence and proactive support systems remain vital to minimize future episodes and safeguard her health.

Figure 24: MGTs that can be correctly detected by the proposed framework but not by the original detector. The identified human-generated parts are highlighted in green.

**PROMPT:** Write an essay in 300 words to the prompt "Examine the social factors impacting grocery shopping habits and food accessibility in Forest Park, Illinois, and discuss potential interventions to address these issues."

**LLM RESPONSE:** There has been significant progress in recent decades in understanding the molecular mechanisms that determine cell fate specification in the pregastrulation embryo. Studies in model organisms such as Drosophila, Xenopus, zebrafish, and the soil nematode Caenorhabditis elegans have revealed that cell fate determination relies on the precise regulation of gene expression. The expression of specific transcription factors, signaling pathways, and other key molecules are tightly regulated in space and time, leading to the differentiation of undifferentiated embryonic cells into cells of the mesoderm, endoderm, and ectoderm germ layers. C. elegans provides an excellent model to study the molecular basis of cell fate specification in the early embryo. The C. elegans embryo contains only a few precursor cells that generate all postembryonic cells. The cell lineage is largely invariant between individuals, providing a simple blueprint to study cell fate determination. Powerful molecular tools including RNA interference allow for the systematic perturbation of gene function in the embryo. Embryogenesis in C. elegans is also very rapid, with cell fate decisions made within a few hours of fertilization, facilitating detailed analysis. Studies in C. elegans have identified key transcription factors, regulatory elements, and signaling pathways involved in cell fate determination. For example, the end-1 and end-3 transcription factors are required for endoderm development. The tbx-2 transcription factor determines the fate of mesodermal blastomeres. The Wnt/β-catenin asymmetry pathway generates differences between the anterior and posterior of the embryo that are required to specify ectoderm and endomesoderm. Mutations in these genes result in embryos lacking entire germ layers and tissues. C. elegans also provides temporal resolution to study the order of molecular events in cell fate determination. For instance, Wnt/β-catenin signaling occurs before and is required for the asymmetric expression of end-1/end-3. end-1/end-3 expression then induces downstream targets that execute the endoderm fate. *Using time-lapse fluorescence microscopy, it is possible to visualize the dynamics of these molecular determinants in living embryos with single-cell resolution. Mathematical modeling and computational analysis of these dynamics have provided insights into the robustness and logic underlying cell fate decisions. In summary, significant progress has been made in understanding how transcription factors, signaling pathways, and other molecules specify cell fates in the early embryo before gastrulation. The free-living nematode C. elegans provides a powerful model to study these molecular mechanisms systematically owing to its simple and well-characterized embryogenesis, genetic tractability, and live imaging capabilities.* Continued research in C. elegans and other model organisms promises to yield a comprehensive picture of how the precise regulation of gene expression in space and time determines cell identity in embryonic development.

**PROMPT:** Write an essay in 400 words to the prompt "In Bob Dylan's song Scarlet Town, the interplay between embedded songs helps invoke the overall theme. Explore how Dylan presents a world full of both good and bad and argues that combining the two forms can lead to eventual good. Analyze the distinct features of the town, the narrator's acknowledgement of his imperfections, and the significance of the events and themes presented in the song.."

**LLM RESPONSE:** In his Critique of Judgment, Immanuel Kant argues that reason allows the human mind to grasp concepts that surpass the limits of imagination, particularly in experiencing the mathematical and dynamic sublimes. For Kant, the imagination has a maximum, finite magnitude it can comprehend, based on the maximum size of images it can form. Reason, however, is capable of conceiving of infinitely large magnitudes through a process of successive addition and expansion. This allows reason to exceed imagination. Kant first addresses how we can obtain concepts of extremely large magnitudes that seem to surpass imagination. He argues that we cannot have an image or intuition of infinitely large magnitudes, as imagination has a maximum. We can, however, think infinity through the intellectual idea of successive addition: by representing an object and recognizing we can always add more to its size. This process can continue indefinitely, giving us a concept of infinity that surpasses what we can imagine. *However, this is merely a concept - we cannot have a corresponding sensible intuition. For Kant, this method of measuring magnitude through concepts rather than intuition is the aesthetic way of judging size. It has limitations, as without intuition, we cannot assign a precise magnitude or gain a full sense of the object's size. The aesthetic estimation also cannot be communicated precisely to others.* However, it allows reason to conceive the immeasurable, enabling the experience of the mathematical sublime. Kant holds that we can never have knowledge of an actual infinite as an object. The infinite is not a quantity that we can perceive or grasp totally. However, reason can conceive of the infinite negatively, as that which surpasses any assignable quantity. *For Kant, we form the idea of infinity by realizing any finite quantity we posit can be exceeded. The infinite is a concept of reason alone, though it arises in the attempt to grasp sensible objects aesthetically that seem to surpass imagination. True sublimity for Kant thus refers to the ability of reason to conceive ideas that exceed the imagination, giving us intimations of infinity. The sublime arises when imagination is overcome in grasping an object aesthetically, and reason takes over, forming an idea of absolute greatness not limited to what imagination alone can comprehend.* In this way, the sublime displays the ultimate ability of human reason to transcend sensibility, forming concepts not tethered to the constraints of imagination or experience. The sublime reveals our capacity for formulating rational ideas, even those surpassing intuition. To conclude, Kant argues reason allows us to conceive concepts beyond imagination, especially in experiencing the mathematical and dynamic sublime. While imagination has a maximum, finite magnitude, reason can represent infinity through successive addition. *Estimating magnitudes aesthetically has limits but gives intimations of the immeasurable. For Kant, true sublimity refers to reason conceiving ideas surpassing imagination, giving a sense of infinity that highlights reason's transcendence of sensibility. Overall, the sublime reveals our highest cognitive faculty: the ability to form rational ideas independent of experience.*

**PROMPT:** Write an essay in 600 words to the prompt "What are the thematic elements and stylistic characteristics of blues poetry? Provide examples from the essay to support your answer.."

**LLM RESPONSE:** Judicial review is the process by which the courts review the lawfulness of decisions or actions made by public bodies such as central government departments, local authorities, tribunals, and other decision-making bodies. A claim for judicial review is a legal challenge to the way in which such a decision was made, rather than the merits or content of the actual decision. To bring a claim for judicial review in England and Wales, there are a number of procedural requirements that X would need to fulfil. Firstly, X would need to have sufficient interest in the matter, known as'standing'. Standing is established if X can show that the decision being challenged directly affects them or would affect them differently from the public at large. Given that X was excluded from school by the decision, X would likely be directly affected by the decision and have standing to bring a judicial review claim. Secondly, X would need to act promptly in bringing the claim. Under the Civil Procedure Rules (CPR), claims for judicial review must be filed within 3 months of the grounds for the claim arising. The court has discretion to extend this time limit, but promptness is expected given the public nature of decisions under review. X would thus need to file his claim within 3 months of being notified of the exclusion decision. Thirdly, X would need to apply for permission to proceed with the claim. This requires filing court forms setting out the grounds for review along with evidence to support those grounds. *Permission will be granted if the court considers that X has an arguable case warranting a review. At the permission stage, the court will assess the grounds put forward by X for their potential to succeed. Common grounds for review include procedural unfairness, irrationality/ unreasonableness, illegality, and lack of proportionality. X could potentially argue that the decision to exclude him was disproportionate on the basis that the exclusion was too severe a punishment and not rationally connected to the aims of discipline and good order in the school.* The principle of proportionality requires that decisions impacting individual rights are proportionate to the legitimate aims pursued. The court would consider whether the exclusion was rationally connected to its aims, whether less restrictive measures were available, and whether the impact on X's right to education was excessive. Given the severity of exclusion, X may have grounds to argue disproportionality. In considering Y's involvement in the decision to exclude X, the rule against bias may apply if Y had a close relationship with X or a vested interest in seeing X excluded. The rule against bias requires that decisions are made impartially, by individuals with no pecuniary or personal interest in the outcome. If Y was biased or appeared to act unfairly in recommending X's exclusion, the decision and process may be flawed and unlawful on the grounds of apparent bias. In conclusion, there are clear procedural and substantive grounds on which X could challenge the school's decision to exclude him through judicial review. By acting promptly to file a claim, securing permission to proceed, and arguing that the decision was disproportionate or apparently biased, X could have reasonable prospects of success in his application for judicial review of the school's decision.

Figure 25: MGTs that can be correctly detected by the proposed framework but not by the original detector. The identified human-generated parts are highlighted in green.

**PROMPT:** Write an essay in 300 words to the prompt "To what extent do the works of Dr. Jekyll and Mr. Hyde and Frankenstein convey the message that science can be a tool for violence and lack of integrity? Examine the actions and motivations of the scientists in each novel and explore how their experiments lead to destructive consequences."

**LLM RESPONSE:** Determining whether a theory is scientific or pseudoscientific has been a long-standing challenge in philosophy of science. Several philosophers have proposed demarcation criteria to differentiate science from non-science. Karl Popper proposed the falsification criterion, that for a theory to be scientific it must be falsifiable. Thomas Kuhn proposed the puzzle-solving criterion, that science progresses through paradigm shifts to solve conceptual puzzles. Imre Lakatos proposed hard core theories protected by an auxiliary belt of auxiliary hypotheses. And Paul Thagard examined why astrology fails to meet scientific criteria. Popper's falsification criterion states that for a theory to be scientific, it must be falsifiable - able to be proven false through observations or experiments. According to Popper, pseudosciences like astrology are not falsifiable because they can always be adjusted to fit new evidence. While falsification is an important part of science, it is too simplistic as a demarcation criterion. Many scientific theories are hard to falsify in practice and scientists do not always abandon theories when faced with falsifying evidence. Kuhn's puzzle-solving criterion sees science as progressing through revolutions that shift scientific paradigms. Normal science operates within a paradigm, solving puzzles that fit existing theories. When too many anomalies accumulate, scientific revolutions occur that lead to new paradigms. This view captures some elements of how science works in practice. However, it is difficult to determine what counts as a puzzle or paradigm shift. Pseudosciences can also experience shifts to new theories without becoming genuinely scientific. Lakatos proposed evaluating research programs rather than individual theories. A scientific research program has a hard core of basic principles surrounded by a protective belt of auxiliary hypotheses. The hard core is preserved, while the protective belt is modified and expanded. For Lakatos, astrology lacks a progressive problem-shift and has too much ad hoc modification of its protective belt. However, determining what counts as ad hoc modification or a progressive problem-shift can still be subjective. Thagard examined why astrology should not be considered scientifically valid. He noted that astrology lacks key criteria like explanatory coherence, which requires that a theory mesh well with other currently accepted theories. Astrology also lacks predictive success, practical applications, and conceptual coherence since there is no mechanism explaining how the positions of stars and planets could influence human lives. However, proponents of astrology could argue that it still meets some conditions of science and more criteria are needed to definitely prove it is pseudoscience. In conclusion, while falsification, puzzle-solving, hard cores, and explanatory coherence all capture important aspects of science, there is no definitive and universal set of criteria to differentiate science from non-science. Demarcation will always remain fuzzy. However, for a theory to be scientifically valid, some key conditions must be met: It must be consistent with existing scientific theories and knowledge. It must offer explanations and mechanisms, not just descriptions and predictions. It must lead to new discoveries and applications. And it must aim to be empirically testable and falsifiable, even if practical limitations exist. When these conditions are lacking, the likelihood of a theory being genuinely scientific diminishes. The debate on demarcation continues, but these types of criteria point the way to determining what science should aim for to meet accepted standards of validity.

**PROMPT:** Write an essay in 600 words to the prompt "Discuss the warning signs of domestic violence and abuse and the appropriate actions that healthcare providers should take when they suspect these forms of abuse in their patients.."

**LLM RESPONSE:** Edward Bond's notorious play Outraged poses a radical challenge to traditional Freudian interpretations of the Oedipus complex. Rather than portraying the complex as a psychological struggle, Bond uses it as a metaphor to critique deeper societal problems. The play employs dark humor and violence to make the audience uncomfortable and confront humanity's capacity for depravity. At the center of the play is the character Len, a practical and severely flawed everyman. While Len initially appears relatable, even likable, his ordinary nature makes his disturbing actions all the more unsettling. The play suggests that within each ordinary person lies the potential for violence and moral failure. Len's desire to kill his stepfather, Frank, does not stem from an unconscious sexual rivalry, as in a traditional Oedipal narrative. Rather, Len's animosity arises from Frank's abusive and predatory behavior. Frank is a figure who revels in violence and uses his position of power to exploit others. The play implies that figures like Frank who embody society's darkest aspects arise not from individual pathology but from systemic failings. The real "complex" here is not Oedipal but rather humanity's tendency to turn a blind eye to injustice and moral corruption. The play's disturbing and violent events implicate the audience through their gradual escalation. At first, Frank's abusive behavior and the suffering of his wife and stepchildren seem sadly familiar, even mundane. But as the violence crescendos, the audience realizes with discomfort that their initial nonchalance makes them complicit. The play suggests ordinary people can become desensitized to extraordinary cruelty, and inaction in the face of injustice enables its perpetuation. Len's eventual parricide is not a victory but another disturbing moral failure. While Frank is a cruel man who deserves punishment, violence should not be met with more violence. Len's actions jeopardize his own humanity, even as he rids the world of a moral monster. The play proposes no easy answers and refuses neat resolutions, leaving the audience unsettled with more questions than solutions. In sum, Outraged utilizes a subversive reinterpretation of the Oedipus story to challenge society's moral failings. With its unsettling events and unlikeable yet ordinary characters, the play implicates the audience's own darkness and capacity to ignore injustice. By the play's end, the true "complex" in question is humanity's tendency to remain outraged at metaphorical monsters like Oedipus, all while enabling real monsters like Frank. Bond suggests the Oedipus complex may reveal less about psychological drives than our unsettling societal tendencies - tendencies the play brutally and humorously exposes.

**PROMPT:** Write an essay in 1300 words to the prompt "Explain how the supply and demand theory can be used as a tool for navigating the market, providing examples of companies that have successfully utilized this theory to their advantage.."

**LLM RESPONSE:** Employers owe a duty of care to their employees to ensure a safe working environment, both physically and mentally. However, the extent of this duty is debated, especially regarding obligations to support employee mental health and wellbeing. The case of Somerset County Council v Barber highlighted the complexities in determining how far an employer's duty extends regarding foreseeable psychiatric harm. In Barber, the House of Lords found that Somerset County Council was liable for the nervous breakdown of an employee, Barber, due to the unreasonable workload and pressures placed on him. Their judgment affirmed that employers have a duty to take reasonable care for the mental health and safety of employees in the workplace. However, the court also noted that employers could not be expected to predict and prevent all psychiatric harm, especially that arising from an employee's own peculiar vulnerability or susceptibility. The ruling in Barber has been criticized as posing too high a burden on employers and for judging the case with the benefit of hindsight. However, others argue it achieved an appropriate balance between employer and employee interests. Workplaces have changed dramatically in the 30 years since Barber was decided, with longer working hours, greater job insecurity, and more isolated working. This amplification of workplace stressors suggests employers should shoulder more responsibility for employee wellbeing. That said, there are arguments against saddling employers with open-ended liability for employee mental health issues. Employees have a degree of personal responsibility for their own wellbeing and for raising issues with their employer. Employment contracts also outline expected working conditions, workloads and hours, limiting employers' duty to account for all possible sources of employee stress. Moreover, psychiatric harm can be challenging to predict and prevent due to the individual nature of mental health. In conclusion, while employers should promote workplace wellbeing and take reasonable steps to identify foreseeable sources of stress and mental harm, they cannot be insurer against all possible psychiatric injury. Balance is needed between employee interests in a safe working environment, and employers' constraints in fully determining and controlling the roots of mental ill health for a diverse range of employees. Overall, the House of Lords in Barber achieved a reasonable compromise, but further clarity is still needed on the extent of responsibility employers can fairly bear for the psychological wellbeing of their workforce.

Figure 26: MGTs that can be correctly detected by the proposed framework but not by the original detector. The identified human-generated parts are highlighted in green.