# OpenReview forum: "Machine-Generated Text Detection Requires Fewer Machine-Human Mixed Texts"
_ICLR.cc/2026/Conference — Submitted to ICLR 2026_

### Official Review · Reviewer_122r · 2025-10-29

**Soundness:** 3
**Presentation:** 3
**Contribution:** 3
**Rating:** 8
**Confidence:** 4

**Summary:**

This paper addresses the challenge of machine-generated text (MGT) detection by first revealing the prevalence of implicit human-machine mixed texts. The authors provide a theoretical analysis of how such ``mixed texts" impact detection and, based on this, propose a novel stacked detection framework. Experimental results demonstrate that the approach outperforms existing detectors, and its boosting strategy can function in a training-free manner.

**Strengths:**

The investigation of human-machine mixed texts is a significant and compelling aspect of machine-generated text (MGT) research.

The authors' analysis of this phenomenon is well-founded, and the method they propose is reasonable.

The experimental validation is thorough, and the fact that their boosting strategy can operate in a training-free manner is a particularly useful and practical feature.

**Weaknesses:**

To better frame the contribution, the introduction should reference existing research on human-machine mixed texts (Sec. 2.2) earlier, even if the focus of this work is different. The current framing inadvertently suggests no prior work exists on this topic.

Additionally, the experiments would be more complete if it could include comparisons and discussion with these relevant methods.

**Questions:**

See the above.

---

> ### Author Response · Authors · 2025-11-17
> **Response to Reviewer 122r**
>
> Many thanks for your positive comments and constructive feedback. Please see the responses to your comments below.
>
> > **[W1]** To better frame the contribution, the introduction should reference existing research on human-machine mixed texts (Sec. 2.2) earlier, even if the focus of this work is different. The current framing inadvertently suggests no prior work exists on this topic.
> >
>
> We thank the reviewer for this excellent suggestion regarding the paper's framing. We agree that referencing existing research on **explicit** mixed text earlier allows us to more sharply define our novel contribution regarding **implicit** mixed text. Therefore, we have revised the Introduction to explicitly cite relevant prior works [1,2,3] immediately when introducing the concept. This creates a clear boundary between previous "human-machine collaboration" studies and our "latent consistency" analysis. Accordingly, we clarify the "First, in addition to explicit human-machine mixed text scenarios, ..." in the introduction to "First, although some research [1,2,3] has begun to focus on explicit mixed text, that is, text completed through human-machine collaboration, ...".
>
> > **[W2]** Additionally, the experiments would be more complete if it could include comparisons and discussion with these relevant methods.
> >
>
> We agree to compare our method against finer-grained detectors. To address this, we have added a sentence-level detector, **SeqXGPT [1],** which can identify human and machine texts with finer-grained granularity and help avoid potential mixing of human-LLM texts. Because SeqXGPT is a model-based detector, we applied our stacked enhancement framework to it (denoted **SeqXGPT-STK**). The results are shown in Table 1 below (Table 18 in the revised version). First, our framework significantly boosts SeqXGPT’s performance (e.g., improving the average AUROC on the Essay dataset from 84.45% to 85.32%). This demonstrates that our "self-enhancement" strategy is flexible and effective even when applied to sentence-level architectures. Second, while SeqXGPT outperforms feature-based approaches, it remains less competitive than the paragraph-level model-based methods (like OpenAI-D). This is because SeqXGPT's direct aggregation of sentence labels often fails to capture the holistic context of a paragraph, leading it to misclassify mixed text as human.
>
> **Table1. Performance concerning AUROC/100. Detectors are trained on ChatGPT text.**
>
> | Dataset | Method | ChatGPT  | GPT4All  | ChatGPT-turbo  | ChatGLM  | Dolly  | Claude  | Avg. |
> | --- | --- | --- | --- | --- | --- | --- | --- | --- |
> | Essay | SeqXGPT | **100** | 99.99 | 52.18 | 99.95 | 99.91 | 54.67 | 84.45 |
> | Essay | **SeqXGPT-STK** | **100** | **100** | **53.17** | **100** | **99.99** | **58.75** | **85.32** |
> | Reuters | SeqXGPT | 99.34 | **99.48** | 61.00 | 99.54 | **99.51** | 60.78 | 86.61 |
> | Reuters | **SeqXGPT-STK** | **99.44** | 99.25 | **65.69** | **99.75** | 99.30 | **76.85** | **90.05** |
> | SQuAD1 | SeqXGPT | 50.96 | 46.89 | 45.62 | 47.17 | 51.18 | 49.99 | 48.64 |
> | SQuAD1 | **SeqXGPT-STK** | **53.89** | **49.24** | **52.17** | **50.20** | **54.06** | **53.19** | **52.12** |
>
> [1] Seqxgpt: Sentence-level ai-generated text detection. EMNLP 2023.
>
> [2] Llm-as-a-coauthor: Can mixed human-written and machine-generated text be detected? NAACL 2024.
>
> [3] Llm-detector: Improving ai-generated chinese text detection with open-source llm instruction tuning. arxiv 2024.

---

### Official Review · Reviewer_ycey · 2025-10-29

**Soundness:** 2
**Presentation:** 2
**Contribution:** 2
**Rating:** 4
**Confidence:** 4

**Summary:**

This paper addresses Machine-Generated Text Detection. By analyzing Jaccard similarity, the authors find that even fully LLM-generated texts contain substantial overlaps with human-written content, termed implicit hybrid text. They show that a higher mixing ratio (α) increases detection difficulty and propose a Stacked Detection Enhancement Framework that filters suspected human segments (thereby reducing α) to improve detection, optimized via a Hard-EM approximation.

**Strengths:**

The paper discusses the phenomenon of implicit hybrid text in MGT, and I highly appreciate the authors’ insight into this important issue.

**Weaknesses:**

- The paper lacks an in-depth analysis of implicit hybrid text. What characteristics do such texts exhibit? Are they in machine-generated text truly misclassified as human-written by the model ? What statistical patterns do they show?
- The validation of implicit hybrid text is rather coarse:
    - The Jaccard similarity metric (based on word sets) only reflects word-level overlap and cannot accurately capture fixed expressions or compositional templates shared between machine-generated and human-written texts.
    For instance, if the word order within a sentence is rearranged, the Jaccard score remains 1, this means the *vocabulary* is identical, not necessarily the *writing pattern*.
    In addition, the paired dataset used may induce topic bias, potentially exaggerating the implicit mixing phenomenon. The authors should explore this on more diverse datasets.
    - The analysis and detection methods primarily rely on word-level or sentence-level statistics.
    However, the core behavior of generative models lies in their conditional probability distributions (logits level): even identical surface sentences can exhibit completely different logits distributions under different contexts.
    Furthermore, most existing statistical metrics for detection also rely on logits-based computations; therefore, relying solely on lexical similarity is insufficient to uncover the true properties of implicit hybrid text.

**Questions:**

- Is the proposed enhancement method effective only for document-level detection? Would it fail or become less applicable for shorter texts?
- Is the proposed enhancement method effective for statistical-based methods？

---

> ### Author Response · Authors · 2025-11-17
> **Response to Reviewer ycey [1/3]**
>
> Many thanks for your constructive feedback. Please see the responses to your comments below.
>
> > **[W1]** The paper lacks an in-depth analysis of implicit hybrid text. What characteristics do such texts exhibit? Are they in machine-generated text truly misclassified as human-written by the model ? What statistical patterns do they show?
> >
>
> Thank you for the questions. We agree that this analysis is crucial. In fact, we have already provided a detailed case study to answer this question in **Appendix H.13 of the initial version** (now Appendix H.15 in the revised version):
>
> - **The characteristics and statistical patterns of such texts.** Our qualitative analysis in **Appendix H.13**  identifies several key distinctions between the human-like sequences and the machine-generated text: (1) **Sentence Structure:** The human-like sequences (filtered by our model) tend to be "relatively short and succinct". In contrast, the machine-generated parts often feature "more complex sentence structures, including frequent subordinate clauses". (2) **Sentence Length:** We found that human-like sequences generally favor "short to medium-length sentences (around 12-15 words)". The machine-generated text, however, averages "around 20-25 words per sentence". (3) **Linguistic Style:** The human-like parts typically use "concise analytical language," whereas the machine-generated text is "more formal" and "frequently incorporates technical terms".
> - **Are they in MGT truly misclassified as human-written by the model?** The answer is yes. Figures 20-24 (now Figures 22-26 in the revised version) show text examples that are "failure cases" for the original detector but "win cases" for the enhancement detector. Our enhanced framework, by identifying and filtering the green-highlighted "human-like parts", correctly classified the remaining, more discriminative text. To validate that these green-highlighted parts are "human-like", we used a voting mechanism across five detection models to create pseudo-labels for them. Unsurprisingly, these parts were consistently identified as human-generated.
>
> > **[W2-1]** The Jaccard similarity metric (based on word sets) only reflects word-level overlap and cannot accurately capture fixed expressions or compositional templates shared between machine-generated and human-written texts. For instance, if the word order within a sentence is rearranged, the Jaccard score remains 1, this means the *vocabulary* is identical, not necessarily the *writing pattern*. In addition, the paired dataset used may induce topic bias, potentially exaggerating the implicit mixing phenomenon. The authors should explore this on more diverse datasets.
> >
>
> We have revised the paper to address both.
>
> **On the Jaccard Similarity Metric.** We fully agree with you that different orders can also yield a Jaccard score of 1. In practice, we carefully examined the texts with 100% similarity and found them to be consistent (including word order). Accordingly, we have revised the paper (Introduction and Section 2.2) to state that "Jaccard similarity was only used to assess human-likeness, and texts with 100% similarity and same word order (in fact, all texts with 100% similarity in our experiments met this condition) demonstrate the existence of implicit mixed text”.
>
> **On Topic Bias and Diverse Datasets.** We agree that the paired dataset may induce topic bias. However, we would like to clarify why this bias would lead to an **underestimation**, not an exaggeration, of the implicit mixing phenomenon. Specifically, within a given dataset where $H$ is the small set of observable human texts (i.e., limited topics) and $S$ the set of machine texts, the proportion of implicitly mixed text based on the consistency metric is $P=\frac{|H\cap M|}{|M|}$. Since the actual human texts $H_{all}$ are far more abundant than observable ones $H$, i.e., $H\subseteq H_{all}$, it logically follows that $P_{all}=\frac{|H_{all}\cap M|}{|M|}\ge P$. Our metric is a conservative lower bound on the true mixing. The fact that we find this mixing even in a small, topic-paired dataset suggests the real-world phenomenon is likely much larger.
> Furthermore, as you suggested, we have **provided new empirical evidence on more diverse datasets.** We ran our analysis on a Chinese text dataset [1] and a code dataset [2]. As shown in Tables 1 and 2 below (Figure 10 in Appendix H.3 in the revised version), we observe the same implicit mixing phenomenon, especially pronounced in the code domain. These results confirm our findings are not an artifact of a single, topic-biased dataset.
>
> **Table 1. Implicit Mixed Chinese Text Evaluation.**
>
> | LLM | Number of Human Sentences | Number of Machine Sentences | Number of Consistent Sentences  |
> | --- | --- | --- | --- |
> | GPT3.5 | 24525 | 28431 | 183 |
> | Qwen | 24525 | 45500 | 112 |
> | ChatGLM | 24525 | 52628 | 36 |
> | Baichuan | 24525 | 32782 | 19 |

---

> ### Author Response · Authors · 2025-11-17
> **Response to Reviewer ycey [2/3]**
>
> Table 2. Implicit Mixed Code Text Evaluation.
>
> | LLM | Number of Human Sentences | Number of Machine Sentences | Number of Consistent Sentences  |
> | --- | --- | --- | --- |
> | ChatGPT | 21950 | 6329 | 1740 |
>
> > **[W2-2]** The analysis and detection methods primarily rely on word-level or sentence-level statistics. However, the core behavior of generative models lies in their conditional probability distributions (logits level): even identical surface sentences can exhibit completely different logits distributions under different contexts. Furthermore, most existing statistical metrics for detection also rely on logits-based computations; therefore, relying solely on lexical similarity is insufficient to uncover the true properties of implicit hybrid text.
> >
>
> We fully agree that the core behavior of generative models lies in their logits and that identical sentences can have different generative probabilities depending on context. This is precisely why our **final detection framework** is designed the way it is. We apologize if our motivational analysis (Section 2.2) obscured this distinction. Let us clarify:
>
> - **Jaccard is Only for Motivation.** As we stated in our response to your [w2-1], Jaccard similarity is only a one-time empirical probe to assess human-likeness. We neither aim to calculate an exact proportion of this mixing nor use it as an evaluation criterion. **This Jaccard (lexical) analysis is not part of our final detection framework.**
> - **The Framework is Distribution-Based, Not Lexical.** You are absolutely correct that detection should rely on logits/distributions. Our framework is designed exactly for this. Specifically, our framework is decoupled from specific features. It wraps around "model-based" detectors (e.g., RoBERTa-based ChatGPT-D, OpenAI-D) . These deep learning models do exactly what you suggest: they learn distinguishing features from the high-dimensional embedding space (logits/hidden states) rather than surface text. As noted in the Introduction, these model-based methods "implicitly learn distinguishing features" that feature-engineering fails to capture. Our Visualization (Appendix H.8 in the initial submission, now Appendix H.10 in the revised version) confirms this: the t-SNE plots of the hidden states show that our framework helps the model separate the *deep feature representations* of human vs. machine text, not just surface words.
>
> > **[Q1]** Is the proposed enhancement method effective only for document-level detection? Would it fail or become less applicable for shorter texts?
> >
>
> Our paper mainly focuses on paragraph-level detection. This is because the core mechanism of our stacked framework relies on filtering text into potentially shorter segments, which fundamentally performs better on longer, document-level content, as discussed in the Limitations section (Appendix F.7). Nevertheless, we are also willing to explore the potential of shorter texts.
> Following your suggestion, we evaluated our framework on the Essay dataset with a strict 64-word limit. As shown in Table 3 below (Table 16 in the revised version), while detection is indeed more challenging on shorter text (consistent with our Theorem 1 and Theorem 3 that detection difficulty correlates with text length), our framework still provides a consistent and encouraging improvement over the base detectors.
>
> **Table 3. Detection performance (AUROC) on the Essay dataset when the text length is at most 64. Detectors are trained on ChatGPT texts.**
>
> | Method | ChatGPT | GPT4All | ChatGPT-turbo | ChatGLM | Dolly | Claude | Avg. |
> | --- | --- | --- | --- | --- | --- | --- | --- |
> | ChatGPT-D | 89.94 | 86.29 | 85.13 | 96.66 | 70.23 | 64.79 | 82.17 |
> | **ChatGPT-STK** | **91.33** | **88.17** | **87.69** | **97.15** | **75.16** | **69.44** | **84.82** |
> | OpenAI-D | 93.50 | 93.10 | 89.93 | 98.09 | 85.36 | 72.74 | 88.79 |
> | **OpenAI-STK** | **95.68** | **94.21** | **93.69** | **98.59** | **88.84** | **81.67** | **92.11** |
> | MPU | **96.28** | **95.77** | **96.01** | 99.24 | 81.00 | 85.57 | 92.31 |
> | MPU-STK | 96.13 | 95.66 | 95.81 | **99.29** | **83.38** | **86.11** | **92.72** |
> | RADAR | 98.63 | 96.25 | 97.87 | 99.30 | 92.29 | 93.31 | 96.28 |
> | **RADAR-STK** | **98.77** | **96.39** | **98.07** | **99.38** | **92.58** | **93.63** | **96.47** |

---

> ### Author Response · Authors · 2025-11-17
> **Response to Reviewer ycey [3/3]**
>
> > **[Q2]** Is the proposed enhancement method effective for statistical-based methods？
> >
>
> To answer directly: **No, the specific Hard-EM optimization framework does not apply to training-free statistical methods.** As we discuss in the Limitations (Appendix F.7) , our framework relies on the iterative "self-enhancement" loop (M-step) to update the model parameters. Statistical methods like DetectGPT or Fast-DetectGPT are zero-shot and do not require parameter optimization, making them incompatible with the M-step. In the paper, we chose to focus on model-based detectors because of their superior performance. As noted in our Introduction, model-based approaches generally outperform feature-engineering/statistical methods because they can implicitly learn complex, distinguishing features from the entire text. Our experiments confirm this: even the baseline model-based detectors (Table 1) significantly outperform statistical baselines like DetectGPT.
>
> However, we would like to emphasize that although the Hard-EM algorithm is specific to trainable models, the core concept, that filtering "human-like" parts (reducing $\alpha$) improves detection, is universal. Future work could explore "static" filtering strategies adapted for statistical metrics.
>
> Thank you for your valuable review. We hope our responses have clarified that most of your concerns regarding the (W1) analysis were already addressed in detail in the initial submission. We also clarified the confusion regarding our model-driven approach in (W2-2). Furthermore, following your suggestions (W2-1, Q1), we have conducted additional experiments on diverse text and short-text scenarios, which further demonstrate our framework. If our revision has satisfactorily addressed your concerns, we kindly request that you consider revising your final rating of our manuscript. If you have any additional concerns or comments that we may have missed in our responses, we would appreciate any further feedback to help us further enhance our work.
>
> [1] Towards Reliable Detection of LLM-Generated Texts: A Comprehensive Evaluation Framework with CUDRT. *arXiv 2024*.
>
> [2] The imitation game: Detecting human and AI-generated texts in the era of ChatGPT and BARD. *Journal of Information Science* 2024.

---

> > ### Author Response · Authors · 2025-11-28
> >
> > Dear Reviewer ycey,
> >
> > Thank you once again for your valuable comments on our submission. **As the discussion phase is approaching its end**, we would like to kindly confirm whether we have sufficiently addressed all of your concerns (or at least part of them). Should there be any remaining questions or areas requiring further clarification, please do not hesitate to let us know. If our revision has satisfactorily addressed your concerns, we would greatly appreciate your consideration in adjusting the evaluation scores accordingly.
> >
> > We sincerely look forward to your feedback.
> >
> > Best regards,
> >
> > The authors

---

### Official Review · Reviewer_kQVt · 2025-10-31

**Soundness:** 3
**Presentation:** 1
**Contribution:** 3
**Rating:** 2
**Confidence:** 3

**Summary:**

The article under review proposes a two-stage method for the detection of machine generated and mixed machine-human generated text. A two step algorithm is proposed, whereby text is first run through an E-step where sentences judged very similar to human sentences are removed, and then run through an M-step where an off the shelf detector is run on the remaining sentences.

The idea is an intriguing one which is certainly worthy of detailed explanation. A good deal of theoretical justification is given, I have my criticisms of this theoretical justification given below. The authors also keep separate a theoretical discussion of their two step process from the practical implementation, I understand why they did this but am concerned that this makes it extremely hard to understand what the authors actually do. Nevertheless, I am intrigued by the basic idea, which I think could make a very strong article if better explained.

**Strengths:**

1. The idea is intriguing and I think very novel. It makes a lot of sense for detecting mixed text.
2. As far as I can tell, the results are strong, with a significant uptick in performance for all the detectors tested. This is good news! I would like to see the weaker parts of this paper removed in order that the strengths can be better explained.

**Weaknesses:**

1. I have spent a lot of time reading the article, I believe I have the ideal background to understand this article, yet I do not understand the final algorithm used. To be precise, I think Figure 2 gives a very clear overview of what is happening, subject to understanding the E-Step and the M-Step. And I *think* (but am not certain) that the E-step is to filter out all sentences with sufficiently high Jacard score compared to a large reference database of human text and the M-step is to use an off the shelf model, but I am not completely certain. This should be explained on page 2.
2. Throughout, the authors refer to sufficiently 'human like' machine generated text as human text. For example, they describe 'LLMs, with their powerful generation capabilities, can generate texts consistent with human writing in simple sentence structures, fixed phrases, and so on.' My understanding is that the authors refer to this as 'human text' in what follows. I think this terminology is confusing, and I don't see what it adds. Part of my confusion comes because, in Figure 2, we have to think of two different definitions of human text, one is 'written by humans' and the other is 'considered very likely written by humans by whatever filter is used at the E-step'. If the authors wish to keep this terminology, then I would like to see an extremely precise definition (e.g. a sentence is considered human if it has 100% Jaccard similarity with a sentence in the human written collection of reference texts).
3. I think RQ1 only really makes sense with the authors odd framing off 'human text'. I would reframe it 'given a collection of reference texts, what proportion of sentences in machine generated text are identical to sentences present in the reference texts?' But on the whole, I think this framing and this question distract from the much more interesting proposition that the authors algorithm can improve algorithms for detecting machine generated text.
4. I find section 2.1, and the theoretical sections that rely on it, to be rather weak. Specifically, text is considered in two settings, firstly where each sentence is picked iid from a collection of reasonable sentences, and secondly where some limited dependence on previous sentences is introduced. This resembles neither human language nor machine generated language. In particular, the non-IID setting is still a very constricted mathematical model of language, in which (for example) the probability of a sentence appearing at position k of a text is independent of the order in which the previous k-1 sentences appear. This is a profound restriction on models of language.
5. In the problem definiton (line 115) a characterization of the problem of detecting machine generated text is given. But this holds only if one accepts that both human and machine text are generated according to the very unnatural model of language model described in my previous comment.

6. Throughout pages 3-6 there is a series of descriptions where the authors first propose a method, and then describe how that method is unfeasible so a modification is made. The mathematics here is difficult, and I lose track of which of the mathematics is dependent on the (to my mind unrealistic) assumptions of section 2.1, and which is not. Moreover, at no point is a final description given of precisely what the algorithm used is. I think this is a shame, again, I like your basic idea! But I think you confuse the reader by putting lots of hard maths on a very unstable set of basic assumptions about language, rather than giving a clear explanation of exactly how your algorithm works.

**Questions:**

1. Figure 2 looks like it should work with detectors such as fast-detect GPT for the M-step. This would be a very natural experiment to run. Did you try, and if not is there a natural reason why not? In particular, if you tried it and it failed then I think this would be useful information for the reader.

2. I do not know whether the formula on line 110 is a mathematical formula or a vague heuristic. It looks like a precise formula, and I am reading it as a precise characterization of probability distributions. But, as I understand it, expectation only makes sense when random variables take values in a vector space. Should this formula be read with a semantic mapping from sentences to some vector space? Does s_i mean the ith sentence or some mapping of the ith sentence?

3. Usually the algorithm boxes help me understand what an author has done. But in your case, I am completely unable to find the information I need, where formulae are referenced the formulae do not say which model is being used for e.g. computing the values used in equation (3).

4. Table 1 does not include the information from Squad, fine for saving space, but given that the information from Squad looks less good for your argument I would like to see at least some discussion of it.

5. In section H.2, I am slightly concerned about different hyperparameter settings being used to generate different table entries. Can you confirm that the set which you used to choose your hyperparameters was not the test set? You write No-
tably, the default settings are obtained through grid search, aiming to show the maximum potential
of detection enhancement., again, this is all performed on a separate training set right? Could you explain the way in which different algorithms are performed on different language models, in particular does your black box setting still have different parameter settings according to which language model is generating the text?

6. Please make your article easy to skim read. I can read the DetectGPT article and in five minutes I understand both the key idea and the implementation. I think I understand your key idea after five minutes from the helpful Figure 2, but I have spent several hours with your article and still cannot say with certainty what your final implementation is.

Again, I really like your basic idea! I have tried to be direct with my comments, this is meant to be helpful, my hope is that you can make significant changes and I can be in a position to recommend acceptance.

---

> ### Author Response · Authors · 2025-11-17
> **Response to Reviewer kQVt [1/7]**
>
> Thank you very much for the valuable feedback. Please find the point-to-point responses below. Any further comments and discussions are welcome!
>
> > **[W1]** I have spent a lot of time reading the article, I believe I have the ideal background to understand this article, yet I do not understand the final algorithm used. To be precise, I think Figure 2 gives a very clear overview of what is happening, subject to understanding the E-Step and the M-Step. And I *think* (but am not certain) that the E-step is to filter out all sentences with sufficiently high Jacard score compared to a large reference database of human text and the M-step is to use an off the shelf model, but I am not completely certain. This should be explained on page 2.
> >
>
> Thank you for this critical feedback. The reviewer's assumption that the Jacard score was part of the E-step is completely understandable given that we presented it in Section 2.2, and we apologize for this. We must first clarify the most crucial point: **Jaccard similarity is NOT part of our detection framework or algorithm.** We used Jaccard similarity only in Section 2.2 as a one-time empirical analysis to provide evidence that "implicit mixed text" exists.
> Your confusion about "E-step" and "M-step" in Figure 2 is also understandable, as we mistakenly used **training terminology** to describe the **inference process**.
>
> **The Inference Process.** This is what Figure 2 illustrates. It is a two-pass inference pipeline using the already-trained detector $f(S, \theta)$:
>
> 1. **The Filtering Step** (top-right of Figure 2): This is not a Jaccard score. In this step, the "Unknown Text" is split into sub-sequences (Sub-text 1, 2, 3...). The trained detector $f(S, \theta)$ runs on each sub-sequence, and any sequence that the detector classifies as confidently “human-like”(according to Eq. 3) is filtered out.
> 2. **The Detection Step** (bottom-right of Figure 2): This is not a different off-the-shelf model. In this step, the remaining text is concatenated and fed back into the exact same detector $f(S, \theta)$. The output of this second pass is the final prediction (AI-generated or Human-written) for the whole text.
>
> **The Training Process.** The training process (described in Sec 3.2) is a "self-enhancement" loop. This is where the "E-step" and "M-step" terminology truly applies, as we use a Hard-EM inspired optimization:
>
> 1. **(E-step):** The current detector $f(S, \theta^t)$ filters the training data to find the best-fit latent variables $z_i$ (i.e., identifying the "human-like" sequences).
> 2. **(M-step):**  The detector's parameters $\theta^{t+1}$ are then updated (it learns) only on this new, filtered dataset to optimize Eq. 4.
> 3. **(Next Iteration):** This new, improved detector $f(S, \theta^{t+1})$ is then used in the next E-step, allowing it to filter even more accurately. This interwoven process allows the detector to iteratively bootstrap its own performance.
>
> In the revised version, we have significantly revised Section 3.2 (Lines 332-336) and the caption of Figure 2 to clearly illustrate the inference process of our framework, and revised the Introduction (Lines 87-89) and Section 3.2 (Lines 337-342) to highlight the training process of our framework. Besides, we have revised Figure 2 to explicitly state that our framework uses the same detector in this two-pass process and removed the training terms (E-step and M-step). We thank the reviewer again for making the presentation of our framework clearer.

---

> ### Author Response · Authors · 2025-11-17
> **Response to Reviewer kQVt [2/7]**
>
> > **[W2]** Throughout, the authors refer to sufficiently 'human like' machine generated text as human text. For example, they describe 'LLMs, with their powerful generation capabilities, can generate texts consistent with human writing in simple sentence structures, fixed phrases, and so on.' My understanding is that the authors refer to this as 'human text' in what follows. I think this terminology is confusing, and I don't see what it adds. Part of my confusion comes because, in Figure 2, we have to think of two different definitions of human text, one is 'written by humans' and the other is 'considered very likely written by humans by whatever filter is used at the E-step'. If the authors wish to keep this terminology, then I would like to see an extremely precise definition (e.g. a sentence is considered human if it has 100% Jaccard similarity with a sentence in the human written collection of reference texts).
> >
>
> We thank the reviewer for this crucial suggestion. You are correct that we refer to this subset of human-like machine-generated text as “human text” in the paper. To avoid confusion and follow your suggestions, we have adopted the more precise term **“consistent text”**, i.e., common text that can be generated by both humans and machines. This adjustment should resolve your concerns without altering our theoretical framework.
>
> Under this terminology, there are two types of text filtered by the E-step in Figure 2:
>
> - For machine text, the filtered portion is high-confidence consistent text.
> - For human text, the filtered portion may be high-confidence consistent text or the human-only portion (i.e., beyond consistent text), either of which is possible.
>
> We have updated the Introduction, Section 2.2, and Section 3.2 to consistently adopt this new, precise terminology. We believe this modification, prompted by your feedback, significantly clarifies the paper.
>
> > **[W3]** I think RQ1 only really makes sense with the authors odd framing off 'human text'. I would reframe it 'given a collection of reference texts, what proportion of sentences in machine generated text are identical to sentences present in the reference texts?' But on the whole, I think this framing and this question distract from the much more interesting proposition that the authors algorithm can improve algorithms for detecting machine generated text.
> >
>
> We sincerely appreciate the reviewer’s valuable feedback. We agree with the reviewer’s reframing of RQ1, namely to demonstrate the existence of “consistent text” (as per our newly adopted terminology), i.e., implicitly mixed text. Accordingly, we will revise RQ1 from “is it possible that mixed text exists even if the text is entirely machine-generated?” to “is it possible for comment texts to exist that are consistent with those of humans, even if the text is entirely machine-generated?”
>
> Furthermore, we concur that the proposed stacked detection enhancement framework is our core contribution. However, it is undeniable that the implicit mixed text revealed under RQ1 extends beyond existing explicitly mixed scenarios, indicating that the practical detection challenge may be far more severe than recognized, and advocating for attention to implicit mixed text detection. As such, it is not only motivation but also a noteworthy contribution. In the Introduction of the revised version, we have enriched the section specifically addressing RQ3 (Lines 82-91). This revision balances our contributions, thus highlighting the core enhancement framework design. We thank the reviewer again for the valuable suggestions regarding the paper’s structure.

---

> ### Author Response · Authors · 2025-11-17
> **Response to Reviewer kQVt [3/7]**
>
> > **[W4]** I find section 2.1, and the theoretical sections that rely on it, to be rather weak. Specifically, text is considered in two settings, firstly where each sentence is picked iid from a collection of reasonable sentences, and secondly where some limited dependence on previous sentences is introduced. This resembles neither human language nor machine generated language. In particular, the non-IID setting is still a very constricted mathematical model of language, in which (for example) the probability of a sentence appearing at position k of a text is independent of the order in which the previous k-1 sentences appear. This is a profound restriction on models of language.
> **[W5]** In the problem definiton (line 115) a characterization of the problem of detecting machine generated text is given. But this holds only if one accepts that both human and machine text are generated according to the very unnatural model of language model described in my previous comment.
> >
>
> We greatly appreciate the reviewer’s constructive feedback. We acknowledge that the mathematical frameworks in Section 2.1 are not realistic generative models of language. We would like to clarify that our intention was not to propose it as a realistic generative model. Instead, our goal was to employ a **tractable mathematical framework, drawn from prior work [1,2,3]** and proven practical in various natural language tasks, to formally examine how the proportion of mixed text ($\alpha$) theoretically affects detection performance. Such a simplified model is very common in classical machine learning theory, e.g., the strict IID assumption of Rademacher complexity and VC-dimension analyses [4], the fixed learning rate assumption of convergence-rate analyses [5], and the conditional independence assumption of the Naive Bayes classifier [6]. Finally, we would like to clarify that the proposed framework draws on these theoretical results only to inform the core idea of filtering and **does not rely on the assumptions** presented in Section 2.1.
>
> As you suggested, we have revised the Limitation section (Appendix F.7) to make this distinction explicit: our theory is a tool to isolate a variable, not an attempt to replicate the full complexity of real-world language generation.

---

> ### Author Response · Authors · 2025-11-17
> **Response to Reviewer kQVt [4/7]**
>
> > **[W6]** Throughout pages 3-6 there is a series of descriptions where the authors first propose a method, and then describe how that method is unfeasible so a modification is made. The mathematics here is difficult, and I lose track of which of the mathematics is dependent on the (to my mind unrealistic) assumptions of section 2.1, and which is not. Moreover, at no point is a final description given of precisely what the algorithm used is. I think this is a shame, again, I like your basic idea! But I think you confuse the reader by putting lots of hard maths on a very unstable set of basic assumptions about language, rather than giving a clear explanation of exactly how your algorithm works.
> **[Q3]** Usually the algorithm boxes help me understand what an author has done. But in your case, I am completely unable to find the information I need, where formulae are referenced the formulae do not say which model is being used for e.g. computing the values used in equation (3).
> >
>
> Thank you for your invaluable feedback, and we apologize for the confusion regarding the boundary between our theoretical motivations and the proposed framework. Let us clarify, as we have done in the revised version:
>
> - Regarding the assumptions in Section 2.1: as mentioned in our response to your [W4 & W5], these assumptions are not intended to perfectly capture real-world complexity, which is generally infeasible in classical machine learning theory. Instead, they follow existing and empirically practical assumptions [1,2,3] to **merely explain why filtering could be beneficial, rather than the foundation of the proposed framework**. The proposed framework only inspired the core idea of filtering from theoretical results and does not rely on the assumptions presented in Section 2.1.
> - Regarding the presentation of the methodology: although this idea of filtering is intuitive, it requires careful optimization-design objectives (since the localization of each subsequence is unknown, we adopt an EM algorithm) and efficient optimization strategies (using a hard-EM approach and an approximation of the latent variable $z_i$ , reducing computational complexity from exponential to quadratic). This is why we present the derivation of the framework in a rigorous manner in Section 3.2.
>
> In addition, following your constructive feedback, we have included a detailed description of the algorithm in the revised version (Overall Framework part of Section 3.2):
>
> - **Inference Pipeline** (Lines 2-7 in the revised algorithm). This is used at test time with the trained detector $f(S,\theta)$.
>     1. **Split (Line 3):** The "Unknown Text" is first split into a set of smaller sequences (e.g., Sub-text 1, 2, 3...).
>     2. **Identify "Human" Parts (Line 4):** Any sub-sequence that the detector **$f(S, \theta)$** classifies as *confidently human* is identified. We use a very strict threshold ($r_e \ll 0.5$) and a maximum filter ratio ($\tau$) to ensure we only filter high-confidence “human” text.
>     3. **Create "Filted text"(Line 5):** We concatenate all the *remaining* sub-sequences into a new, shorter "Filted text".
>     4. **Final Prediction (Line 6):** We run the **exact same detector model $f(S, \theta)$** a *second time*, but on this new "Filter text". The output of this second pass is the final prediction (Human-written or AI-generated) for the entire text.
> - **Training Process** (Lines 8-16 in the revised algorithm). The confusing "E-step" and "M-step" terminology applies only to the training process, where we use Hard-EM to learn the parameters $\theta$ of this single detector.
>     1. **Hard E-step (Line 11):** We calculate the best-fit latent variables $z_i$. We do this by running the entire Inference Pipeline (Lines 2-7) on the training data using the current detector, $f(S, \theta^t)$. This gives us the filtered text for each training sample.
>     2. **Hard M-step (Line 12):** We update the detector's parameters to $\theta^{t+1}$. We do this by optimizing the objective function in Eq. 4 only on the filtered data we obtained from the Hard E-step.

---

> ### Author Response · Authors · 2025-11-17
> **Response to Reviewer kQVt [5/7]**
>
> > **[Q1]** Figure 2 looks like it should work with detectors such as fast-detect GPT for the M-step. This would be a very natural experiment to run. Did you try, and if not is there a natural reason why not? In particular, if you tried it and it failed then I think this would be useful information for the reader.
> >
>
> Since the proposed method relies on the Hard-EM training process, it currently applies only to **model-based detectors** (e.g., ChatGPT-D, OpenAI-D) that can be trained end-to-end. In contrast, zero-shot methods like Fast-DetectGPT do not require optimization and thus are not applicable in the M-step. This is also what we emphasized in the Limitation section (Appendix F.7).
>
> Regarding compatibility with other detectors, in theory, any detector could be used in the E-step (for filtering), while the detector to be enhanced could be used in the M-step. However, since the E-step would not be updated, this enhancement strategy using different detectors is static (the remaining texts during each training round’s M-step would remain unchanged). In contrast, in the proposed framework, the E-step and M-step use the same detector (e.g., ChatGPT-D), enabling them to “self-enhance” through iterative collaboration: (1) in the E-step, the detector $f(S,\theta^t)$ filters out “human” sequences, thereby supplying higher-quality text to the same detector in the M-step; (2) the higher-quality model $f(S,\theta^{t+1})$ learned in the M-step then serves as a detector in the next E-step, filtering out more accurate “human” sequences. This interwoven process leads to self-enhancement, which is not possible with static enhancement. Moreover, using the same detector avoids the cost of introducing additional detectors, making it more practical.
>
> > **[Q2]** I do not know whether the formula on line 110 is a mathematical formula or a vague heuristic. It looks like a precise formula, and I am reading it as a precise characterization of probability distributions. But, as I understand it, expectation only makes sense when random variables take values in a vector space. Should this formula be read with a semantic mapping from sentences to some vector space? Does s_i mean the ith sentence or some mapping of the ith sentence?
> >
>
> Thank you for your careful review; it helps us clarify our formal notation. You are exactly right: for the expectation $\mathbb{E}[\cdot]$ and the summation $\sum$ to be meaningful, they must operate in a vector space. This is indeed the case. In this model, which follows the practical setting from prior work [1], each sentence $s_i$ is represented by its semantic embedding vector. Therefore, $\frac{\sum_{k=1}^{i-1}s_{k}}{i-1}$ is the centroid vector of the embeddings of the previous $i-1$ sentences. This is what we referred to as the "average meaning" in the original draft. We have revised Section 2.1 (Line 119) to state this explicitly. We apologize for the initial lack of clarity.
>
> > **[Q4]** Table 1 does not include the information from Squad, fine for saving space, but given that the information from Squad looks less good for your argument I would like to see at least some discussion of it.
> >
>
> We agree this is an essential point and have added an explicit discussion to Appendix H.4 of the revised version. The SQuAD1 results, while appearing "less good," are not a weakness in our argument. On the contrary, they provide strong empirical evidence for our paper's two central claims.
>
> 1. **Why All Detectors Fail on SQuAD1 (Proves Our Theory).** Our core theoretical claim, as argued in Theorem 1 and Theorem 3, is that the mixed text significantly hinders detection. The SQuAD1 dataset is the perfect example of this. As we describe in Section 4.1 and Appendix H.1, SQuAD1 is an explicit mixed-text dataset in which each sample consists of a human-written question concatenated with either a human or a machine answer. The poor performance on highly mixed SQuAD1 directly empirically validates the main theoretical findings of this paper.
> 2. **Why Our Enhancement is Less Pronounced (Proves Our Mechanism's Limits).** Our framework's effectiveness relies on the base detector's ability to confidently identify and filter the "human-like" parts in the E-step. As we discuss in our limitations in Appendix F.7, if a detector is "incapable of recognizing subsequences (i.e., very weak detectors), we cannot guarantee our effectiveness". Therefore, the enhancement is not significant due to the weak performance on highly mixed SQuAD1. It is worth noting that **improving weak detectors is not our focus, as they are impractical in practice**. Nevertheless, our enhancement effect remains encouraging and positive.
>
> We have added these discussions in Appendix H.4 to explicitly highlight how the SQuAD1 results strongly resonate with and support the core ideas of our paper.

---

> ### Author Response · Authors · 2025-11-17
> **Response to Reviewer kQVt [6/7]**
>
> > **[Q5]** In section H.2, I am slightly concerned about different hyperparameter settings being used to generate different table entries. Can you confirm that the set which you used to choose your hyperparameters was not the test set? You write Notably, the default settings are obtained through grid search, aiming to show the maximum potential of detection enhancement., again, this is all performed on a separate training set right? Could you explain the way in which different algorithms are performed on different language models, in particular does your black box setting still have different parameter settings according to which language model is generating the text?
> >
>
> Thank you for the feedback on experimental integrity. We are pleased to confirm the details of our experiments and address your concerns.
>
> First, we want to confirm that **we did not use the test set for hyperparameter selection**. As described in Appendix H.1, we split the dataset into training, validation, and test sets (2:1:1 ratio). All "grid search" optimization for hyperparameters ($r_e, k, \tau$) was performed on the **validation set**. The final results reported in all tables are from the evaluation on the held-out **test set**.
>
> Second, to be perfectly clear: **our framework does not require different parameter settings for each target LLM.** The "different hyperparameter settings" you noted in Appendix H.2 are not tuned per-target-LLM. Instead, they are tuned once for each (Base Detector + Dataset) combination. A concrete example will make this clear:
>
> - To get the ChatGPT-STK results for the Essay dataset, we took the detector and the validation set. We performed a grid search once and found the best hyperparameter set ($r_e=0.01$, $k=3$, and $\tau=0.2$).
> - We used this set of hyperparameters to train the final model on the training set (e.g., ChatGPT text in Table 1, and other LLM text in Tables 3-14).
> - This final, fixed detector (with its fixed parameters) was then evaluated with **no changes** across the test sets for all target LLMs (ChatGPT, GPT-4, GPT-turbo, ChatGLM, etc.) .
>
> The parameters are fixed based on the dataset and are completely "blind" to the specific target LLMs to be tested or trained. We have updated Appendix H.2 to make it more explicit.

---

> ### Author Response · Authors · 2025-11-17
> **Response to Reviewer kQVt [7/7]**
>
> > **[Q6]** Please make your article easy to skim read. I can read the DetectGPT article and in five minutes I understand both the key idea and the implementation. I think I understand your key idea after five minutes from the helpful Figure 2, but I have spent several hours with your article and still cannot say with certainty what your final implementation is.
> **[Comment]** Again, I really like your basic idea! I have tried to be direct with my comments, this is meant to be helpful, my hope is that you can make significant changes and I can be in a position to recommend acceptance.
> >
>
> Thank you for your comment! Accordingly, we have taken your extensive feedback to heart and performed a major revision of the paper with the primary goal of making our framework and algorithm simple to understand.
>
> 1. **Clarifying the Framework (W1, W6, Q1, Q3).**
>     - **Jaccard Similarity vs. Framework:** We have revised the Introduction and Section 2.1 to state explicitly that Jaccard similarity (Figure 1) is only used to assess human-likeness, and added the number of consistent texts to motivate the existence of implicit mixed text. It is not used in our framework's E-step or filtering process.
>     - **A Clearer Algorithm:** We have completely revised Introduction, Section 3.2, Algorithm 1, and Figure 2 to remove all ambiguity. We now explicitly separate the "Inference Pipeline" (the final algorithm you asked for) from the "Training Process" (the actual E/M steps). We believe this change makes the implementation as clear as the DetectGPT paper you mentioned.
>     - **Applicability:** We clarified in Appendix F.7 that our Hard-EM framework requires a model-based, trainable detector for the M-step to enable the self-enhancement loop.
> 2. **Improving Terminology and Framing (W2, W3).**
>     - **"Consistent Text":** We have removed the confusing "human text" terminology and replaced it throughout the paper with the precise term **"consistent text"**.
>     - **Focus on the Framework:** As you suggested, we have streamlined the discussion of RQ1 (the problem) and significantly enriched the discussion of RQ3 (the solution) to better highlight our core contribution: the stacked enhancement framework.
> 3. **Clarifying Theoretical Assumptions(W4, W5, W6).**
>     - We have added a clarification to our Limitations section (Appendix F.7) explicitly stating that the models in Section 2.1 are only for theoretical motivation. They are standard tools used to isolate the "mixed proportion" ($\alpha$) variable. We reiterate that our framework does not depend on these assumptions.
> 4. **Clarifying Experimental Details (Q4, Q5).**
>     - **SQuAD1 Results:** We have added a new discussion of the SQuAD1 results, explaining how it provides strong empirical evidence for our paper's two central claims.
>     - **Hyperparameters:** We clarified that all hyperparameters were tuned once on a separate validation set (not the test set) for each (Dataset + Base Detector) combination and then fixed for all target LLMs, ensuring a fair, black-box evaluation.
>
> We once again sincerely thank the reviewers for their direct and highly constructive feedback, and apologize for the confusing statements in our paper. We believe this major revision, guided by your comments, has solved these clarity issues and made the paper significantly stronger. We hope it now meets your standards for acceptance. If you have any additional concerns or comments that we may have missed in our responses, we would appreciate any further feedback to help us further enhance our work.
>
> [1] Position: On the possibilities of AI-generated text detection. In *Forty-first International Conference on Machine Learning*. 2024.
>
> [2] Latent Dirichlet allocation (LDA) and topic modeling: models, applications, a survey. *Multimedia tools and applications*, 2019.
>
> [3] Topics as entity clusters: Entity-based topics from large language models and graph neural networks. In *Proceedings of COLING 2024*.
>
> [4] *Foundations of machine learning*. MIT press, 2018.
>
> [5] Beyond the regret minimization barrier: optimal algorithms for stochastic strongly-convex optimization. *The Journal of Machine Learning Research*, 2014.
>
> [6] On the optimality of the simple Bayesian classifier under zero-one loss. *Machine learning*, 1997.

---

> > ### Comment · Reviewer_kQVt · 2025-11-24
> > **Thanks for your responses**
> >
> > Many thanks for your responses, which I think clarify many of my questions. I'll take some time to read your article again in detail and look forward to the discussion stage with other reviewers, which I enter with an open mind.

---

> > > ### Author Response · Authors · 2025-11-26
> > >
> > > Thank you for your valuable time and your willingness to re-evaluate our work with an open mind. It is encouraging to hear that our previous responses have helped clarify your initial concerns. We understand that re-reading the paper in detail requires significant time, below is a summary of our main response for your convenience:
> > >
> > > - **Jaccard Similarity:** We clarify that Jaccard similarity is NOT part of the enhancement framework.
> > > - **Algorithm Clarity:** We rewrote Section 3.2, Algorithm 1, and Figure 2 to distinguish the reasoning and training processes that propose the framework.
> > > - **Terminology:** We define "human text" in machine text as "consistent text" to avoid confusion.
> > > - **Theoretical Assumptions:** We clarify that the simplifying assumptions (derived from previous research) are only used for heuristic filtering; our framework does NOT rely on these assumptions to function.
> > > - **Experiments:** We clarify the applicability of zero-shot methods, the discussion of SQuAD1 Results, and the correctness of parameter selection.
> > >
> > > We believe this major revision, guided by your constructive feedback, has significantly improved the paper's clarity. We hope these explanations and the revised version allow you to reconsider the current score.

---

### Official Review · Reviewer_nLWU · 2025-11-01

**Soundness:** 2
**Presentation:** 2
**Contribution:** 2
**Rating:** 4
**Confidence:** 4

**Summary:**

The paper introduced a stacked detection enhancement that uses any detector to filter high-confidence human sequences from a passage and then runs the same detector on the residual text.

**Strengths:**

Test in Essay, Reuters, SQuAD1 datasets and ChatGPT, GPT-4, ChatGLM, Dolly, Claude/StableLM models. And the results show consistent gains in strict low-FPR regimes.

**Weaknesses:**

Please check the questions.

**Questions:**

Removing human-like parts to help has been discussed in prior mixed-authorship and localization work, like RoFT, TriBERT, and CoAuthor.

Using Jaccard similarity of word sets per sentence to define that many AI sentences are “human” is not correct. For example,  "Thank you for XXX" will become 100% overlap regardless of authorship.

The sample-complexity bounds rely on the assumption that each sentence is independent and identically distributed. In the real world, it's impossible as sentences are interdependent, attribution is latent and ambiguous, and α varies by domain and author.

TPR@FPR's improvement can be caused by easier post-filter distributions rather than better AI and human separation.

Add full ROC/PR curves and expected cost under plausible base rates.

---

> ### Author Response · Authors · 2025-11-17
> **Response to Reviewer nLWU [1/2]**
>
> Thank you very much for the valuable feedback. Please find the point-to-point responses below. Any further comments and discussions are welcome!
>
> > **[Q1]** Removing human-like parts to help has been discussed in prior mixed-authorship and localization work, like RoFT, TriBERT, and CoAuthor.
> >
>
> We thank the reviewer for pointing to this important line of work, as it helps us to more precisely articulate our novel contributions (Appendix F.3), which differ from this prior art in three key ways:
>
> - **Focus on Implicit Mixed Text.** Our primary distinction is the problem we identify. Unlike the explicit forms of collaboration in existing works (e.g., a machine continuing human text in RoFT), we demonstrate in Section 2.2 that **implicit** "mixed" text exists even in entirely machine-generated texts. This finding makes the "mixed text" problem a far more general and fundamental challenge than just scenarios of explicit mixed-authorship.
> - **Theoretical Foundation.** To our knowledge, our paper is the first to theoretically demonstrate the impact of these mixed texts on detection. While previous work has empirically shown that mixed-authorship text is hard to detect, it lacks theoretical guarantees. We take the first step to provide a theoretical guarantee (Theorems 1 & 3) that the detection difficulty is proportional to the "mixed degree" ($\alpha$).
> - **A General Enhancement Framework.** Inspired by this theoretical finding, our method is not a new, specific localization model. Instead, we propose a conceptual enhancement framework that is decoupled from any specific detector. As shown in Figure 2 in the paper, our stacked framework acts as a "boosting strategy" that can be applied to any existing model-based detector (e.g., OpenAI-D, ChatGPT-D, MPU, and RADAR).
>
> We have revised the Contribution section (Appendix B) to more clearly differentiate our work, including adding these valuable references.
>
> > **[Q2]** Using Jaccard similarity of word sets per sentence to define that many AI sentences are “human” is not correct. For example, "Thank you for XXX" will become 100% overlap regardless of authorship.
> >
>
> You are correct that a 100% Jaccard similarity does not classify an AI-generated sentence as "human". Our original intention was not to prove authorship, but to provide evidence for a category of text that is indistinguishable from human text. To avoid confusion, we have revised the statement to state that "Jaccard similarity was only used to assess human-likeness, and texts with 100% similarity and same word order (in fact, all texts with 100% similarity in our experiments met this condition) demonstrate the existence of implicit mixed text”. Furthermore, we no longer label this text "human". Instead, we have introduced the term **"consistent text"** to define any sequence that is common to both human and machine distributions. This would easily resolve your concerns without changing our theoretical framework.
>
> > **[Q3]** The sample-complexity bounds rely on the assumption that each sentence is independent and identically distributed. In the real world, it's impossible as sentences are interdependent, attribution is latent and ambiguous, and α varies by domain and author.
> >
>
> Thank you for your question. We would like to address the two distinct points of your concern:
>
> - **On the Non-IID Assumption:** We are happy to clarify that our theoretical analysis is not limited to the IID assumption but also considers more realistic non-IID settings in **Appendix D of the initial submission**. Theorems 3 (Appendix D.1) and 4 (Appendix D.2) confirm that our central finding holds: the detection difficulty is still proportional to the "mixed degree" ($\alpha$), even when sentences are interdependent.
> - **On Latent Attribution and Variable $\alpha$:** We fully agree that in the real world, attribution is latent and $\alpha$ is variable. Our theory is not intended to model this full, unfeasible complexity. Rather, it is designed to **isolate the core mechanism** of the mixing ratio ($\alpha$) to ensure theoretical tractability. This factor-fixing simplification is common in foundational theoretical work (e.g., Rademacher complexity and VC-dimension analyses [1] assume a fixed data distribution, and convergence-rate analyses fix the learning rate [2]).
>
> We have revised the Limitations section (Appendix F.7) to clarify the scope of our theory and to highlight that future theoretical work could focus on these valuable extensions, such as latent attribution and variable $\alpha$.

---

> ### Author Response · Authors · 2025-11-17
> **Response to Reviewer nLWU [2/2]**
>
> > **[Q4]** TPR@FPR's improvement can be caused by easier post-filter distributions rather than better AI and human separation.
> >
>
> Thank you for your valuable comment. You are correct if our E-step merely discarded human-like text and the M-step made no updates. We are pleased to clarify that our framework goes far beyond this, intentionally using the same detector in both steps to create a **virtuous cycle**:
>
> 1. **(E-step):** The detector filters 'human-like' sequences. As you noted, this creates an easier distribution.
> 2. **(M-step):** By training on this easier distribution, the same detector can learn a **more discriminative boundary** and become a higher-quality detector.
> 3. **(Next Iteration):** This new, improved detector is then used in the next E-step, allowing it to filter even more accurately. This interwoven process allows the detector to iteratively bootstrap its own performance.
>
> Besides, in the initial submission, we have provided direct visual proof in Figure 15 of Appendix H.8 (Appendix H.10 of the revised version). The T-SNE visualization clearly shows that the feature representations from the enhanced detector (e.g., ChatGPT-STK) are significantly more separable than those from the original detector. This confirms the improvement is not an artifact of an easier distribution, but an enhancement of the detector's discriminative power.
>
> We have added this discussion to the revised version (Appendix F.5 and Section 3.2) to clarify this core mechanism.
>
> > **[Q5]** Add full ROC/PR curves and expected cost under plausible base rates.
> >
>
> As suggested, we have included ROC curves in Figure 14 of the revised manuscript. As observed, when focusing on extremely low FPR (0%–5%), the enhanced detectors show significant advantages, which are highly valuable for MGT detection that prioritizes low false positives.
>
> Furthermore, in Appendix H.5 of the revised manuscript, we have provided a detailed analysis of total expected cost as a function of the base rates. Concretely, we define a formal expected cost function: $E(Cost)=P(Human)\cdot P(FP)\cdot C_{FP}+P(Machine)\cdot P(FN)\cdot C_{FN}$, where $P(Human)$ and $P(Machine)$ are the base rates. We model a plausible cost asymmetry, $C_{FP}=10$ and $C_{FN}=1$, to reflect the significant repercussions of misclassifying human-generated text. Table 1 below (Table 15 in the revised manuscript) reports the total expected cost across a range of plausible MGT base rates (10% to 50%), demonstrating that our method achieves a lower total expected cost in imbalanced real-world scenarios.
>
> **Table 1. Expected cost in the Essay dataset. Detectors are trained on ChatGPT text.**
>
> | Method | 10% | 20% | 30% | 40% | 50% |
> | --- | --- | --- | --- | --- | --- |
> | ChatGPT-D | 0.0635 | 0.0723 | 0.0939 | 0.0951 | 0.1070 |
> | **ChatGPT-STK** | **0.0450** | **0.0529** | **0.0574** | **0.0517** | **0.0550** |
> | OpenAI-D | 0.0561 | 0.0497 | 0.0490 | 0.0493 | 0.0430 |
> | **OpenAI-STK** | **0.0450** | **0.0400** | **0.0406** | **0.0348** | **0.0330** |
> | MPU | 0.0450 | 0.0400 | 0.0350 | 0.0300 | 0.0250 |
> | **MPU-STK** | 0.0450 | 0.0400 | 0.0350 | 0.0300 | 0.0250 |
> | RADAR | 0.0524 | 0.0561 | 0.0546 | 0.0637 | 0.0630 |
> | **RADAR-STK** | **0.0450** | **0.0400** | **0.0462** | **0.0469** | **0.0450** |
>
> Thank you again for your valuable time and constructive comments. We hope that the point-by-point responses above have clarified your main concerns, particularly our contribution in (Q1), the non-IID assumption in (Q3), and the concerns about AI and human separation in (Q4), which are **key analyses provided in the appendix of our initial submission**. Furthermore, we have fully adopted your valuable suggestion regarding (Q5), supplementing it with ROC curves and expected cost analysis in the revised manuscript. If our revision has satisfactorily addressed your concerns, we kindly request that you consider revising your final rating of our manuscript. If you have any additional concerns or comments that we may have missed in our responses, we would appreciate any further feedback to help us further enhance our work.
>
> [1] *Foundations of machine learning*. MIT press, 2018.
>
> [2]  Beyond the regret minimization barrier: optimal algorithms for stochastic strongly-convex optimization. *The Journal of Machine Learning Research*, 2014.

---

> > ### Author Response · Authors · 2025-11-28
> >
> > Dear Reviewer nLWU,
> >
> > Thank you once again for your valuable comments on our submission. **As the discussion phase is approaching its end**, we would like to kindly confirm whether we have sufficiently addressed all of your concerns (or at least part of them). Should there be any remaining questions or areas requiring further clarification, please do not hesitate to let us know. If our revision has satisfactorily addressed your concerns, we would greatly appreciate your consideration in adjusting the evaluation scores accordingly.
> >
> > We sincerely look forward to your feedback.
> >
> > Best regards,
> >
> > The authors

---

### Official Review · Reviewer_ZF6K · 2025-11-05

**Soundness:** 2
**Presentation:** 2
**Contribution:** 2
**Rating:** 4
**Confidence:** 2

**Summary:**

This paper challenges the common assumption in AI-generated text detection that a text is entirely machine- or human-generated. It reveals that human–machine mixed texts—where parts of a document resemble human writing even if produced by an LLM—are both common and harmful to detection accuracy.

**Strengths:**

(1) The work is theoretically sound, presenting a clear derivation that links mixed-text proportion to detection sample complexity and empirical accuracy.

**Weaknesses:**

(1) The analysis assumes independence among text segments, which may oversimplify linguistic dependencies in realistic mixed texts.
(2) The STK framework improves performance across detectors, but the paper does not deeply analyze what kinds of sentences are being filtered or how the stack changes the detector’s decision boundary.]
(3) The evaluation focuses mainly on English datasets and standard text genres (essay, news, QA). It remains unclear how well the mixed-text phenomenon and the proposed framework generalize to multilingual, multimodal, or code-switched content—settings increasingly relevant to global AI text use.
(4) while cross-LLM and paraphrase robustness are tested, the paper does not analyze computational efficiency (runtime, memory) or sensitivity to the number of stacked iterations.

**Questions:**

(1) Could you design an experiment where the degree of mixing is systematically varied (e.g., by concatenating known proportions of human and LLM sentences) to empirically validate the theoretical curve?
(2) How would your framework handle non-English or code-switched text, where sentence segmentation and tokenization are more ambiguous?
(3) Many real-world MGT scenarios involve human-edited or paraphrased AI outputs. How resilient is the stacked detection approach to such cases, where human post-editing blurs the segment boundary between human and machine?

---

> ### Author Response · Authors · 2025-11-17
> **Response to Reviewer ZF6K [1/3]**
>
> Thank you very much for the valuable feedback. Please find the point-to-point responses below. Any further comments and discussions are welcome!
>
> > **[W1]** The analysis assumes independence among text segments, which may oversimplify linguistic dependencies in realistic mixed texts.
> >
>
> In fact, we have carried out analyses under the non-IID setting in **Appendix D in the initial submission (also Appendix D in the revised version)**. From Theorems 3 (Appendix D.1) and 4 (Appendix D.2), we can derive conclusions similar to those obtained in the IID setting: the presence of mixed text ($\alpha>0$) hinders detection.
>
> > **[W2]** The STK framework improves performance across detectors, but the paper does not deeply analyze what kinds of sentences are being filtered or how the stack changes the detector’s decision boundary.
> >
>
> Thank you for this valuable comment. To explore this, we have provided specific analyses in **Appendix H.13 in the initial submission** (now Appendix H.15 in the revised version). Our study shows these filtered sentences tend to have fingerprints used to identify human-written text:
>
> - **Sentence Structure:** The human-like sequences (filtered by our framework) tend to be "relatively short and succinct". In contrast, the machine-generated parts often feature "more complex sentence structures, including frequent subordinate clauses and detailed arguments".
> - **Sentence Length:** We found that human-like sequences generally favor "short to medium-length sentences (around 12-15 words)". The machine-generated text, however, averages "around 20-25 words per sentence".
> - **Linguistic Style:** The human-like parts typically use "concise analytical language," whereas the machine-generated text is "more formal" and "frequently incorporates technical terms".
>
> Therefore, this filtering process improves the detector’s decision boundary through **iterative self-enhancement**: (1) in the E-step, the detector identifies and filters out high confidence “human” sequences, thereby supplying more distinctive remaining texts to the same detector in the M-step; (2) in the M-step, the detector learns on the remaining texts, leading to improvements that are used in the following E-step for even more accurate filtering of “human” sequences. This intertwined process leads to self-enhancement. Notably, we also have visualized the features extracted by the detectors before and after enhancement using T-SNE in Appendix H.8 of the initial submission (Appendix H.10 of the revised version). It can be seen that the enhanced detectors have clearer decision boundaries.
>
> > **[W3]** The evaluation focuses mainly on English datasets and standard text genres (essay, news, QA). It remains unclear how well the mixed-text phenomenon and the proposed framework generalize to multilingual, multimodal, or code-switched content—settings increasingly relevant to global AI text use.
> **[Q2]** How would your framework handle non-English or code-switched text, where sentence segmentation and tokenization are more ambiguous?
> >
>
> Thank you for this excellent suggestion, which prompted us to test the generalization of our framework beyond English. We agree that multilingual and code contexts are critical (noting that our focus is MGT detection rather than multimodal data detection). Therefore, we have conducted evaluations on Chinese texts (GPT3.5, Qwen, ChatGLM, Baichuan) [1] and code texts (ChatGPT) [2] as you suggested:
>
> - **Mixed-text Phenomenon.** We report the number of consistent sentences as shown in Tables 1 and 2 below.  We observe the same implicit mixing phenomenon, especially pronounced in the code texts. Notably, in the Chinese text evaluation, given the relatively small number of referenced human texts (24,525), the small number of consistent texts is still very valuable.
> - **Detection Performance.** Table 3 below presents the performance of both the original detectors and their enhanced versions (-STK) for detecting Chinese MGTs and code MGTs. We observe that the proposed enhancement strategy exhibits promising potential for both Chinese and code texts.
>
> We believe these new results strongly support the generalizability of our motivation and framework. We have included these experimental results in Appendix H.3 (Figure 10) and H.4 (Table 14) of the revised version to strengthen the paper.
>
> **Table 1. Implicit Mixed Chinese Text Evaluation.**
>
> | LLM | Number of Human Sentence | Number of Machine Sentence | Number of  Consistent Sentence |
> | --- | --- | --- | --- |
> | GPT3.5 | 24525 | 28431 | 183 |
> | Qwen | 24525 | 45500 | 112 |
> | ChatGLM | 24525 | 52628 | 36 |
> | Baichuan | 24525 | 32782 | 19 |
>
> **Table 2. Implicit Mixed Code Text Evaluation.**
>
> | LLM | Number of Human Sentence | Number of Machine Sentence | Number of  Consistent Sentence |
> | --- | --- | --- | --- |
> | ChatGPT | 21950 | 6329 | 1740 |

---

> ### Author Response · Authors · 2025-11-17
> **Response to Reviewer ZF6K [2/3]**
>
> **Table 3. Performance (AUROC) on Chinese texts and code texts. Detectors are trained
> on GPT3.5 (on Chinese texts) and ChatGPT (on code texts).**
>
> |  | Chinese Text |  |  |  |  | Code Text |
> | --- | --- | --- | --- | --- | --- | --- |
> |  | GPT3.5 | Baichuan | ChatGLM | Qwen | Avg. | ChatGPT |
> | ChatGPT-D | 92.45 | 94.60 | 76.70 | 92.91 | 89.16 | 99.35 |
> | **ChatGPT-STK** | **94.12** | **95.58** | **80.11** | **94.16** | **91.00** | **99.63** |
> | OpenAI-D | 92.75 | 96.57 | 87.51 | 93.83 | 92.66 | 99.50 |
> | **OpenAI-STK** | **95.76** | **97.93** | **88.55** | **95.04** | **94.32** | **99.53** |
> | MPU | **96.22** | **98.08** | 80.12 | **95.38** | 92.45 | 99.79 |
> | **MPU-STK** | 96.02 | 97.44 | **83.77** | 95.30 | **93.13** | **99.90** |
> | RADAR | 92.43 | 93.26 | 85.41 | 87.55 | 89.66 | 99.55 |
> | **RADAR-STK** | **92.65** | **93.28** | **85.48** | **88.51** | **89.98** | **99.64** |
>
> > **[W4]** while cross-LLM and paraphrase robustness are tested, the paper does not analyze computational efficiency (runtime, memory) or sensitivity to the number of stacked iterations.
> >
>
> Thank you for raising these important practical considerations. We have addressed all three points as follows:
>
> - **Computational Efficiency (Runtime):** In fact, we did include a detailed analysis of this from both theoretical and empirical perspectives in **Appendix F.6** and **H.12 of the initial submission** (Appendix F.6 and H.14 in the revised version). These results confirm that our framework's runtime does not exceed twice that of the original detector, which is consistent with its two-step design.
> - **Computational Efficiency (Memory):** Our framework is designed to be highly memory-efficient. The core idea is that the same detector model is used for both the E-step (filtering) and the M-step (final detection). Because no new models or large parameters are loaded, the peak memory usage is almost identical to that of the original detector. To validate this, we have **added an empirical memory evaluation**, as shown in Table 4 below (Table 20 in the revised version). The results confirm that the additional memory overhead from our framework is negligible.
> - **Analysis of “Stacked Iterations”: This is a key point of clarification.** The term "stacked" in our framework refers to the fixed, two-step architecture inspired by the Hard-EM algorithm, not a variable number of layers or iterations. Therefore, the number of "stacks" is fixed at two (one for the E-step, one for the M-step) and is a core part of the framework's design, not a tunable hyperparameter. A sensitivity analysis on this number is thus not applicable to our method.
>
> **Table 4. Memory consumption.**
>
> | Method | Essay | Reuters | SQuAD1 |
> | --- | --- | --- | --- |
> | ChatGPT-D/OpenAI-D/MPU | 9.36G | 9.36G | 9.15G |
> | ChatGPT-STK/OpenAI-STK/MPU-STK | 9.38G | 9.38G | 9.20G |
> | RADAR | 10.45G | 10.45G | 10.38G |
> | RADAR-STK | 10.47G | 10.47G | 10.42G |
>
> > **[Q1]** Could you design an experiment where the degree of mixing is systematically varied (e.g., by concatenating known proportions of human and LLM sentences) to empirically validate the theoretical curve?
> >
>
> This experiment is a key to empirically validating our theoretical findings (Theorem 1), so we did include this exact analysis in **Appendix H.7 of our initial submission** (now Appendix H.9 in the revised version). Specifically, we constructed test texts by replacing a known number of machine-generated sentences (ranging from 1 to 5) with their human-written counterparts. We can find that the performance of all detectors gradually decreases as the degree of text mixing increases. This provides a clear empirical validation for our theoretical claim that mixed text harms detection. We also observed that the superiority of our framework becomes even more significant in these challenging, high-mix scenarios, highlighting the effectiveness of our method.

---

> ### Author Response · Authors · 2025-11-17
> **Response to Reviewer ZF6K [3/3]**
>
> > **[Q3]** Many real-world MGT scenarios involve human-edited or paraphrased AI outputs. How resilient is the stacked detection approach to such cases, where human post-editing blurs the segment boundary between human and machine?
> >
>
> This is a critical point for any practical detection system. We agree on its importance and evaluated two distinct, related scenarios **in our initial submission**. Specifically, we explored robustness to paraphrase attacks in Section 4.2 and Appendix H.5 (now Appendix H.6 in the revised version) and to known mixed proportions in Appendix H.7 (now Appendix H.9 in the revised version). Our results show that the stacked framework surprisingly enhances robustness under these real-world scenarios.
> Our framework is robust to this "blurring" because the E-step learns to filter out any sequence that appears "human-like," regardless of its true origin. These scenarios aim to "humanize" machine text, while our case study (Appendix H.13, now Appendix H.15 in the revised version) shows that our framework can learn to filter these “human-like” sequences, thereby focusing on more discriminative machine features.
>
> Thank you again for your valuable time and constructive comments. We hope that the point-by-point responses above have clarified your concerns, **especially those (W1, W2, W4, Q1, Q3), which have been analyzed in detail in the initial submission**. If our revision has satisfactorily addressed your concerns, may we kindly request that you consider revising your final rating of our manuscript? If you have any additional concerns or comments that we may have missed in our responses, we would appreciate any further feedback to help us further enhance our work.
>
> [1] Towards Reliable Detection of LLM-Generated Texts: A Comprehensive Evaluation Framework with CUDRT. *arXiv 2024*.
>
> [2] The imitation game: Detecting human and AI-generated texts in the era of ChatGPT and BARD. *Journal of Information Science* 2024.

---

> > ### Author Response · Authors · 2025-11-28
> >
> > Dear Reviewer ZF6K,
> >
> > Thank you once again for your valuable comments on our submission. **As the discussion phase is approaching its end**, we would like to kindly confirm whether we have sufficiently addressed all of your concerns (or at least part of them). Should there be any remaining questions or areas requiring further clarification, please do not hesitate to let us know. If our revision has satisfactorily addressed your concerns, we would greatly appreciate your consideration in adjusting the evaluation scores accordingly.
> >
> > We sincerely look forward to your feedback.
> >
> > Best regards,
> >
> > The authors

---

### Author Response · Authors · 2025-12-03
**Summary of Reviewer Consensus and Rebuttal Improvements**

Dear Area Chair,

Thanks for handling our submission. Due to the unexpected bug in OpenReview, the current ratings do not reflect the major clarifications provided. Below is a summary of the strengths identified by the reviewers and how we addressed the primary concerns raised during the review process.

**Consensus on Strengths.** In summary, reviewers recognized the novelty of the problem formulation and the practical utility of our proposed framework.

- **Novelty & Significance:** Reviewers **122r** and **kQVt** highlighted that the investigation of implicit human-machine mixed texts is a "significant and compelling aspect" of MGT research and an "intriguing and very novel" idea. Reviewer **ZF6K** noted that challenging “the common assumption in AI-generated text detection that a text is entirely machine- or human-generated” is a key contribution.
- **Theoretical Foundation:** Reviewer **ZF6K** found the work "theoretically sound," praising the derivation linking mixed-text proportion to detection complexity. Reviewer **122r** also noted the analysis was "well-founded."
- **Empirical Performance:** Reviewers **nLWU** and **122r** commended the thorough experimental validation, noting "consistent gains in strict low-FPR regimes" and the practical utility of our "training-free" boosting strategy.

**Response Summary.** The primary concerns raised involved methodological clarity and generalization.

- **Theoretical Assumptions:** Reviewers **ZF6K**, **nLWU**, and **kQVt** questioned the IID assumption in our initial theoretical analysis. We clarified that we have carried out analyses under the non-IID setting in **Appendix D in the initial submission (also Appendix D in the revised version)**. From Theorems 3 (Appendix D.1) and 4 (Appendix D.2), we can derive conclusions similar to those obtained in the IID setting: the presence of mixed text ($\alpha>0$) hinders detection.
- **Methodological Clarity & Terminology:** Reviewers **kQVt** and **nLWU** were confused by our initial use of "human text" to describe machine-generated text that resembles human writing, and the role of Jaccard similarity in the algorithm. We clarified that **Jaccard similarity is NOT part of the detection pipeline**; it was solely a motivational metric. We replaced the confusing terminology with **"consistent text"** throughout the paper. We also completely rewrote Section 3.2 and Algorithm 1 to clearly distinguish the inference pipeline from the training process.
- **Generalization:** Reviewers **ZF6K** and **nLWU** asked how the framework handles non-English, code-switched, or short texts. We conducted new experiments on **Chinese datasets** and **Code generation**, as well as a **short-text (64-word limit)** scenario, as shown in Appendix H.3 and H.4 in the revised version.
- **Comparison with Prior Work:** Reviewer **122r** and **nLWU** requested better framing against prior mixed-text work (like RoFT) and sentence-level detectors. We revised the introduction to explicitly differentiate our focus on **implicit** mixing (latent consistency). We also added a comparison with **SeqXGPT** (a sentence-level detector), showing our framework can even boost SeqXGPT's own performance when applied as a wrapper.

This paper provides the first theoretical guarantee that implicit mixed text hinders detection and offers a plug-and-play framework that universally boosts existing detectors. The reviewers agreed on the novelty and strong empirical results. We have successfully addressed the clarity issues regarding the algorithm and terminology that led to the initially lower scores from specific reviewers.

We respectfully request that you consider these improvements and the positive consensus on the paper's core contribution when making your final decision.

Best Regards,

Authors

---

### Meta-Review · Area_Chair_SaXu · 2025-12-22

**Summary:**

The reviewers’ major concerns are about the assumptions of the theory, unclear presentation and confusing terminology, insufficient convincing validation of the proposed concept “implicit hybrid (consistent) text”, and gaps in empirical evaluation (generalizability to more settings, ablation studies, detailed metrics, and comparisons with more relevant mixed-authorship baselines).

**Reviewer Concerns:**

Some of the concerns were addressed during the rebuttal, such as assumptions of the theory. The authors provided additional experiments and explanations to improve the presentation and terminology, show generalizability, expand experimental settings, and include comparisons with more relevant baselines. However, the paper will benefit from more substantial revision for clarity and from a more comprehensive and integrated experimental evaluation.

**Reviewer Scores:**

Reviewer ZF6K may increase the scores a little bit. Reviewer nLWU, kQVt, ycey may or may not increase the score. Reviewer 122r may keep the positive score.

---

### Decision · Program_Chairs · 2026-01-26

Reject